# Learning Two-layer Neural Networks with Symmetric Inputs

**Rong Ge**
Computer Science Department
Duke University
`rongge@cs.duke.edu`

**Rohith Kuditipudi**
Computer Science Department
Duke University
`rohith.kuditipudi@duke.edu`

**Zhize Li**
Institute for Interdisciplinary Information Sciences
Tsinghua University
`zz-li14@mails.tsinghua.edu.cn`

**Xiang Wang**
Computer Science Department
Duke University
`xwang@cs.duke.edu`

## Abstract

We give a new algorithm for learning a two-layer neural network under a general class of input distributions. Assuming there is a ground-truth two-layer network

$$y = A\sigma(Wx) + \xi,$$

where $A, W$ are weight matrices, $\xi$ represents noise, and the number of neurons in the hidden layer is no larger than the input or output, our algorithm is guaranteed to recover the parameters $A, W$ of the ground-truth network. The only requirement on the input $x$ is that it is symmetric, which still allows highly complicated and structured input.

Our algorithm is based on the method-of-moments framework and extends several results in tensor decompositions. We use spectral algorithms to avoid the complicated non-convex optimization in learning neural networks. Experiments show that our algorithm can robustly learn the ground-truth neural network with a small number of samples for many symmetric input distributions.

## 1 Introduction

Deep neural networks have been extremely successful in many tasks related to images, videos and reinforcement learning. However, the success of deep learning is still far from being understood in theory. In particular, learning a neural network is a complicated non-convex optimization problem, which is hard in the worst-case. The question of whether we can efficiently learn a neural network still remains generally open, even when the data is drawn from a neural network. Despite a lot of recent effort, the class of neural networks that we know how to provably learn in polynomial time is still very limited, and many results require strong assumptions on the input distribution.

In this paper we design a new algorithm that is capable of learning a two-layer[1] neural network for a general class of input distributions. Following standard models for learning neural networks, we assume there is a ground truth neural network. The input data $(x, y)$ is generated by first sampling the input $x$ from an input distribution $\mathcal{D}$, then computing $y$ according to the ground truth network that is unknown to the learner. The learning algorithm will try to find a neural network $f$ such that $f(x)$ is as close to $y$ as possible over the input distribution $\mathcal{D}$. Learning a neural network is known to be a hard problem even in some simple settings (Goel et al., 2016; Brutzkus & Globerson, 2017), so we need to make assumptions on the network structure or the input distribution $\mathcal{D}$, or both. Many works have worked with a simple input distribution (such as Gaussians) and try to learn more and more

---

[1] There are different ways to count the number of layers. Here by two-layer network we refer to a fully-connected network with two layers of edges (two weight matrices). This is considered to be a three-layer network if one counts the number of layers for nodes (e.g. in Goel & Klivans (2017)) or a one-hidden layer network if one just counts the number of hidden layers.

complex networks (Tian, 2017; Brutzkus & Globerson, 2017; Li & Yuan, 2017; Soltanolkotabi, 2017; Zhong et al., 2017). However, the input distributions in real life are distributions of very complicated objects such as texts, images or videos. These inputs are highly structured, clearly not Gaussian and do not even have a simple generative model.

We consider a type of two-layer neural network, where the output $y$ is generated as

$$y = A\sigma(Wx) + \xi. \tag{1}$$

Here $x \in \mathbb{R}^d$ is the input, $W \in \mathbb{R}^{k \times d}$ and $A \in \mathbb{R}^{k \times k}$ are two weight matrices[2]. The function $\sigma$ is the standard ReLU activation function $\sigma(x) = \max\{x, 0\}$ applied entry-wise to the vector $Wx$, and $\xi$ is a noise vector that has $\mathbb{E}[\xi] = 0$ and is independent of $x$. Although the network only has two layers, learning similar networks is far from trivial: even when the input distribution is Gaussian, Ge et al. (2017b) and Safran & Shamir (2018) showed that standard optimization objective can have bad local optimal solutions. Ge et al. (2017b) gave a new and more complicated objective function that does not have bad local minima.

For the input distribution $\mathcal{D}$, our only requirement is that $\mathcal{D}$ is symmetric. That is, for any $x \in \mathbb{R}^d$, the probability of observing $x \sim \mathcal{D}$ is the same as the probability of observing $-x \sim \mathcal{D}$. A symmetric distribution can still be very complicated and cannot be represented by a finite number of parameters. In practice, one can often think of the symmetry requirement as a "factor-2" approximation to an *arbitrary input distribution*: if we have arbitrary training samples, it is possible to augment the input data with their negations to make the input distribution symmetric, and it should take at most twice the effort in labeling both the original and augmented data. In many cases (such as images) the augmented data can be interpreted (for images it will just be negated colors) so reasonable labels can be obtained.

## 1.1 OUR RESULTS

When the input distribution is symmetric, we give the first algorithm that can learn a two-layer neural network. Our algorithm is based on the method-of-moments approach: first estimate some correlations between $x$ and $y$, then use these information to recover the model parameters. More precisely we have

**Theorem 1** (informal)**.** *If the data is generated according to Equation* (1)*, and the input distribution $x \sim \mathcal{D}$ is symmetric. Given exact correlations between $x, y$ of order at most 4, as long as $A, W$ and input distribution are not degenerate, there is an algorithm that runs in poly($d$) time and outputs a network $\hat{A}, \hat{W}$ of the same size that is effectively the same as the ground-truth network: for any input $x$, $\hat{A}\sigma(\hat{W}x) = A\sigma(Wx)$.*

Of course, in practice we only have samples of $(x, y)$ and cannot get the exact correlations. However, our algorithm is robust to perturbations, and in particular can work with polynomially many samples.

**Theorem 2** (informal)**.** *If the data is generated according to Equation* (1)*, and the input distribution $x \sim \mathcal{D}$ is symmetric. As long as the weight matrices $A, W$ and input distributions are not degenerate, there is an algorithm that uses poly($d, 1/\epsilon$) time and number of samples and outputs a network $\hat{A}, \hat{W}$ of the same size that computes an $\epsilon$-approximation function to the ground-truth network: for any input $x$, $\|\hat{A}\sigma(\hat{W}x) - A\sigma(Wx)\|^2 \leq \epsilon$.*

In fact, the algorithm recovers the original parameters $A, W$ up to scaling and permutations. Here when we say weight matrices are not degenerate, we mean that the matrices $A, W$ should be full rank, and in addition a certain distinguishing matrix that we define later in Section 2 is also full rank. We justify these assumptions using the *smoothed analysis* framework (Spielman & Teng, 2004).

In smoothed analysis, the input is not purely controlled by an adversary. Instead, the adversary can first generate an arbitrary instance (in our case, arbitrary weight matrices $W, A$ and symmetric input distribution $\mathcal{D}$), and the parameters for this instance will be randomly perturbed to yield a perturbed instance. The algorithm only needs to work with high probability on the perturbed instance. This limits the power of the adversary and prevents it from creating highly degenerate cases (e.g. choosing the weight matrices to be much lower rank than $k$). Roughly speaking, we show

---

[2]Here we assume $A \in \mathbb{R}^{k \times k}$ for simplicity, our results can easily be generalized as long as the dimension of output is no smaller than the number of hidden units.

**Theorem 3** (informal). *There is a simple way to perturb the input distribution, $W$ and $A$ such that with high probability, the distance between the perturbed instance and original instance is at most $\lambda$, and our algorithm outputs an $\epsilon$-approximation to the perturbed network with $poly(d, 1/\lambda, 1/\epsilon)$ time and number of samples.*

In the rest of the paper, we will first review related works. Then in Section 2 we formally define the network and introduce some notations. Our algorithm is given in Section 3. Finally in Section 4 we run experiments to show that the algorithm can indeed learn the two-layer network efficiently and robustly. The experiments show that our algorithm works robustly with reasonable number of samples for different (symmetric) input distributions and weight matrices. Due to space constraints, the proof for polynomial number of samples (Theorem 2) and smoothed analysis (Theorem 3) are deferred to the appendix.

## 1.2  RELATED WORK

There are many works in learning neural networks, and they come in many different styles.

**Non-standard Networks**    Some works focus on networks that do not use standard activation functions. Arora et al. (2014) gave an algorithm that learns a network with discrete variables. Livni et al. (2014) and follow-up works learn neural networks with polynomial activation functions. Oymak & Soltanolkotabi (2018) used the rank-1 tensor decomposition for learning a non-overlapping convolutional neural network with differentiable and smooth activation and Gaussian input.

**ReLU network, Gaussian input**    When the input is Gaussian, Ge et al. (2017b) showed that for a two-layer neural network, although the standard objective does have bad local optimal solutions, one can construct a new objective whose local optima are all globally optimal. Several other works (Tian, 2017; Du et al., 2017b; Brutzkus & Globerson, 2017; Li & Yuan, 2017; Soltanolkotabi, 2017; Zhong et al., 2017; Zhang et al., 2018) extend this to different settings.

**General input with score functions**    A closely related work (Janzamin et al., 2015) does not require the input distribution to be Gaussian, but still relies on knowing the score function of the input distribution (which in general cannot be estimated efficiently from samples). Recently, Gao et al. (2018) gave a way to design loss functions with desired properties for one-hidden-layer neural networks with general input distributions based on a new proposed local likelihood score function estimator. For general distributions (including symmetric ones) their estimator can still require number of samples that is exponential in dimension $d$ (as in Assumption 1(d)).

**General input distributions**    There are several lines of work that try to extend the learning results to more general distributions. Du et al. (2017a) showed how to learn a single neuron or a single convolutional filter under some conditions for the input distribution. Daniely et al. (2016); Zhang et al. (2016; 2017); Goel & Klivans (2017); Du & Goel (2018) used kernel methods to learn neural networks when the norm of the weights and input distributions are both bounded (and in general the running time and sample complexity in this line of work depend exponentially on the norms of weights/input). Recently, Du et al. (2018) showed that gradient descent minimizes the training error in an over-parameterized two-layer neural network. They only consider training error while our results also apply to testing error. The work that is most similar to our setting is Goel et al. (2018), where they showed how to learn a single neuron (or a single convolutional filter) for any symmetric input distribution. Our two-layer neural network model is much more complicated.

**Method-of-Moments and Tensor Decomposition**    Our work uses method-of-moments, which has already been applied to learn many latent variable models (see Anandkumar et al. (2014) and references there). The particular algorithm that we use is inspired by an over-complete tensor decomposition algorithm FOOBI (De Lathauwer et al., 2007). Our smoothed analysis results are inspired by Bhaskara et al. (2014) and Ma et al. (2016), although our setting is more complicated and we need several new ideas.

## 2 PRELIMINARIES

In this section, we first describe the neural network model that we learn, and then introduce notations related to matrices and tensors. Finally we will define distinguishing matrix, which is a central object in our analysis.

### 2.1 NETWORK MODEL

We consider two-layer neural networks with $d$-dimensional input, $k$ hidden units and $k$-dimensional output, as shown in Figure 1. We assume that $k \leq d$. The input of the neural network is denoted by $x \in \mathbb{R}^d$. Assume that the input $x$ is i.i.d. drawn from a symmetric distribution $\mathcal{D}$[3]. Let the two weight matrices in the neural network be $W \in \mathbb{R}^{k \times d}$ and $A \in \mathbb{R}^{k \times k}$. The output $y \in \mathbb{R}^k$ is generated as follows:

$$y = A\sigma(Wx) + \xi, \tag{2}$$

where $\sigma(\cdot)$ is the element-wise ReLU function and $\xi \in \mathbb{R}^k$ is zero-mean random noise, which is independent with input $x$. Let the value of hidden units be $h \in \mathbb{R}^k$, which is equal to $\sigma(Wx)$. Denote $i$-th row of matrix $W$ as $w_i^\top$ ($i = 1, 2, ..., k$). Also, let $i$-th column of matrix $A$ be $a_i$ ($i = 1, 2, ..., k$). By property of ReLU activations, for any constant $c > 0$, scaling the $i$-th row of $W$ by $c$ while scaling the $i$-th column of $A$ by $1/c$ does not change the function computed by the network. Therefore without loss of generality, we assume every row vector of $W$ has unit norm.

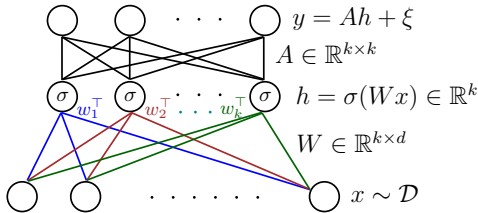

Figure 1: Network model.

### 2.2 NOTATIONS

We use $[n]$ to denote the set $\{1, 2, \cdots, n\}$. For two random variables $X$ and $Y$, we say $X \stackrel{d}{=} Y$ if they come from the same distribution.

In the vector space $\mathbb{R}^n$, we use $\langle \cdot, \cdot \rangle$ to denote the inner product of two vectors, and use $\|\cdot\|$ to denote the Euclidean norm. We use $e_i$ to denote the $i$-th standard basis vector. For a matrix $A \in \mathbb{R}^{m \times n}$, let $A_{[i,:]}$ denote its $i$-th row vector, and let $A_{[:,j]}$ denote its $j$-th column vector. Let $A$'s singular values be $\sigma_1(A) \geq \sigma_2(A) \geq \cdots \geq \sigma_{\min(m,n)}(A)$, and denote the smallest singular value be $\sigma_{\min}(A) = \sigma_{\min(m,n)}(A)$. The condition number of matrix $A$ is defined as $\kappa(A) := \sigma_1(A)/\sigma_{\min}(A)$. We use $I_n$ to denote the identity matrix with dimension $n \times n$. The spectral norm of a matrix is denoted as $\|\cdot\|$, and the Frobenius norm as $\|\cdot\|_F$.

We represent a $d$-dimensional linear subspace $\mathcal{S}$ by a matrix $S \in \mathbb{R}^{n \times d}$, whose columns form an orthonormal basis for subspace $\mathcal{S}$. The projection matrix onto the subspace $\mathcal{S}$ is denoted by $\mathrm{Proj}_S = SS^\top$, and the projection matrix onto the orthogonal subspace of $\mathcal{S}$ is denoted by $\mathrm{Proj}_{S^\perp} = I_n - SS^\top$.

For matrix $A \in \mathbb{R}^{m_1 \times n_1}, C \in R^{m_2 \times n_2}$, let the Kronecker product of $A$ and $C$ be $A \otimes C \in \mathbb{R}^{m_1 m_2 \times n_1 n_2}$, which is defined as $(A \otimes C)_{(i_1,i_2),(j_2,j_2)} = A_{i_1,i_2} C_{j_1,j_2}$. For a vector $x \in \mathbb{R}^d$, the Kronecker product $x \otimes x$ has dimension $d^2$. We denote the $p$-fold Kronecker product of $x$ as $x^{\otimes p}$, which has dimension $d^p$.

We often need to convert between vectors and matrices. For a matrix $A \in \mathbb{R}^{m \times n}$, let $\mathrm{vec}(A) \in \mathbb{R}^{mn}$ be the vector obtained by stacking all the columns of $A$. For a vector $a \in \mathbb{R}^{m^2}$, let $\mathrm{mat}(x) \in \mathbb{R}^{m \times m}$ denote the inverse mapping such that $\mathrm{vec}(\mathrm{mat}(a)) = a$. Let $\mathbb{R}_{sym}^{k \times k}$ be the space of all $k \times k$

---

[3]Suppose the density function of distribution $\mathcal{D}$ is $p(\cdot)$, we assume $p(x) = p(-x)$ for any $x \in \mathbb{R}^d$

symmetric matrices, which has dimension $\binom{k}{2} + k$. For convenience, we denote $k_2 = \binom{k}{2}$. For a symmetric matrix $B \in \mathbb{R}^{k \times k}_{sym}$, we denote $\text{vec}^*(B) \in \mathbb{R}^{k_2+k}$ as the vector obtained by stacking all the upper triangular entries (including diagonal entries) of $B$. Note that $\text{vec}(B)$ still has dimension $k^2$. For a vector $b \in \mathbb{R}^{k_2+k}$, let $\text{mat}^*(b) \in \mathbb{R}^{k \times k}_{sym}$ denote the inverse mapping of $\text{vec}^*(\cdot)$ such that $\text{vec}^*(\text{mat}^*(B)) = b$.

### 2.3 Distinguishing Matrix

A central object in our analysis is a large matrix whose columns are closely related to pairs of hidden variables. We call this the *distinguishing matrix* and define it below:

**Definition 1.** *Given a weight matrix $W$ of the first layer, and the input distribution $\mathcal{D}$, the distinguishing matrix $N^{\mathcal{D}} \in \mathbb{R}^{d^2 \times k_2}$ is a matrix whose columns are indexed by $ij$ where $1 \le i < j \le k$, and*

$$N^{\mathcal{D}}_{ij} = \mathbb{E}_{x \sim \mathcal{D}}\big[(w_i^\top x)(w_j^\top x)(x \otimes x)\mathbb{1}\{w_i^\top x w_j^\top x \le 0\}\big].$$

*Another related concept is the augmented distinguishing matrix $M$, which is a $d^2 \times (k_2 + 1)$ matrix whose first $k_2$ columns are exactly the same as distinguishing matrix $N$, and the last column (indexed by 0) is defined as*

$$M^{\mathcal{D}}_0 = \mathbb{E}_{x \sim \mathcal{D}}\big[x \otimes x\big].$$

*For both matrices, when the input distribution is clear from context we use $N$ or $M$ and omit the superscript.*

The exact reason for these definitions will only be clear after we explain the algorithm in Section 3. Our algorithm will require that these matrices are robustly full rank, in the sense that $\sigma_{min}(M)$ is lowerbounded. Intuitively, every column $N^{\mathcal{D}}_{ij}$ looks at the expectation over samples that have opposite signs for weights $w_i, w_j$ ($w_i^\top x w_j^\top x \le 0$, hence the name distinguishing matrix).

Requiring $M$ and $N$ to be full rank prevents several degenerate cases. For example, if two hidden units are perfectly correlated and always share the same sign for *every* input, this is very unnatural and requiring the distinguishing matrix to be full rank prevents such cases. Later in Section C we will also show that requiring a lowerbound on $\sigma_{min}(M)$ is not unreasonable: in the smoothed analysis setting where the nature can make a small perturbation on the input distribution $\mathcal{D}$, we show that for any input distribution $\mathcal{D}$, there exists simple perturbations $\mathcal{D}'$ that are arbitrarily close to $\mathcal{D}$ such that $\sigma_{min}(M^{D'})$ is lowerbounded.

## 3 Our Algorithm

In this section, we describe our algorithm for learning the two-layer networks defined in Section 2.1. As a warm-up, we will first consider a single-layer neural network and recover the results in Goel et al. (2018) using method-of-moments. This will also be used as a crucial step in our algorithm. Due to space constraints we will only introduce algorithm and proof ideas, the detailed proof is deferred to Section A in appendix. Throughout this section, when we use $\mathbb{E}[\cdot]$ without further specification the expectation is over the randomness $x \sim \mathcal{D}$ and the noise $\xi$.

### 3.1 Warm-up: Learning Single-layer Networks

We will first give a simple algorithm for learning a single-layer neural network. More precisely, suppose we are given samples $(x_1, y_1), ..., (x_n, y_n)$ where $x_i \sim \mathcal{D}$ comes from a symmetric distribution, and the output $y_i$ is computed by

$$y_i = \sigma(w^\top x_i) + \xi_i. \tag{3}$$

Here $\xi_i$'s are i.i.d. noises that satisfy $\mathbb{E}[\xi_i] = 0$. Noise $\xi_i$ is also assumed to be independent with input $x_i$. The goal is to learn the weight vector $w$.

The idea of the algorithm is simple: we will estimate the correlations between $x$ and $y$ and the covariance of $x$, and then recover the hidden vector $w$ using these two estimates. The main challenge here is that $y$ is not a linear function on $x$. Goel et al. (2018) gave a crucial observation that allows us to deal with the non-linearity:

---

**Algorithm 1** Learning Single-layer Neural Networks

---

**Input:** Samples $(x_1, y_1), ..., (x_n, y_n)$ generated according to Equation (3).
**Output:** Estimate of weight vector $w$.
  1: Estimate $v = \frac{1}{n} \sum_{i=1}^{n} y_i x_i$.
  2: Estimate $C = \frac{1}{n} \sum_{i=1}^{n} x_i x_i^\top$
  3: **return** $2C^{-1}v$.

---

**Lemma 1.** *Suppose $x \sim \mathcal{D}$ comes from a symmetric distribution and $y$ is computed as in* (3)*, then*

$$\mathbb{E}[yx] = \frac{1}{2}\mathbb{E}[xx^\top]w.$$

Importantly, the right hand side of Lemma 1 does not contain the ReLU function $\sigma$. This is true because if $x$ comes from a symmetric distribution, averaging between $x$ and $-x$ can get rid of non-linearities like ReLU or leaky-ReLU. Later we will prove a more general version of this lemma (Lemma 6).

Using this lemma, it is immediate to get a method-of-moments algorithm for learning $w$: we just need to estimate $\mathbb{E}[yx]$ and $\mathbb{E}[xx^\top]$, then we know $w = 2(\mathbb{E}[xx^\top])^{-1}\mathbb{E}[yx]$. This is summarized in Algorithm 1.

### 3.2 LEARNING TWO-LAYER NETWORKS

In order to learn the weights of the network defined in Section 2.1, a crucial observation is that we have $k$ outputs as well as $k$ hidden-units. This gives a possible way to reduce the two-layer problem to the single-layer problem. For simplicity, we will consider the noiseless case in this section, where

$$y = A\sigma(Wx). \tag{4}$$

Let $u \in \mathbb{R}^k$ be a vector and consider $u^\top y$, it is clear that $u^\top y = (u^\top A)\sigma(Wx)$. Let $z_i$ be the normalized version $i$-th row of $A^{-1}$, then we know $z_i$ has the property that $z_i^\top A = \lambda_i e_i^\top$ where $\lambda_i > 0$ is a constant and $e_i$ is a basis vector.

The key observation here is that if $u = z_i$, then $u^\top A = \lambda_i e_i^\top$. As a result, $u^\top y = \lambda_i e_i^\top \sigma(Wx) = \sigma(\lambda_i w_i^\top x)$ is the output of a single-layer neural network with weight equal to $\lambda_i w_i$. If we know all the vectors $\{z_1, ..., z_k\}$, the input/output pairs $(x, z_i^\top y)$ correspond to single-layer networks with weight vectors $\{\lambda_i w_i\}$. We can then apply the algorithm in Section 3.1 (or the algorithm in Goel et al. (2018)) to learn the weight vectors.

When $u^\top A = \lambda_i e_i$, we say that $u^\top y$ is a pure neuron. Next we will design an algorithm that can find all vectors $\{z_i\}$'s that generate pure neurons, and therefore reduce the problem of learning a two-layer network to learning a single-layer network.

**Pure Neuron Detector** In order to find the vector $u$ that generates a pure neuron, we will try to find some property that is true if and only if the output can be represented by a single neuron.

Intuitively, using ideas similar to Lemma 1 we can get a property that holds for all pure neurons:

**Lemma 2.** *Suppose $\hat{y} = \sigma(w^\top x)$, then $\mathbb{E}[\hat{y}^2] = \frac{1}{2}w^\top \mathbb{E}[xx^\top]w$, and $\mathbb{E}[\hat{y}x] = \frac{1}{2}\mathbb{E}[xx^\top]w$. As a result we have*

$$\mathbb{E}[\hat{y}^2] = 2\mathbb{E}[\hat{y}x^\top]\mathbb{E}[xx^\top]^{-1}\mathbb{E}[\hat{y}x].$$

As before, the ReLU activation does not appear because of the symmetric input distribution. For $\hat{y} = u^\top y$, we can estimate all of these moments ($\mathbb{E}[\hat{y}^2], \mathbb{E}[\hat{y}x^\top], \mathbb{E}[xx^\top]$) using samples and check whether this condition is satisfied. However, the problem with this property is that even if $z = u^\top y$ is not pure, it may still satisfy the property. More precisely, if $\hat{y} = \sum_{i=1}^{k} c_i \sigma(w_i^\top x)$, then we have

$$2\mathbb{E}[\hat{y}x^\top]\mathbb{E}[xx^\top]^{-1}\mathbb{E}[\hat{y}x] - \mathbb{E}[\hat{y}^2] = \sum_{1 \le i < j \le k} c_i c_j \mathbb{E}\big[(w_i^\top x)(w_j^\top x)\mathbb{1}\{w_i^\top x w_j^\top x \le 0\}\big]$$

The additional terms may accidentally cancel each other which leads to a false positive. To address this problem, we consider a higher order moment:

**Lemma 3.** *Suppose $\hat{y} = \sigma(w^\top x)$, then*

$$\mathbb{E}[\hat{y}^2(x \otimes x)] = 2\mathbb{E}\left[\hat{y} \cdot (\mathbb{E}[\hat{y}x^\top]\mathbb{E}[xx^\top]^{-1}x) \cdot (x \otimes x)\right].$$

*Moreover, if $\hat{y} = \sum_{i=1}^k c_i \sigma(w_i^\top x)$ where $c \in \mathbb{R}^k$ is a $k$-dimensional vector, we have*

$$2\mathbb{E}\left[\hat{y} \cdot (\mathbb{E}[\hat{y}x^\top]\mathbb{E}[xx^\top]^{-1}x) \cdot (x \otimes x)\right] - \mathbb{E}[\hat{y}^2(x \otimes x)] = \sum_{1 \le i < j \le k} c_i c_j N_{ij}.$$

*Here $N_{ij}$'s are columns of the distinguishing matrix defined in Definition 1.*

The important observation here is that there are $k_2 = \binom{k}{2}$ extra terms in $2\mathbb{E}\left[\hat{y} \cdot (\mathbb{E}[\hat{y}x^\top]\mathbb{E}[xx^\top]^{-1}x) \cdot (x \otimes x)\right] - \mathbb{E}[\hat{y}^2(x \otimes x)]$ that are multiples of $N_{ij}$, which are $d^2$ (or $\binom{d+1}{2}$ considering their symmetry) dimensional objects. When the distinguishing matrix is full rank, we know its columns $N_{ij}$ are linearly independent. In that case, if the sum of the extra terms is 0, then the coefficient in front of each $N_{ij}$ must also be 0. The coefficients are $c_i c_j$ which will be non-zero if and only if both $c_i, c_j$ are non-zero, therefore to make all the coefficients 0 at most one of $\{c_i\}$ can be non-zero. This is summarized in the following Corollary:

**Corollary 1** (Pure Neuron Detector)**.** *Define $f(u) := 2\mathbb{E}\left[(u^\top y) \cdot (\mathbb{E}[(u^\top y)x^\top]\mathbb{E}[xx^\top]^{-1}x) \cdot (x \otimes x)\right] - \mathbb{E}[(u^\top y)^2(x \otimes x)]$. Suppose the distinguishing matrix is full rank, if $f(u) = 0$ for unit vector $u$, then $u$ must be equal to one of $\pm z_i$.*

We will call the function $f(u)$ a *pure neuron detector*, as $u^\top y$ is a pure neuron if and only if $f(u) = 0$. Therefore, to finish the algorithm we just need to find all solutions for $f(u) = 0$.

**Linearization** The main obstacle in solving the system of equations $f(u) = 0$ is that every entry of $f(u)$ is a quadratic function in $u$. The system of equations $f(u) = 0$ is therefore a system of quadratic equations. Solving a generic system of quadratic equations is NP-hard. However, in our case this can be solved by a technique that is very similar to the FOOBI algorithm for tensor decomposition (De Lathauwer et al., 2007). The key idea is to linearize the function by thinking of each degree 2 monomial $u_i u_j$ as a separate variable. Now the number of variables is $k_2 + k = \binom{k}{2} + k$ and $f$ is linear in this space. In other words, there exists a matrix $T \in \mathbb{R}^{d^2 \times (k_2+k)}$ such that $T\text{vec}^*(uu^\top) = f(u)$. Clearly, if $u^\top y$ is a pure neuron, then $T\text{vec}^*(uu^\top) = f(u) = 0$. That is, $\{\text{vec}^*(z_i z_i^\top)\}$ are all in the nullspace of $T$. Later in Section A we will prove that the nullspace of $T$ consists of exactly these vectors (and their combinations):

**Lemma 4.** *Let $T$ be the unique $\mathbb{R}^{d^2 \times (k_2+k)}$ matrix that satisfies $T\text{vec}^*(uu^\top) = f(u)$ (where $f(u)$ is defined as in Corollary 1), suppose the distinguishing matrix is full rank, then the nullspace of $T$ is exactly the span of $\{\text{vec}^*(z_i z_i^\top)\}$.*

Based on Lemma 4, we can just estimate the tensor $T$ from the samples we are given, and its smallest singular directions would give us the span of $\{\text{vec}^*(z_i z_i^\top)\}$.

**Finding $z_i$'s from span of $z_i z_i^\top$'s** In order to reduce the problem to a single-layer problem, the final step is to find $z_i$'s from span of $z_i z_i^\top$'s. This is also a step that has appeared in FOOBI and more generally other tensor decomposition algorithms, and can be solved by a simultaneous diagonalization. Let $Z$ be the matrix whose rows are $z_i$'s, which means $Z = \text{diag}(\lambda)A^{-1}$. Let $X = Z^\top D_X Z$ and $Y = Z^\top D_Y Z$ be two random elements in the span of $z_i z_i^\top$, where $D_X$ and $D_Y$ are two random diagonal matrices. Both matrices $X$ and $Y$ can be diagonalized by matrix $Z$. In this case, if we compute $XY^{-1} = Z^\top D_X D_Y^{-1}(Z^\top)^{-1}$, since $z_i$ is a column of $Z^\top$, we know

$$XY^{-1}z_i = Z^\top D_X D_Y^{-1}(Z^\top)^{-1}z_i = Z^\top D_X D_Y^{-1}e_i = \frac{D_X(i,i)}{D_Y(i,i)}Z^\top e_i = \frac{D_X(i,i)}{D_Y(i,i)}z_i.$$

That is, $z_i$ is an eigenvector of $XY^{-1}$! The matrix $XY^{-1}$ can have at most $k$ eigenvectors and there are $k$ $z_i$'s, therefore the $z_i$'s are the only eigenvectors of $XY^{-1}$.

**Lemma 5.** *Given the span of $z_i z_i^\top$'s, let $X, Y$ be two random matrices in this span, with probability 1 the $z_i$'s are the only eigenvectors of $XY^{-1}$.*

Using this procedure we can find all the $z_i$'s (up to permutations and sign flip). Without loss of generality we assume $z_i^\top A = \lambda_i e_i^\top$. The only remaining problem is that $\lambda_i$ might be negative. However, this is easily fixable by checking $\mathbb{E}[z_i^\top y] = \lambda_i \mathbb{E}[\sigma(w_i^\top x)]$. Since $\sigma(w_i^\top x)$ is always nonnegative, $\mathbb{E}[z_i^\top y]$ has the same sign as $\lambda_i$, and we can flip $z_i$ if $\mathbb{E}[z_i^\top y]$ is negative.

### 3.3 Detailed Algorithm and Guarantees

We can now give the full algorithm, see Algorithm 2. The main steps of this algorithm is as explained in the previous section. Steps 2 - 5 constructs the pure neuron detector and finds the span of $\mathrm{vec}^*(z_i z_i^\top)$ (as in Corollary 1); Steps 7 - 9 performs simultaneous diagonalization to get all the $z_i$'s; Steps 11, 12 calls Algorithm 1 to solve the single-layer problem and outputs the correct result.

---

**Algorithm 2** Learning Two-layer Neural Networks

**Input:** Samples $(x_1, y_1), ..., (x_n, y_n)$ generated according to Equation (4)
**Output:** Weight matrices $W$ and $A$.
1: {Finding span of $\mathrm{vec}(z_i z_i^\top)$'s}
2: Estimate empirical moments $\hat{\mathbb{E}}[xx^\top]$, $\hat{\mathbb{E}}[yx^\top]$, $\hat{\mathbb{E}}[y \otimes x^{\otimes 3}]$ and $\hat{\mathbb{E}}[y \otimes y \otimes (x \otimes x)]$.
3: Compute the vector $f(u) = 2\hat{\mathbb{E}}\big[(u^\top y) \cdot (\hat{\mathbb{E}}[(u^\top y)x^\top]\hat{\mathbb{E}}[xx^\top]^{-1}x) \cdot (x \otimes x)\big] - \hat{\mathbb{E}}\big[(u^\top y)^2(x \otimes x)\big]$
   where each entry is expressed as a degree-2 polynomial over $u$.
4: Construct matrix $T \in \mathbb{R}^{d^2 \times (k_2 + k)}$ such that, $T\mathrm{vec}^*(uu^\top) = f(u)$.
5: Compute the $k$-least right singular vectors of $T$, denoted by $\mathrm{vec}^*(U_1), \mathrm{vec}^*(U_2), \cdots, \mathrm{vec}^*(U_k)$. Let $S$ be a $k_2 + k$ by $k$ matrix, where the $i$-th column of $S$ is vector $\mathrm{vec}^*(U_i)$. {Here span $S$ is equal to span of $\{\mathrm{vec}^*(z_i z_i^\top)\}$.}
6: {Finding $z_i$'s from span}
7: Let $X = \mathrm{mat}^*(S\zeta_1)$, $Y = \mathrm{mat}^*(S\zeta_2)$, where $\zeta_1$ and $\zeta_2$ are two independently sampled $k$-dimensional standard Gaussian vectors.
8: Let $z_1, ..., z_k$ be eigenvectors of $XY^{-1}$.
9: For each $z_i$, if $\hat{\mathbb{E}}[z_i^\top y] < 0$ let $z_i \leftarrow -z_i$. {Here $z_i$'s are normalized rows of $A^{-1}$.}
10: {Reduce to 1-Layer Problem}
11: For each $z_i$, let $v_i$ be the output of Algorithm 1 with input $(x_1, z_i^\top y_1), ..., (x_n, z_i^\top y_n)$.
12: Let $Z$ be the matrix whose rows are $z_i^\top$'s, $V$ be the matrix whose rows are $v_i^\top$'s.
13: **return** $V, Z^{-1}$.

---

We are now ready to state a formal version of Theorem 1:

**Theorem 4.** *Suppose $A, W, \mathbb{E}[xx^\top]$ and the distinguishing matrix $N$ are all full rank, and Algorithm 2 has access to the exact moments, then the network returned by the algorithm computes exactly the same function as the original neural network.*

It is easy to prove this theorem using the lemmas we have.

*Proof.* By Corollary 1, we know that after Step 5 of Algorithm 2, the span of columns of $S$ is exactly equal to the span of $\{\mathrm{vec}^*(z_i z_i^\top)\}$. By Lemma 5, we know the eigenvectors of $XY^{-1}$ at Step 8 are exactly the normalized version of rows of $A^{-1}$. Without loss of generality, we will fix the permutation and assume $z_i^\top A = \lambda_i e_i^\top$. In Step 9, we use the fact that $\mathbb{E}[z_i^\top y] = \lambda_i \mathbb{E}[\sigma(w_i^\top x)]$ where $\mathbb{E}[\sigma(w_i^\top x)]$ is always positive because $\sigma$ is the ReLU function. Therefore, after Step 9 we can assume all the $\lambda_i$'s are positive.

Now the output $z_i^\top y = \lambda_i \sigma(w_i^\top x) = \sigma(\lambda_i w_i^\top x)$ (again by property of ReLU function $\sigma$), by the design of Algorithm 1 we know $v_i = \lambda_i w_i$. We also know that $Z = \mathrm{diag}(\lambda)A^{-1}$, therefore $Z^{-1} = A\mathrm{diag}(\lambda)^{-1}$. Notice that $Z^{-1}\sigma(Vx) = A\mathrm{diag}(\lambda)^{-1}\sigma(\mathrm{diag}(\lambda)Wx) = A\sigma(Wx)$. These two scaling factors cancel each other, so the two networks compute the same function. $\square$

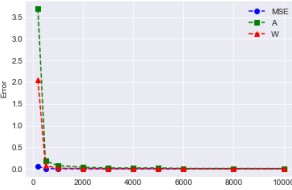 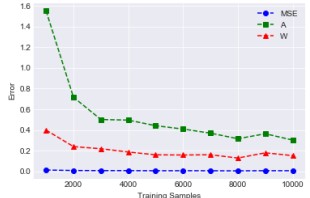 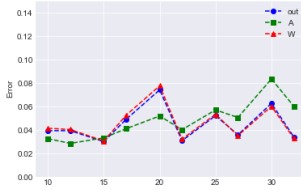

Figure 2: Error in recovering $W$, $A$ and outputs ("MSE") for different numbers of training samples and different dimensions of $W$ and $A$. Each point is the result of averaging across five trials, where on the left $W$ and $A$ are both drawn as random $10 \times 10$ orthonormal matrices and in the center as $32 \times 32$ orthonormal matrices. On the right, given $10,000$ training samples we plot the square root of the algorithm's error normalized by the dimension of $W$ and $A$, which are again drawn as random orthonormal matrices. The input distribution is a spherical Gaussian.

## 4 EXPERIMENTS

In this section, we provide experimental results to validate the robustness of our algorithm for both Gaussian input distributions as well as more general symmetric distributions such as symmetric mixtures of Gaussians.

There are two important ways in which our implementation differs from our description in Section 3.3. First, our description of the simultaneous diagonalization step in our algorithm is mostly for simplicity of both stating and proving the algorithm. In practice we find it is more robust to draw $10k$ random samples from the subspace spanned by the last $k$ right-singular vectors of $T$ and compute the CP decomposition of all the samples (reshaped as matrices and stacked together as a tensor) via alternating least squares (Comon et al., 2009). As alternating least squares can also be unstable we repeat this step 10 times and select the best one. Second, once we have recovered and fixed $A$ we use gradient descent to learn $W$, which compared to Algorithm 1 does a better job of ensuring the overall error will not explode even if there is significant error in recovering $A$. Crucially, these modifications are not necessary when the number of samples is large enough. For example, given 10,000 input samples drawn from a spherical Gaussian and $A$ and $W$ drawn as random $10 \times 10$ orthogonal matrices, our implementation of the original formulation of the algorithm was still able to recover both $A$ and $W$ with an average error of approximately 0.15 and achieve close to zero mean square error across 10 random trials.

### 4.1 SAMPLE EFFICIENCY

First we show that our algorithm does not require a large number of samples when the matrices are not degenerate. In particular, we generate random orthonormal matrices $A$ and $W$ as the ground truth, and use our algorithm to learn the neural network. As illustrated by Figure 2, regardless of the size of $W$ and $A$ our algorithm is able to recover both weight matrices with minimal error so long as the number of samples is a few times of the number of parameters. To measure the error in recovering $A$ and $W$, we first normalize the columns of $A$ and rows of $W$ for both our learned parameters and the ground truth, pair corresponding columns and rows together, and then compute the squared distance between learned and ground truth parameters. Note in the rightmost plot of Figure 2, in order to compare the performance between different dimensions, we further normalize the recovering error by the dimension of $W$ and $A$. It shows that the squared root of normalized error remains stable as the dimension of $A$ and $W$ grows from 10 to 32. In Figure 2, we also show the overall mean square error–averaged over all output units–achieved by our learned parameters.

### 4.2 ROBUSTNESS TO NOISE

Figure 3 demonstrates the robustness of our algorithm to label noise $\xi$ for Gaussian and symmetric mixture of Gaussians input distributions. In this experiment, we fix the size of both $A$ and $W$ to be $10 \times 10$ and again generate both parameters as random orthonormal matrices. The overall mean square error achieved by our algorithm grows almost perfectly in step with the amount of label noise,

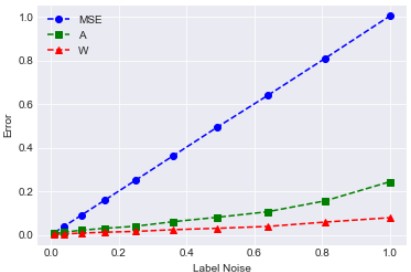 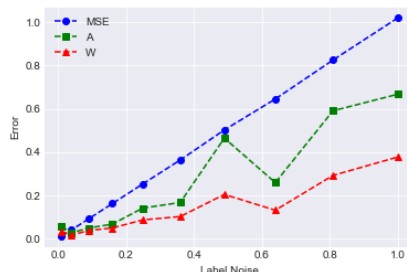

Figure 3: Error in recovering $W$, $A$ and outputs ("MSE") for different amounts of label noise. Each point is the result of averaging across five trials with 10,000 training samples, where for each trial $W$ and $A$ are both drawn as $10 \times 10$ orthonormal matrices. The input distribution on the left is a spherical Gaussian and on the right a mixture of two Gaussians with one component based at the all-ones vector and the other component at its reflection.

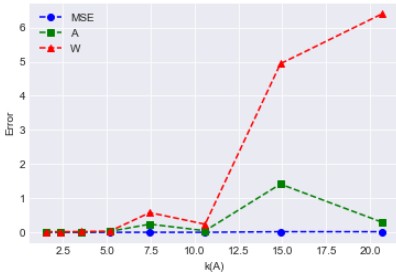 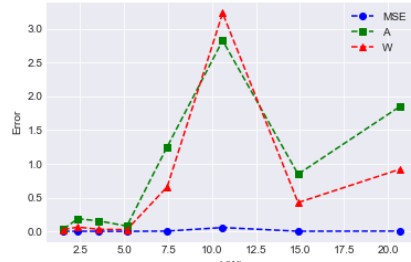

Figure 4: Error in recovering $W$, $A$ and outputs ("MSE"), on the left for different levels of conditioning of $W$ and on the right for $A$. Each point is the result of averaging across five trials with 20,000 training samples, where for each trial one parameter is drawn as a random orthonormal matrix while the other as described in Section 4.3. The input distribution is a mixture of Gaussians with two components, one based at the all-ones vector and the other at its reflection.

indicating that our algorithm recovers the globally optimal solution regardless of the choice of input distribution.

## 4.3 ROBUSTNESS TO CONDITION NUMBER

We've already shown that our algorithm continues to perform well across a range of input distributions and even when $A$ and $W$ are high-dimensional. In all previous experiments however, we sampled $A$ and $W$ as random orthonormal matrices so as to control for their conditioning. In this experiment, we take the input distribution to be a random symmetric mixture of two Gaussians and vary the condition number of either $A$ or $W$ by sampling singular value decompositions $U\Sigma V^\top$ such that $U$ and $V$ are random orthonormal matrices and $\Sigma_{ii} = \lambda^{-i}$, where $\lambda$ is chosen based on the desired condition number. Figure 4 respectively demonstrate that the performance of our algorithm remains steady so long as $A$ and $W$ are reasonably well-conditioned before eventually fluctuating. Moreover, even with these fluctuations the algorithm still recovers $A$ and $W$ with sufficient accuracy to keep the overall mean square error low.

## 5 CONCLUSION

Optimizing the parameters of a neural network is a difficult problem, especially since the objective function depends on the input distribution which is often unknown and can be very complicated. In this paper, we design a new algorithm using method-of-moments and spectral techniques to avoid the

complicated non-convex optimization for neural networks. Our algorithm can learn a network that is of similar complexity as the previous works, while allowing much more general input distributions.

There are still many open problems. The current result requires output to have the same (or higher) dimension than the hidden layer, and the hidden layer does not have a bias term. Removing these constraints are are immediate directions for future work. Besides the obvious ones of extending our results to more general distributions and more complicated networks, we are also interested in the relations to optimization landscape for neural networks. In particular, our algorithm shows there is a way to find the global optimal network in polynomial time, does that imply anything about the optimization landscape of the standard objective functions for learning such a neural network, or does it imply there exists an alternative objective function that does not have any local minima? We hope this work can lead to new insights for optimizing a neural network.

## ACKNOWLEDGEMENT

This work was supported by NSF CCF-1704656.

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

# A    DETAILS OF EXACT ANALYSIS

In this section, we first provide the missing proofs for the lemmas appeared in Section 3. Then we discuss how to handle the noise case (i.e. $y = \sigma(Wx) + \xi$) and give the corresponding algorithm (Algorithm 3). At the end we also briefly discuss how to handle the case when the matrix $A$ has more rows than columns (more outputs than hidden units).

Again, throughout the section when we write $\mathbb{E}[\cdot]$, the expectation is taken over the randomness of $x \sim \mathcal{D}$ and noise $\xi$.

## A.1    MISSING PROOFS FOR SECTION 3

**Single-layer:** To get rid of the non-linearities like ReLU, we use the property of the symmetric distribution (similar to (Goel et al., 2018)). Here we provide a more general version (Lemma 6) instead of proving the specific Lemma 1. Note that Lemma 1 is the special case when $a = w$ and $p = q = 1$ (here $\xi$ does not affect the result since it has zero mean and is independent with $x$, thus $\mathbb{E}[yx] = \mathbb{E}[\sigma(w^\top x)x]$).

**Lemma 6.** *Suppose input $x$ comes from a symmetric distribution, for any vector $a \in \mathbb{R}^d$ and any non-negative integers $p$ and $q$ satisfying that $p + q$ is an even number, we have*

$$\mathbb{E}\big[(\sigma(a^\top x))^p x^{\otimes q}\big] = \frac{1}{2}\mathbb{E}[(a^\top x)^p x^{\otimes q}],$$

*where the expectation is taken over the input distribution.*

*Proof.* Since input $x$ comes from a symmetric distribution, we know that $\mathbb{E}\big[(\sigma(a^\top x))^p x^{\otimes q}\big] = \mathbb{E}\big[(\sigma(-a^\top x))^p(-x)^{\otimes q}\big]$. Thus, we have

$$\mathbb{E}\big[(\sigma(a^\top x))^p x^{\otimes q}\big] = \frac{1}{2}\Big(\mathbb{E}\big[(\sigma(-a^\top x))^p(-x)^{\otimes q}\big] + \mathbb{E}\big[(\sigma(a^\top x))^p x^{\otimes q}\big]\Big).$$

There are two cases to consider: $p$ and $q$ are both even numbers or both odd numbers.

1. For the case where $p$ and $q$ are even numbers, we have

$$\frac{1}{2}\big(\mathbb{E}\big[(\sigma(-a^\top x))^p(-x)^{\otimes q}\big] + \mathbb{E}\big[(\sigma(a^\top x))^p x^{\otimes q}\big]\big)$$
$$= \frac{1}{2}\big(\mathbb{E}\big[(\sigma(-a^\top x))^p x^{\otimes q}\big] + \mathbb{E}\big[(\sigma(a^\top x))^p x^{\otimes q}\big]\big)$$
$$= \frac{1}{2}\mathbb{E}\Big[\big((\sigma(-a^\top x))^p + (\sigma(a^\top x))^p\big)x^{\otimes q}\Big].$$

If $(a^\top x) \le 0$, we know $(\sigma(-a^\top x))^p + (\sigma(a^\top x))^p = (a^\top x)^p + 0 = (a^\top x)^p$. Otherwise, we have $(\sigma(-a^\top x))^p + (\sigma(a^\top x))^p = 0 + (a^\top x)^p = (a^\top x)^p$. Thus,

$$\mathbb{E}\big[(\sigma(a^\top x))^p x^{\otimes q}\big] = \frac{1}{2}\mathbb{E}\Big[\big((\sigma(-a^\top x))^p + (\sigma(a^\top x))^p\big)x^{\otimes q}\Big]$$
$$= \frac{1}{2}\mathbb{E}[(a^\top x)^p x^{\otimes q}].$$

2. For the other case where $p$ and $q$ are odd numbers, we have

$$\frac{1}{2}\big(\mathbb{E}\big[(\sigma(-a^\top x))^p(-x)^{\otimes q}\big] + \mathbb{E}\big[(\sigma(a^\top x))^p x^{\otimes q}\big]\big)$$
$$= \frac{1}{2}\mathbb{E}\Big[\big(-(\sigma(-a^\top x))^p + (\sigma(a^\top x))^p\big)x^{\otimes q}\Big].$$

Similarly, if $(a^\top x) \le 0$, we know $-(\sigma(-a^\top x))^p + (\sigma(a^\top x))^p = -(-a^\top x)^p + 0 = (a^\top x)^p$. Otherwise, we have $-(\sigma(-a^\top x))^p + (\sigma(a^\top x))^p = 0 + (a^\top x)^p = (a^\top x)^p$. Thus,

$$\mathbb{E}\big[(\sigma(a^\top x))^p x^{\otimes q}\big] = \frac{1}{2}\mathbb{E}\Big[\big(-(\sigma(-a^\top x))^p + (\sigma(a^\top x))^p\big)x^{\otimes q}\Big]$$
$$= \frac{1}{2}\mathbb{E}[(a^\top x)^p x^{\otimes q}].$$

$\square$

**Pure neuron detector:** The first step in our algorithm is to construct a pure neuron detector based on Lemma 2 and Lemma 3. We will provide proofs for these two lemmas here.

**Proof of Lemma 2.** This proof easily follows from Lemma 6. Setting $a = w$, $p = 2$ and $q = 0$ in Lemma 6, we have $\mathbb{E}[\hat{y}^2] = \frac{1}{2} w^\top \mathbb{E}[xx^\top] w$. Similarly, we also know $\mathbb{E}[\hat{y}x^\top] = \frac{1}{2} w^\top \mathbb{E}[xx^\top]$ and $\mathbb{E}[\hat{y}x] = \frac{1}{2}\mathbb{E}[xx^\top]w$. Thus, we have $\mathbb{E}[\hat{y}^2] = 2\mathbb{E}[\hat{y}x^\top]\mathbb{E}[xx^\top]^{-1}\mathbb{E}[\hat{y}x]$. $\square$

**Proof of Lemma 3.** Here, we only prove the second equation, since the first equation is just a special case of the second equation. First, we rewrite $\hat{y} = \sum_{i=1}^{k} c_i \sigma(w_i^\top x) = u^\top y$ by letting $u^\top A = c^\top$. Then we transform these two terms in the LHS as follows. Let's look at $\mathbb{E}\big[(u^\top y) \cdot (\mathbb{E}[(u^\top y)x^\top]\mathbb{E}[xx^\top]^{-1}x) \cdot (x \otimes x)\big]$ first. For $\mathbb{E}[(u^\top y)x^\top]\mathbb{E}[xx^\top]^{-1}$, we have

$$\mathbb{E}\big[(u^\top y)x^\top\big]\mathbb{E}\big[xx^\top\big]^{-1}$$
$$=\mathbb{E}\big[(u^\top A\sigma(Wx))x^\top\big]\mathbb{E}\big[xx^\top\big]^{-1}$$
$$=\frac{1}{2}u^\top AW\mathbb{E}\big[xx^\top\big]\mathbb{E}\big[xx^\top\big]^{-1}$$
$$=\frac{1}{2}u^\top AW$$

For any vector $q \in \mathbb{R}^d$, consider $2\mathbb{E}\big[(u^\top y) \cdot (q^\top x) \cdot (x \otimes x)\big]$. We have

$$2\mathbb{E}\big[(u^\top y) \cdot (q^\top x) \cdot (x \otimes x)\big]$$
$$=2\mathbb{E}[(u^\top A\sigma(Wx))(q^\top x)(x \otimes x)]$$
$$=\mathbb{E}\big[(u^\top AWx)(q^\top x)(x \otimes x)\big]$$

Let $q^\top = \mathbb{E}\big[(u^\top y)x^\top\big]\mathbb{E}\big[xx^\top\big]^{-1} = \frac{1}{2}u^\top AW$, we have

$$2\mathbb{E}\big[(u^\top y) \cdot (\mathbb{E}[(u^\top y)x^\top]\mathbb{E}[xx^\top]^{-1}x) \cdot (x \otimes x)\big]$$
$$=\frac{1}{2}\mathbb{E}\big[(u^\top AWx)^2(x \otimes x)\big]$$
$$=\frac{1}{2}\mathbb{E}\big[\langle A^\top u, Wx\rangle^2(x \otimes x)\big]$$
$$=\frac{1}{2}\sum_{1 \le i \le k}(A^\top u)_i^2\mathbb{E}\big[(w_i^\top x)^2(x \otimes x)\big] + \sum_{1 \le i < j \le k}(A^\top u)_i(A^\top u)_j\mathbb{E}\big[(w_i^\top x)(w_j^\top x)(x \otimes x)\big]$$
$$=\frac{1}{2}\sum_{1 \le i \le k}(A^\top uu^\top A)_{ii}\mathbb{E}\big[(w_i^\top x)^2(x \otimes x)\big] + \sum_{1 \le i < j \le k}(A^\top uu^\top A)_{ij}\mathbb{E}\big[(w_i^\top x)(w_j^\top x)(x \otimes x)\big].$$
$$(5)$$

where the second equality holds due to Lemma 6.

Now, let's look at the second term $\mathbb{E}\big[(u^\top y)^2(x \otimes x)\big]$.

$$\mathbb{E}\big[(u^\top y)^2(x \otimes x)\big]$$
$$=\mathbb{E}\big[(u^\top A\sigma(Wx))^2(x \otimes x)\big]$$
$$=\mathbb{E}\big[\langle A^\top u, \sigma(Wx)\rangle^2(x \otimes x)\big]$$
$$=\sum_{1 \le i \le k}(A^\top u)_i^2\mathbb{E}\big[\sigma(w_i^\top x)^2(x \otimes x)\big] + 2\sum_{1 \le i < j \le k}(A^\top u)_i(A^\top u)_j\mathbb{E}\big[\sigma(w_i^\top x)\sigma(w_j^\top x)(x \otimes x)\big]$$
$$=\sum_{1 \le i \le k}(A^\top uu^\top A)_{ii}\mathbb{E}\big[\sigma(w_i^\top x)^2(x \otimes x)\big] + 2\sum_{1 \le i < j \le k}(A^\top uu^\top A)_{ij}\mathbb{E}\big[\sigma(w_i^\top x)\sigma(w_j^\top x)(x \otimes x)\big]$$
$$=\frac{1}{2}\sum_{1 \le i \le k}(A^\top uu^\top A)_{ii}\mathbb{E}\big[(w_i^\top x)^2(x \otimes x)\big] + 2\sum_{1 \le i < j \le k}(A^\top uu^\top A)_{ij}\mathbb{E}\big[\sigma(w_i^\top x)\sigma(w_j^\top x)(x \otimes x)\big].$$
$$(6)$$

Now, we subtract (5) by (6) to obtain

$$2\mathbb{E}\big[(u^\top y) \cdot (\mathbb{E}[(u^\top y)x^\top]\mathbb{E}[xx^\top]^{-1}x) \cdot (x \otimes x)\big] - \mathbb{E}\big[(u^\top y)^2(x \otimes x)\big]$$

$$= \sum_{1 \leq i < j \leq k} (A^\top u u^\top A)_{ij}\mathbb{E}\big[(w_i^\top x)(w_j^\top x)(x \otimes x)\big] - 2\sum_{1 \leq i < j \leq k} (A^\top u u^\top A)_{ij}\mathbb{E}\big[\sigma(w_i^\top x)\sigma(w_j^\top x)(x \otimes x)\big]$$

$$= \sum_{1 \leq i < j \leq k} (A^\top u u^\top A)_{ij}\mathbb{E}\big[[(w_i^\top x)(w_j^\top x)(x \otimes x)\mathbb{1}\{w_i^\top x w_j^\top x \leq 0\}\big] \tag{7}$$

$$= \sum_{1 \leq i < j \leq k} (A^\top u u^\top A)_{ij} N_{ij}, \tag{8}$$

where (7) uses (9) of the following Lemma 7, and (8) uses the definition of distinguishing matrix $N$ (Definition 1). $\qquad\square$

**Lemma 7.** *Given input $x$ coming from a symmetric distribution, for any vector $a, b \in \mathbb{R}^d$, we have*

$$\frac{1}{2}\mathbb{E}\big[(a^\top x)(b^\top x)\big] - \mathbb{E}\big[\sigma(a^\top x)\sigma(b^\top x)\big] = \frac{1}{2}\mathbb{E}\big[(a^\top x)(b^\top x)\mathbb{1}\{a^\top x b^\top x \leq 0\}\big]$$

*and*

$$\frac{1}{2}\mathbb{E}\big[(a^\top x)(b^\top x)(x \otimes x)\big] - \mathbb{E}\big[\sigma(a^\top x)\sigma(b^\top x)(x \otimes x)\big] = \frac{1}{2}\mathbb{E}\big[(a^\top x)(b^\top x)(x \otimes x)\mathbb{1}\{a^\top x b^\top x \leq 0\}\big], \tag{9}$$

*where the expectation is taken over the input distribution.*

*Proof.* Here we just prove the first identity, because the proof of the second one is almost identical. First, we rewrite $\frac{1}{2}\mathbb{E}\big[(a^\top x)(b^\top x)\big]$ as follows

$$\frac{1}{2}\mathbb{E}\big[(a^\top x)(b^\top x)\big] = \frac{1}{2}\mathbb{E}\big[(a^\top x)(b^\top x)\mathbb{1}\{a^\top x b^\top x \leq 0\}\big] + \frac{1}{2}\mathbb{E}\big[(a^\top x)(b^\top x)\mathbb{1}\{a^\top x b^\top x > 0\}\big].$$

Thus, we only need to show that $\frac{1}{2}\mathbb{E}\big[(a^\top x)(b^\top x)\mathbb{1}\{a^\top x b^\top x > 0\}\big] = \mathbb{E}\big[\sigma(a^\top x)\sigma(b^\top x)\big]$. It's clear that

$$\mathbb{E}\big[\sigma(a^\top x)\sigma(b^\top x)\big] = \mathbb{E}\big[\sigma(a^\top x)\sigma(b^\top x)\mathbb{1}\{a^\top x b^\top x > 0\}\big].$$

Since input $x$ comes from a symmetric distribution, we have

$$\mathbb{E}\big[\sigma(a^\top x)\sigma(b^\top x)\mathbb{1}\{a^\top x b^\top x > 0\}\big] = \mathbb{E}\big[\sigma(-a^\top x)\sigma(-b^\top x)\mathbb{1}\{a^\top x b^\top x > 0\}\big]$$

$$= \frac{1}{2}\Big(\mathbb{E}\big[\sigma(a^\top x)\sigma(b^\top x)\mathbb{1}\{a^\top x b^\top x > 0\}\big]$$

$$+ \mathbb{E}\big[\sigma(-a^\top x)\sigma(-b^\top x)\mathbb{1}\{a^\top x b^\top x > 0\}\big]\Big)$$

$$= \frac{1}{2}\mathbb{E}\Big[\big(\sigma(a^\top x)\sigma(b^\top x) + \sigma(-a^\top x)\sigma(-b^\top x)\big)\mathbb{1}\{a^\top x b^\top x > 0\}\Big]$$

$$= \frac{1}{2}\mathbb{E}\big[(a^\top x)(b^\top x)\mathbb{1}\{a^\top x b^\top x > 0\}\big].$$

When $a^\top x b^\top x > 0$, we know $a^\top x$ and $b^\top x$ are both positive or both negative. In either case, we know that $\sigma(a^\top x)\sigma(b^\top x) + \sigma(-a^\top x)\sigma(-b^\top x) = (a^\top x)(b^\top x)$. Thus, we have

$$\mathbb{E}\big[\sigma(a^\top x)\sigma(b^\top x)\big] = \mathbb{E}\big[\sigma(a^\top x)\sigma(b^\top x)\mathbb{1}\{a^\top x b^\top x > 0\}\big]$$

$$= \frac{1}{2}\mathbb{E}\big[(a^\top x)(b^\top x)\mathbb{1}\{a^\top x b^\top x > 0\}\big],$$

which finished our proof. $\qquad\square$

**Finding span:** Now, we find the span of $\{\text{vec}^*(z_i z_i^\top)\}$. First, we recall that $f(u) = 2\mathbb{E}\big[(u^\top y) \cdot (\mathbb{E}[(u^\top y)x^\top]\mathbb{E}[xx^\top]^{-1}x) \cdot (x \otimes x)\big] - \mathbb{E}\big[(u^\top y)^2(x \otimes x)\big]$. Then, according to (8), we have

$$f(u) = 2\mathbb{E}\big[(u^\top y) \cdot (\mathbb{E}[(u^\top y)x^\top]\mathbb{E}[xx^\top]^{-1}x) \cdot (x \otimes x)\big] - \mathbb{E}\big[(u^\top y)^2(x \otimes x)\big] = \sum_{1 \leq i < j \leq k} (A^\top u u^\top A)_{ij} N_{ij}.$$

It is not hard to verify that $u^\top y$ is a pure neuron if and only if $f(u) = 0$. Note that $f(u) = 0$ is a system of quadratic equations. So we linearize it by increasing the dimension (i.e., consider $u_i u_j$ as a single variable) similar to the FOOBI algorithm. Thus the number of variable is $\binom{k}{2} + k = k_2 + k$, i.e.,

$$\exists T \in \mathbb{R}^{d^2 \times (k_2 + k)} \text{ such that } T\text{vec}^*(U) = f(U) = \sum_{1 \le i < j \le k} (A^\top U A)_{ij} N_{ij}. \qquad (10)$$

Now, we prove the Lemma 4 which shows the null space of $T$ is exactly the span of $\{\text{vec}^*(z_i z_i^\top)\}$.

**Proof of Lemma 4.** We divide the proof to the following two cases:

1. For any vector $\text{vec}^*(U)$ belongs to the null space of $T$, we have $T\text{vec}^*(U) = 0$. Note that the RHS of (10) equals to 0 if and only if $A^\top U A$ is a diagonal matrix since the distinguishing matrix $N$ is full column rank and $A^\top U A$ is symmetric. Thus $\text{vec}^*(U)$ belongs to the span of $\{\text{vec}^*(z_i z_i^\top)\}$ since $U = Z^\top D Z$ for some diagonal matrix $D$.

2. For any vector $\text{vec}^*(U)$ belonging to the span of $\{\text{vec}^*(z_i z_i^\top)\}$, $U$ is a linear combination of $z_i z_i^\top$'s. Furthermore, $T\text{vec}^*(U)$ is a linear combination of $T\text{vec}^*(z_i z_i^\top)$. Note that $A^\top z_i$ only has one non-zero entry due to the definition of $z_i$, for any $i \in [k]$. Thus all coefficients in the RHS of (10) are 0. We get $T\text{vec}^*(U) = 0$.

$\square$

**Finding $z_i$'s:** Now, we prove the final Lemma 5 which finds all $z_i$'s from the span of $\{\text{vec}^*(z_i z_i^\top)\}$ by using simultaneous diagonalization. Given all $z_i$'s, this two-layer network can be reduced to a single-layer one. Then one can use Algorithm 1 to recover the first layer parameters $w_i$'s.

**Proof of Lemma 5.** As we discussed before this lemma, we have $XY^{-1} = Z^\top D_X D_Y^{-1} (Z^\top)^{-1}$. According to the following Lemma 8 (i.e., all diagonal elements of $D_x D_y^{-1}$ are non-zero and distinct), the matrix $XY^{-1}$ have $k$ eigenvectors and there are $k$ $z_i$'s, therefore $z_i$'s are the only eigenvectors of $XY^{-1}$.
$\square$

**Lemma 8.** *With probability $1$, all diagonal elements of $D_X$ and $D_Y$ are non-zero and all diagonal elements of $D_X D_Y^{-1}$ are distinct, where $X = Z^\top D_X Z$ and $Y = Z^\top D_Y Z$ are defined in Line 7 of Algorithm 2.*

*Proof.* First, we know there exist diagonal matrices $\{D_i : i \in [k]\}$ such that $\text{mat}^*(\text{vec}^*(U_i)) = Z^\top D_i Z$ for all $i \in [k]$, where $\{\text{vec}^*(U_i) : i \in [k]\}$ are the $k$-least right singular vectors of $T$ (see Line 5 of Algorithm 2). Then, let the vector $d_i \in \mathbb{R}^k$ be the diagonal elements of $D_i$, for all $i \in k$. Let matrix $Q \in \mathbb{R}^{k \times k}$ be a matrix where its $i$-th column is $d_i$. Then $D_X = \text{diag}(Q\zeta_1)$ and $D_Y = \text{diag}(Q\zeta_2)$, where $\zeta_1$ and $\zeta_2$ are two random $k$-dimensional standard Gaussian vectors (see Line 7 of Algorithm 2).

Since $\{\text{vec}^*(U_i) : i \in [k]\}$ are singular vectors of $T$, we know $d_1, d_2, \cdots, d_k$ are independent. Thus, $Q$ has full rank and none of its rows are zero vectors. Let $i$-th row of $Q$ be $q_i^\top$. Let's consider $D_X$ first. In order for $i$-th diagonal element of $D_X$ to be zero, we need $q_i^\top \zeta_1 = 0$. Since $q_i$ is not a zero vector, we know the solution space of $q_i \zeta_1 = 0$ is a lower-dimension manifold in $\mathbb{R}^k$, which has zero measure. Since finite union of zero-measure sets still has measure zero, the event that zero valued elements exist in the diagonal of $D_X$ or $D_Y$ happens with probability zero.

If $i$-th and $j$-th diagonal elements of $D_X D_Y^{-1}$ have same value, we have $q_i^\top \zeta_1 (q_i^\top \zeta_2)^{-1} = q_j^\top \zeta_1 (q_j^\top \zeta_2)^{-1}$. Again, we know the solution space is a lower-dimensional manifold in $\mathbb{R}^{2k}$ space, with measure zero. Since finite union of zero-measure sets still has measure zero, the event that duplicated diagonal elements exist in $D_X D_Y^{-1}$ happens with probability zero. $\square$

## A.2 NOISY CASE

Now, we discuss how to handle the noisy case (i.e. $y = \sigma(Wx) + \xi$). The corresponding algorithm is described in Algorithm 3. Note that the noise $\xi$ only affects the first two steps, i.e., pure neuron detector (Lemma 3) and finding span of $\text{vec}^*(z_i z_i^\top)$ (Lemma 4). It does not affect the last two steps, i.e., finding $z_i$'s from the span (Lemma 5) and learning the reduced single-layer network. Because

---

**Algorithm 3** Learning Two-layer Neural Networks with Noise

---

**Input:** Samples $(x_1, y_1), ..., (x_n, y_n)$ generated according to Equation (2)
**Output:** Weight matrices $W$ and $A$.

1: {*Finding span of* $\text{vec}^*(z_i z_i^\top)$}
2: Use first half of samples (i.e. $\{(x_i, y_i)\}_{i=1}^{n/2}$) to estimate empirical moments $\hat{\mathbb{E}}[xx^\top]$, $\hat{\mathbb{E}}[yx^\top]$, $\hat{\mathbb{E}}[yy^\top]$, $\hat{\mathbb{E}}[y \otimes x^{\otimes 3}]$ and $\hat{\mathbb{E}}[y \otimes y \otimes (x \otimes x)]$.
3: Compute the vector $f(u) = 2\hat{\mathbb{E}}\big[(u^\top y) \cdot (\hat{\mathbb{E}}[(u^\top y)x^\top]\hat{\mathbb{E}}[xx^\top]^{-1}x) \cdot (x \otimes x)\big] - \hat{\mathbb{E}}\big[(u^\top y)^2(x \otimes x)\big] + \big(\hat{\mathbb{E}}[(u^\top y)^2] - 2\hat{\mathbb{E}}[(u^\top y)x^\top]\hat{\mathbb{E}}[xx^\top]^{-1}\hat{\mathbb{E}}[(u^\top y)x]\big)\hat{\mathbb{E}}[x \otimes x]$ where each entry is expressed as a degree-2 polynomial over $u$.
4: Construct matrix $T \in \mathbb{R}^{d^2 \times (k_2 + k)}$ such that, $T\text{vec}^*(uu^\top) = f(u)$.
5: Compute the $k$-least right singular vectors of $T$, denoted by $\text{vec}^*(U_1), \text{vec}^*(U_2), \cdots, \text{vec}^*(U_k)$. Let $S$ be a $k_2 + k$ by $k$ matrix, where the $i$-th column of $S$ is vector $\text{vec}^*(U_i)$. {Here span $S$ is equal to span of $\{\text{vec}^*(z_i z_i^\top)\}$.}
6: {*Finding* $z_i$'s *from span*}
7: Let $X = \text{mat}^*(S\zeta_1)$, $Y = \text{mat}^*(S\zeta_2)$, where $\zeta_1$ and $\zeta_2$ are two independently sampled $k$-dimensional standard Gaussian vectors.
8: Let $z_1, ..., z_k$ be eigenvectors of $XY^{-1}$.
9: For each $z_i$, use the second half of samples $\{(x_i, y_i)\}_{i=n/2+1}^n$ to estimate $\hat{\mathbb{E}}[z_i^\top y]$. If $\hat{\mathbb{E}}[z_i^\top y] < 0$ let $z_i \leftarrow -z_i$. {Here $z_i$'s are normalized rows of $A^{-1}$.}
10: {*Reduce to 1-Layer Problem*}
11: For each $z_i$, let $v_i$ be the output of Algorithm 1 with input $\{(x_j, z_i^\top y_j)\}_{j=n/2+1}^n$.
12: Let $Z$ be the matrix whose rows are $z_i^\top$'s, $V$ be the matrix whose rows are $v_i^\top$'s.
13: **return** $V$, $Z^{-1}$.

---

Lemma 5 is independent of the model and Lemma 1 is linear wrt. noise $\xi$, which has zero mean and is independent of input $x$.

Many of the steps in Algorithm 3 are designed with the robustness of the algorithm in mind. For example, in step 5 for the exact case we just need to compute the null space of $T$. However if we use the empirical moments the null space might be perturbed so that it has small singular values. The separation of the input samples into two halves is also to avoid correlations between the steps, and is not necessary if we have the exact moments.

**Modification for pure neuron detector:** Recall that in the noiseless case, our pure neuron detector contains a term $\mathbb{E}[(u^\top y)^2(x \otimes x)]$, which causes a noise square term in the noisy case. Here, we modify our pure neuron detector to cancel the extra noise square term. In the following lemma, we state our modified pure neuron detector in Equation 11, and give it a characterization.

**Lemma 9.** *Suppose* $y = A\sigma(Wx) + \xi$, *for any* $u \in \mathbb{R}^k$, *we have*

$$f(u) := 2\mathbb{E}\big[(u^\top y) \cdot (\mathbb{E}[(u^\top y)x^\top]\mathbb{E}[xx^\top]^{-1}x) \cdot (x \otimes x)\big] - \mathbb{E}\big[(u^\top y)^2(x \otimes x)\big]$$

$$+ \big(\mathbb{E}[(u^\top y)^2] - 2\mathbb{E}[(u^\top y)x^\top]\mathbb{E}[xx^\top]^{-1}\mathbb{E}[(u^\top y)x]\big)\mathbb{E}[x \otimes x] \quad (11)$$

$$= \sum_{1 \leq i < j \leq k} (A^\top uu^\top A)_{ij} N_{ij} - \Big(\sum_{1 \leq i < j \leq k} (A^\top uu^\top A)_{ij} m_{ij}\Big)\mathbb{E}[x \otimes x], \quad (12)$$

*where* $N_{ij}$'s *are columns of the distinguishing matrix (Definition 1), and* $m_{ij} := \mathbb{E}\big[(w_i^\top x)(w_j^\top x)\mathbb{1}\{w_i^\top x w_j^\top x \leq 0\}\big]$.

We defer the proof of this lemma to the end of this section. Recall that the augmented distinguishing matrix $M$ consists of the distinguishing matrix $N$ plus column $\mathbb{E}[x \otimes x]$. Now, we need to assume the augmented distinguishing matrix $M$ is full rank.

**Modification for finding span:** For Lemma 4, as we discussed above, here we assume the augmented distinguishing matrix $M$ is full rank. The corresponding lemma is stated as follows (the proof is exactly the same as previous Lemma 4):

**Lemma 10.** *Let $T$ be the unique $\mathbb{R}^{d^2 \times (k_2+k)}$ matrix that satisfies $T vec^*(uu^\top) = f(u)$ (defined in Equation 11), suppose the augmented distinguishing matrix is full rank, then the nullspace of $T$ is exactly the span of $\{vec^*(z_i z_i^\top)\}$.*

Similar to Theorem 4, we provide the following theorem for the noisy case. The proof is almost the same as Theorem 4 by using the noisy version lemmas (Lemmas 9 and 10).

**Theorem 5.** *Suppose $\mathbb{E}[xx^\top], A, W$ and the augmented distinguishing matrix $M$ are all full rank, and Algorithm 3 has access to the exact moments, then the network returned by the algorithm computes exactly the same function as the original neural network.*

Now, we only need to prove Lemma 9 to finish this noise case.

**Proof of Lemma 9.** Similar to (5) and (6), we deduce these three terms in RHS of (11) one by one as follows. For the first term, it is exactly the same as (5) since the expectation is linear wrt. $\xi$. Thus, we have

$$
\begin{aligned}
&2\mathbb{E}\big[(u^\top y) \cdot (\mathbb{E}[(u^\top y)x^\top]\mathbb{E}[xx^\top]^{-1}x) \cdot (x \otimes x)\big] \\
=&\frac{1}{2} \sum_{1 \leq i \leq k} (A^\top uu^\top A)_{ii}\mathbb{E}\big[(w_i^\top x)^2(x \otimes x)\big] + \sum_{1 \leq i < j \leq k} (A^\top uu^\top A)_{ij}\mathbb{E}\big[(w_i^\top x)(w_j^\top x)(x \otimes x)\big].
\end{aligned}
\tag{13}
$$

Now, let's look at the second term $\mathbb{E}\big[(u^\top y)^2(x \otimes x)\big]$ which is slightly different from (6) due to the noise $\xi$. Particularly, we add the third term to cancel this extra noise square term later.

$$
\begin{aligned}
&\mathbb{E}\big[(u^\top y)^2(x \otimes x)\big] \\
=&\mathbb{E}\big[(u^\top(A\sigma(Wx) + \xi))^2(x \otimes x)\big] \\
=&\mathbb{E}\big[(u^\top A\sigma(Wx))^2(x \otimes x)\big] + \mathbb{E}\big[(u^\top \xi)^2(x \otimes x)\big] \\
=&\frac{1}{2} \sum_{1 \leq i \leq k} (A^\top uu^\top A)_{ii}\mathbb{E}\big[(w_i^\top x)^2(x \otimes x)\big] + 2 \sum_{1 \leq i < j \leq k} (A^\top uu^\top A)_{ij}\mathbb{E}\big[\sigma(w_i^\top x)\sigma(w_j^\top x)(x \otimes x)\big]
\end{aligned}
\tag{14}
$$

$$
+ \mathbb{E}\big[(u^\top \xi)^2(x \otimes x)\big],
\tag{15}
$$

where (15) uses (6).

For the third term, we have

$$
\begin{aligned}
&\mathbb{E}\big[(u^\top y)^2\big] - 2\mathbb{E}\big[(u^\top y)x^\top\big]\mathbb{E}\big[xx^\top\big]^{-1}\mathbb{E}\big[(u^\top y)x\big] \\
=&\mathbb{E}\big[(u^\top A\sigma(Wx))^2\big] + \mathbb{E}\big[(u^\top \xi)^2\big] - \frac{1}{2}\mathbb{E}\big[\langle A^\top u, Wx \rangle^2\big] \\
=&\sum_{1 \leq i \leq k} (A^\top uu^\top A)_{ii}\mathbb{E}\big[\sigma(w_i^\top x)^2\big] - \frac{1}{2} \sum_{1 \leq i \leq k} (A^\top uu^\top A)_{ii}\mathbb{E}\big[(w_i^\top x)^2\big] \\
&\quad + 2 \sum_{1 \leq i < j \leq k} (A^\top uu^\top A)_{ij}\mathbb{E}\big[\sigma(w_i^\top x)\sigma(w_j^\top x)\big] - \sum_{1 \leq i < j \leq k} (A^\top uu^\top A)_{ij}\mathbb{E}\big[(w_i^\top x)(w_j^\top x)\big] \\
&\quad + \mathbb{E}\big[(u^\top \xi)^2\big] \\
=&- \sum_{1 \leq i < j \leq k} (A^\top uu^\top A)_{ij}\mathbb{E}\big[(w_i^\top x)(w_j^\top x)\mathbb{1}\{w_i^\top x w_j^\top x \leq 0\}\big] + \mathbb{E}\big[(u^\top \xi)^2\big] \\
=&- \sum_{1 \leq i < j \leq k} (A^\top uu^\top A)_{ij}m_{ij} + \mathbb{E}\big[(u^\top \xi)^2\big],
\end{aligned}
\tag{16}
$$

where the third equality holds due to Lemma 6 and Lemma 7, and (16) uses the definition of $m_{ij}$.

---

**Algorithm 4** Learning Two-layer Neural Networks with Non-square $A$

---

**Input:** Samples $(x_1, y_1), ..., (x_n, y_n)$ generated according to Equation (2).
**Output:** Weight matrices $W \in \mathbb{R}^{k \times d}$ and $A \in^{l \times k}$.
 1: Using half samples (i.e. $\{(x_i, y_i)\}_{i=1}^{n/2}$) to estimate empirical moments $\hat{\mathbb{E}}[yx^\top]$.
 2: Let $P$ be a $l \times k$ matrix, which columns are left singular vectors of $\hat{\mathbb{E}}[yx^\top]$.
 3: Run Algorithm 3 on samples $\{(x_i, P^\top y_i)\}_{i=n/2}^n$. Let the output of Algorithm 3 be $V, Z^{-1}$.
 4: **return** $V, PZ^{-1}$.

---

Finally, we combine these three terms (13–16) as follows:

$$
\begin{aligned}
f(u) &= 2\mathbb{E}\big[(u^\top y) \cdot (\mathbb{E}[(u^\top y)x^\top]\mathbb{E}[xx^\top]^{-1}x) \cdot (x \otimes x)\big] - \mathbb{E}\big[(u^\top y)^2(x \otimes x)\big] \\
&\quad + \Big(\mathbb{E}\big[(u^\top y)^2\big] - 2\mathbb{E}\big[(u^\top y)x^\top\big]\mathbb{E}\big[xx^\top\big]^{-1}\mathbb{E}\big[(u^\top y)x\big]\Big)\mathbb{E}[x \otimes x] \\
&= \sum_{1 \le i < j \le k} (A^\top uu^\top A)_{ij}\mathbb{E}\big[(w_i^\top x)(w_j^\top x)(x \otimes x)\big] \\
&\quad - 2\sum_{1 \le i < j \le k} (A^\top uu^\top A)_{ij}\mathbb{E}\big[\sigma(w_i^\top x)\sigma(w_j^\top x)(x \otimes x)\big] - \mathbb{E}\big[(u^\top \xi)^2\big]\mathbb{E}\big[x \otimes x\big] \\
&\quad + \Big(-\sum_{1 \le i < j \le k} (A^\top uu^\top A)_{ij}m_{ij} + \mathbb{E}\big[(u^\top \xi)^2\big]\Big)\mathbb{E}\big[x \otimes x\big] \\
&= \sum_{1 \le i < j \le k} (A^\top uu^\top A)_{ij}N_{ij} - \Big(\sum_{1 \le i < j \le k} (A^\top uu^\top A)_{ij}m_{ij}\Big)\mathbb{E}[x \otimes x], \tag{17}
\end{aligned}
$$

where (17) uses (9) (same as (7)). $\qquad\square$

### A.3 EXTENSION TO NON-SQUARE $A$

In this paper, for simplicity, we have assumed that the dimension of output equals the number of hidden units and thus $A$ is a $k \times k$ square matrix. Actually, our algorithm can be easily extended to the case where the dimension of output is at least the number of hidden units. In this section, we give an algorithm for this general case, by reducing it to the case where $A$ is square. The pseudo-code is given in Algorithm 4.

**Theorem 6.** *Suppose $\mathbb{E}[xx^\top], W, A$ and the augmented distinguishing matrix $M$ are all full rank, and Algorithm 4 has access to the exact moments, then the network returned by the algorithm computes exactly the same function as the original neural network.*

*Proof.* Let the ground truth parameters be $A \in \mathbb{R}^{l \times k}$ and $W \in \mathbb{R}^{k \times d}$. The samples are generated by $y = A\sigma(Wx) + \xi$, where the noise $\xi$ is independent with input $x$. We have $\mathbb{E}[yx^\top] = \frac{1}{2}AW\mathbb{E}[xx^\top]$. Since both $W$ and $\mathbb{E}[xx^\top]$ are full-rank, we know the column span of $\mathbb{E}[yx^\top]$ are exactly the column span of $A$. Furthermore, we know the columns of $P$ is a set of orthonormal basis for the column span of $A$.

For a ground truth neural network with weight matrices $W$ and $P^\top A$, the generated sample will just be $(x, P^\top y)$. According to Theorem 5, we know for any input $x$, we have $Z^{-1}\sigma(Vx) = P^\top A\sigma(Wx)$. Thus, we have

$$
PZ^{-1}\sigma(Vx) = PP^\top A\sigma(Wx) = A\sigma(Wx),
$$

where the second equality holds since $PP^\top$ is just the projection matrix to the column span of $A$. $\qquad\square$

## B ROBUSTNESS OF MAIN ALGORITHM

In this section we will show that even if we do not have access to the exact moments, as long as the empirical moments are estimated with enough (polynomially many) samples, Algorithm 2 and

Algorithm 3 can still learn the parameters robustly. We will focus on Algorithm 3 as it is more general, the result for Algorithm 2 can be viewed as a corollary when the noise $\xi = 0$. Throughout this section, we will use $\hat{V}, \hat{Z}^{-1}$ to denote the results of Algorithm 3 with empirical moments, and use $V, Z^{-1}$ for the results when the algorithm has access to exact moments, similarly for other intermediate results. For the robustness of Algorithm 3, we prove the following theorem.

**Theorem 7.** *Assume that the norms of $x, \xi, A$ are bounded by $\|x\| \leq \Gamma, \|\xi\| \leq P_2, \|A\| \leq P_1$, the covariance matrix and the weight matrix are robustly full rank: $\sigma_{\min}(\mathbb{E}[xx^\top]) \geq \gamma, \sigma_{\min}(A) \geq \beta$. Further assume that the augmented distinguishing matrix has smallest singular values $\sigma_{\min}(M) \geq \alpha$. For any small enough $\epsilon$, for any $\delta < 1$, given $poly\big(\Gamma, P_1, P_2, d, 1/\epsilon, 1/\gamma, 1/\alpha, 1/\beta, 1/\delta\big)$ number of i.i.d. samples, let the output of Algorithm 3 be $\hat{V}, \hat{Z}^{-1}$, we know with probability at least $1 - \delta$,*

$$\|A\sigma(Wx) - \hat{Z}^{-1}\sigma(\hat{V}x)\| \leq \epsilon,$$

*for any input $x$.*

In order to prove the above Theorem, we need to show that each step of Algorithm 3 is robust. We can divide Algorithm 3 into three steps: finding the span of $\text{vec}^*(z_i z_i^\top)$'s; finding $z_i$'s from the span of $\text{vec}^*(z_i z_i^\top)$'s; recovering first layer using Algorithm 1. We will first state the key lemmas that prove every step is robust to noise, and finally combine them to show our main theorem.

First, we show that with polynomial number of samples, we can approximate the span of $\text{vec}^*(z_i z_i^\top)$'s in arbitrary accuracy. Let $\hat{T}$ be the empirical estimate of $T$, which is the pure neuron detector matrix as defined in Algorithm 3. As shown in Lemma 10, the null space of $T$ is exactly the span of $\text{vec}^*(z_i z_i^\top)$'s. We use standard matrix perturbation theory (see Section D.2) to show that the null space of $T$ is robust to small perturbations. More precisely, in Lemma 11, we show that with polynomial number of samples, the span of $k$ least singular vectors of $\hat{T}$ is close to the null space of $T$.

**Lemma 11.** *Under the same assumptions as in Theorem 7, let $S \in \mathbb{R}^{(k_2+k)\times k}$ be the matrix whose $k$ columns are the $k$ least right singular vectors of $T$. Similarly define $\hat{S} \in \mathbb{R}^{(k_2+k)\times k}$ for empirical estimate $\hat{T}$. Then for any $\epsilon \leq \gamma/2$, for any $\delta < 1$, given $O\big(\frac{d^3\Gamma^{14}(\Gamma P_1\sqrt{k}+P_2)^6\log(\frac{d}{\delta})}{\gamma^4\epsilon^2}\big)$ number of i.i.d. samples, we know with probability at least $1 - \delta$,*

$$\|SS^\top - \hat{S}\hat{S}^\top\| \leq \frac{\sqrt{2}\epsilon}{\alpha\beta^2}.$$

The proof of the above lemma is in Section B.1. Basically, we need to lowerbound the spectral gap ($k_2$-th singular value of $T$) and to upperbound the Frobenius norm of $T - \hat{T}$. Standard matrix perturbation bound shows that if the perturbation is much smaller than the spectral gap, then the null space is preserved.

Next, we show that we can robustly find $z_i$'s from the span of $\text{vec}^*(z_i z_i^\top)$'s. Since this step of the algorithm is the same as the simultaneous diagonalization algorithm for tensor decompositions, we use the robustness of simultaneous diagonalization (Bhaskara et al., 2014) to show that we can find $z_i$'s robustly. The detailed proof is in Section B.2.

**Lemma 12.** *Suppose that $\|SS^\top - \hat{S}\hat{S}^\top\|_F \leq \epsilon, \|A\| \leq P_1, \|\xi\| \leq P_2, \sigma_{\min}(\mathbb{E}[xx^\top]) \geq \gamma, \sigma_{\min}(A) \geq \beta$. Let $\hat{X} = mat^*(\hat{S}\zeta_1), \hat{Y} = mat^*(\hat{S}\zeta_2)$, where $\zeta_1$ and $\zeta_2$ are two independent standard Gaussian vectors. Let $z_1, \cdots z_k$ be the normalized row vectors of $A^{-1}$. Let $\hat{z}_1, ..., \hat{z}_k$ be the eigenvectors of $\hat{X}\hat{Y}^{-1}$ (after sign flip). For any $\delta > 0$ and small enough $\epsilon$, with $O\big(\frac{(\Gamma P_1\sqrt{k}+P_2)^2\log(d/\delta)}{\epsilon^2}\big)$ number of i.i.d. samples in $\hat{\mathbb{E}}[\hat{z}_i^\top y]$, with probability at least $1 - \delta$ over the randomness of $\zeta_1, \zeta_2$ and i.i.d. samples, there exists a permutation $\pi(i) \in [k]$ such that*

$$\|\hat{z}_i - z_{\pi[i]}\| \leq poly(d, P_1, 1/\beta, \epsilon, 1/\delta),$$

*for any $1 \leq i \leq k$.*

Finally, given $\hat{z}_i$'s, the problem reduces to a one-layer problem. We will first give an analysis for Algorithm 1 as a warm-up. When we call Algorithm 1 from Algorithm 3, the situation is slightly different. Note we reserve fresh samples for this step, so that the samples used by Algorithm 1 are

still independent with the estimate $\hat{z}_i$ (learned using the other set of samples). However, since $\hat{z}_i$ is not equal to $z_i$, this introduces an additional error term $(\hat{z}_i - z_i)^\top y$ which is not independent of $x$ and cannot be captured by $\xi$. We modify the proof for Algorithm 1 to show that the algorithm is still robust as long as $\|\hat{z}_i - z_i\|$ is small enough.

**Lemma 13.** *Assume that $\|x\| \leq \Gamma, \|A\| \leq P_1, \|\xi\| \leq P_2$ and $\sigma_{\min}(\mathbb{E}[xx^\top]) \geq \gamma$. Suppose that for each $1 \leq i \leq k$, $\|\hat{z}_i - z_i\| \leq \tau$. Then for any $\epsilon \leq \gamma/2$ and $\delta < 1$, given $O(\frac{(\Gamma^2 + P_2\Gamma)^4 \log(\frac{d}{\delta})}{\gamma^4 \epsilon^2})$ number of samples for Algorithm 1, we know with probability at least $1 - \delta$,*

$$\|v_i - \hat{v}_i\| \leq \frac{2\tau(\Gamma P_1 \sqrt{k} + P_2)\Gamma}{\gamma} + 2\epsilon,$$

*for each $1 \leq i \leq k$.*

Combining the above three lemmas, we prove Theorem 7 in Section B.4.

### B.1    ROBUST ANALYSIS FOR FINDING THE SPAN OF $\{\text{VEC}^*(z_i z_j^\top)\}$'S

We first prove that the step of finding the span of $\{\text{vec}^*(z_i z_j^\top)\}$ is robust. The main idea is based on standard matrix perturbation bounds (see Section D.2). We first give a lowerbound on the $k_2$-th singular value of $T$, giving a spectral gap between the smallest non-zero singular value and the null space. See the lemma below. The proof is given in Section B.1.1.

**Lemma 14.** *Suppose $\sigma_{\min}(M) \geq \alpha, \sigma_{\min}(A) \geq \beta$, we know that matrix $T$ has rank $k_2$ and the $k_2$-th singular value of $T$ is lower bounded by $\alpha\beta^2$.*

Then we show that with enough samples the estimate $\hat{T}$ is close enough to $T$, so Wedin's Theorem (Lemma 25) implies the subspace found is also close to the true nullspace of $T$. The proof is deferred to Section B.1.2.

**Lemma 15.** *Assume that $\|x\| \leq \Gamma, \|A\| \leq P_1, \|\xi\| \leq P_2$ and $\sigma_{\min}(\mathbb{E}[xx^\top]) \geq \gamma > 0$, then for any $\epsilon \leq \gamma/2$, for any $1 > \delta > 0$, given $O(\frac{d^3 \Gamma^{14}(\Gamma P_1 \sqrt{k} + P_2)^6 \log(\frac{d}{\delta})}{\gamma^4 \epsilon^2})$ number of i.i.d. samples, we know*

$$\|\hat{T} - T\|_F \leq \epsilon,$$

*with probability at least $1 - \delta$.*

Finally we combine the above two lemmas and show that the span of the least $k$ right singular vectors of $\hat{T}$ is close to the null space of $T$.

**Lemma 11.** *Under the same assumptions as in Theorem 7, let $S \in \mathbb{R}^{(k_2+k)\times k}$ be the matrix whose $k$ columns are the $k$ least right singular vectors of $T$. Similarly define $\hat{S} \in \mathbb{R}^{(k_2+k)\times k}$ for empirical estimate $\hat{T}$. Then for any $\epsilon \leq \gamma/2$, for any $\delta < 1$, given $O(\frac{d^3 \Gamma^{14}(\Gamma P_1 \sqrt{k} + P_2)^6 \log(\frac{d}{\delta})}{\gamma^4 \epsilon^2})$ number of i.i.d. samples, we know with probability at least $1 - \delta$,*

$$\|SS^\top - \hat{S}\hat{S}^\top\| \leq \frac{\sqrt{2}\epsilon}{\alpha\beta^2}.$$

*Proof.* According to Lemma 15, given $O(\frac{d^3 \Gamma^{14}(\Gamma P_1 \sqrt{k} + P_2)^6 \log(\frac{d}{\delta})}{\gamma^4 \epsilon^2})$ number of i.i.d. samples, we know with probability at least $1 - \delta$,

$$\|\hat{T} - T\|_F \leq \epsilon.$$

According to Lemma 14, we know $\sigma_{k_2}(T) \geq \alpha\beta^2$. Then, due to Lemma 27, we have

$$\begin{aligned}
\|SS^\top - \hat{S}\hat{S}^\top\| \leq & \frac{\sqrt{2}\|T - \hat{T}\|_F}{\sigma_{k_2}(T)} \\
\leq & \frac{\sqrt{2}\epsilon}{\alpha\beta^2}.
\end{aligned}$$

$\square$

$$T = MBCF \in \mathbb{R}^{d^2 \times (k_2 + k)}$$

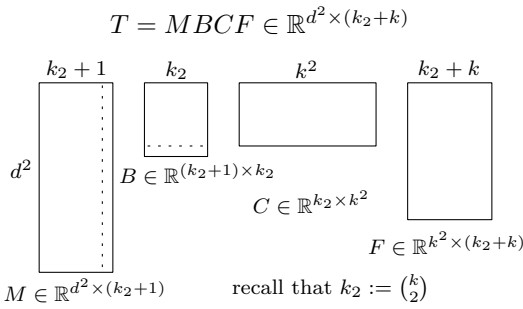

Figure 5: Characterize T as the product of four matrices.

### B.1.1 LOWERBOUNDING $k_2$-TH SINGULAR VALUE OF $T$

In order to lowerbound the $k_2$-th singular value of $T$, we first express $T$ as the product of four simpler matrices, $T = MBCF$, as illustrated in Figure 5. The definitions of these four matrices $M \in \mathbb{R}^{d^2 \times (k_2+1)}, B \in \mathbb{R}^{(k_2+1) \times k_2}, C \in \mathbb{R}^{(k_2 \times k^2)}, F \in \mathbb{R}^{k^2 \times (k_2+k)}$ will be introduced later as we explain their effects. From Lemma 9, we know that

$$T\text{vec}^*(U) = \sum_{1 \le i < j \le k} (A^\top U A)_{ij} M_{ij} - \Big( \sum_{1 \le i < j \le k} (A^\top U A)_{ij} m_{ij} \Big) \mathbb{E}[x \otimes x],$$

for any symmetric $k \times k$ matrix $U$.

Note that $\text{vec}^*(U)$ is a $(k_2 + k)$-dimensional vector. For convenience, we first use matrix $F$ to transform $\text{vec}^*(U)$ to $\text{vec}(U)$, which has $k^2$ dimensions. Matrix $F$ is defined such that $F\text{vec}^*(U) = \text{vec}(U)$, for any $k \times k$ symmetric matrix $U$. Note that this is very easy as we just need to duplicate all the non-diagonal entries.

Second, we hope to get the coefficients $(A^\top U A)_{ij}$'s. Notice that

$$\text{vec}(A^\top U A) = A^\top \otimes A^\top \text{vec}(U) = A^\top \otimes A^\top F\text{vec}^*(U).$$

Since we only care about the elements of $A^\top U A$ at the $ij$-th position for $1 \le i < j \le k$, we just pick corresponding rows of $A^\top \otimes A^\top$ to construct our matrix $C$, which has dimension $k_2 \times k^2$.

The first matrix $M$ is the augmented distinguishing matrix (see Definition 1). In order to better understand the reason that we need matrix $B$, let's first re-write $T\text{vec}^*(U)$ in the following way:

$$T\text{vec}^*(U) = \sum_{1 \le i < j \le k} (A^\top U A)_{ij} \Big( M_{ij} - m_{ij} \mathbb{E}[x \otimes x] \Big).$$

Thus, $T\text{vec}^*(U)$ is just a linear combination of $(M_{ij} - m_{ij}\mathbb{E}[x \otimes x])$'s with coefficients equal to $(A^\top U A)_{ij}$'s. We have already expressed coefficients $(A^\top U A)_{ij}$'s using $CF\text{vec}^*(U)$. Now, we just need to use matrix $B$ to transform the augmented distinguishing matrix $M$ to a $d^2 \times k_2$ matrix, with each column equal to $(M_{ij} - m_{ij}\mathbb{E}[x \otimes x])$. In order to achieve this, the first $k_2$ rows of $B$ is just the identity matrix $I_{k_2}$, and the last row of $B$ is $[-m_{12}, -m_{13}, \cdots, -m_{1k}, -m_{23}, -m_{24}, \cdots]^\top$.

With above characterization of $T$, we are ready to show that the $k_2$-th singular value of $T$ is lower bounded.

**Lemma 14.** *Suppose $\sigma_{\min}(M) \ge \alpha, \sigma_{\min}(A) \ge \beta$, we know that matrix $T$ has rank $k_2$ and the $k_2$-th singular value of $T$ is lower bounded by $\alpha\beta^2$.*

*Proof.* Since matrix $C$ has dimension $k_2 \times k^2$, it's clear that the rank of $T$ is at most $k_2$. We first prove that the rank of $T$ is exactly $k_2$.

Since the first $k_2$ rows of $B$ constitute the identity matrix $I_{k_2}$, we know $B$ is a full-column rank matrix with rank equal to $k_2$. We also know that matrix $M$ is a full column rank matrix with rank $k_2 + 1$. Thus, the product matrix $MB$ is still a full-column rank matrix with rank $k_2$. If we can prove that the product matrix $CF$ has full-row rank equal to $k_2$. It's clear that $T = MBCF$ also has rank $k_2$. Next, we prove that $CF$ has full-row rank.

Since $\sigma_{\min}(A) \geq \beta$, we know $A^\top \otimes A^\top$ is full rank, and a subset of its rows $C$ has full row rank. For the sake of contradiction, suppose that there exists non-zero vector $a \in \mathbb{R}^{k_2}$, such that $\sum_{l=1}^{k_2} a_l (CF)_{[l,:]} = 0$. Note that for any $1 \leq l \leq k_2$, $(CF)_{[l,ii]} = C_{[l,ii]}$ for $1 \leq i \leq k$ and $(CF)_{[l,ij]} = C_{[l,ij]} + C_{[l,ji]}$ for $1 \leq i < j \leq k$. Since $C$ consists of a subset of rows of $A^\top \otimes A^\top$, we know $C_{[l,ij]} = C_{[l,ji]}$ for any $l$ and any $i < j$. Thus, $\sum_{l=1}^{k_2} a_l (CF)_{[l,:]} = 0$ simply implies $\sum_{l=1}^{k_2} a_l C_{[l,:]} = 0$, which breaks the fact that $C$ is full-row rank. Thus, the assumption is false and $CF$ has full-row rank.

Now, let's prove that the $k_2$-th singular value of $T$ is lower bounded. We first show that in the product characterization of $T$, the smallest singular value of each individual matrix is lower bounded. According to the assumption, we know the smallest singular value of $M$ is lower bounded by $\alpha$. Since the first $k_2$ rows of matrix $B$ constitute a $k_2 \times k_2$ identity matrix, we know

$$\sigma_{\min}(B) := \min_{u:\|u\| \leq 1} \|Bu\| \geq \min_{u:\|u\| \leq 1} \|I_{k_2 \times k_2} u\| =: \sigma_{\min}(I_{k_2 \times k_2}) = 1,$$

where $u$ is any $k_2$-dimensional vector.

Since $\sigma_{\min}(A) \geq \beta$, we know $\sigma_{\min}(A^\top \otimes A^\top) \geq \beta^2$. According to the construction of $C$, we know $C$ consists a subset of rows of $A^\top \otimes A^\top$. Denote the indices of the row not picked as $S$. We have

$$\begin{aligned}
\sigma_{\min}(C) &:= \min_{u:\|u\| \leq 1} \|u^\top C\| \\
&= \min_{v:\|v\| \leq 1 \,\wedge\, (v_i = 0, \forall i \in S)} \|v^\top (A^\top \otimes A^\top)\| \\
&\geq \min_{v:\|v\| \leq 1} \|v^\top (A^\top \otimes A^\top)\| \\
&=: \sigma_{\min}(A^\top \otimes A^\top) \\
&\geq \beta^2,
\end{aligned}$$

where $u$ has dimension $k_2$ and $v$ has dimension $k^2$.

We lowerbound the smallest singular value of $CF$ by showing that $\sigma_{\min}(CF) \geq \sigma_{\min}(C)$. For any unit vector $u \in \mathbb{R}^{k_2}$, we know $[u^\top CF]_{ii} = [u^\top C]_{ii}$ for any $i$ and $[u^\top CF]_{ij} = [u^\top C]_{ij} + [u^\top C]_{ji}$ for any $i < j$. We also know $[u^\top C]_{ij} = [u^\top C]_{ji}$ for $i < j$. Thus, we know for any unit vector $u$, $\|u^\top CF\| \geq \|u^\top C\|$, which implies $\sigma_{\min}(CF) \geq \sigma_{\min}(C)$.

Finally, since in the beginning we have proved that matrix $T$ has rank $k_2$, the $k_2$-th singular value is exactly the smallest non-zero singular value of $T$. Denote the smallest non-zero singular of $T$ as $\sigma_{\min}^+(T)$, we have

$$\begin{aligned}
\sigma_{k_2}(T) &= \sigma_{\min}^+(T) \\
&\geq \sigma_{\min}(M)\sigma_{\min}(B)\sigma_{\min}(CF) \\
&\geq \alpha\beta^2,
\end{aligned}$$

where the first inequality holds because both $M$ and $B$ has full column rank. $\qquad\square$

### B.1.2 Upperbounding $\|\hat{T} - T\|_F$

In this section, we prove that given polynomial number of samples, $\|\hat{T} - T\|_F$ is small with high probability. We do this by standard matrix concentration inequalities. Note that our requirements on the norm of $x$ is just for convenience, and the same proof works as long as $x$ has reasonable tail-behavior (e.g. sub-Gaussian).

**Lemma 15.** *Assume that $\|x\| \leq \Gamma, \|A\| \leq P_1, \|\xi\| \leq P_2$ and $\sigma_{\min}(\mathbb{E}[xx^\top]) \geq \gamma > 0$, then for any $\epsilon \leq \gamma/2$, for any $1 > \delta > 0$, given $O(\frac{d^3 \Gamma^{14}(\Gamma P_1 \sqrt{k} + P_2)^6 \log(\frac{d}{\delta})}{\gamma^4 \epsilon^2})$ number of i.i.d. samples, we know*

$$\|\hat{T} - T\|_F \leq \epsilon,$$

*with probability at least $1 - \delta$.*

*Proof.* In order to get an upper bound for $\|\hat{T} - T\|_F$, we first show that $\|\hat{T} - T\|_2$ is upper bounded. We know

$$\|T - \hat{T}\| = \max_{v \in \mathbb{R}^{k^2+k}: \|v\| \leq 1} \|(T - \hat{T})v\|$$

$$\leq \max_{U \in \mathbb{R}^{k \times k}_{sym}: \|U\|_F \leq \sqrt{2}} \|(T - \hat{T})\text{vec}^*(U)\|.$$

For any $k \times k$ symmetric matrix $U$ with eigenvalue decomposition $U = \sum_{i=1}^{k} \lambda_i u^{(i)}(u^{(i)})^\top$, according to the definition of $T$, we know

$$\max_{U \in \mathbb{R}^{k \times k}_{sym}: \|U\|_F \leq \sqrt{2}} \|(T - \hat{T})\text{vec}^*(U)\| = \max_{U \in \mathbb{R}^{k \times k}_{sym}: \|U\|_F \leq \sqrt{2}} \|T\text{vec}^*(U) - \hat{T}\text{vec}^*(U)\|$$

$$\leq \max_{u^{(i)}: \|u^{(i)}\| \leq \sqrt[4]{2}} \|\sum_{i=1}^{k} \lambda_i (f(u^{(i)}) - \hat{f}(u^{(i)}))\|$$

$$\leq \max_{u^{(i)}: \|u^{(i)}\| \leq \sqrt[4]{2}} \sum_{i=1}^{k} |\lambda_i| \|f(u^{(i)}) - \hat{f}(u^{(i)})\|$$

$$\leq \Big( \sum_{i=1}^{k} |\lambda_i| \Big) \max_{u: \|u\| \leq \sqrt[4]{2}} \|f(u) - \hat{f}(u)\|$$

$$\leq \sqrt{k} \sqrt{\sum_{i=1}^{k} \lambda_i^2} \max_{u: \|u\| \leq \sqrt[4]{2}} \|f(u) - \hat{f}(u)\|$$

$$= \sqrt{k} \|U\|_F \max_{u: \|u\| \leq \sqrt[4]{2}} \|f(u) - \hat{f}(u)\|$$

$$\leq \sqrt{2k} \max_{u: \|u\| \leq 2} \|f(u) - \hat{f}(u)\|$$

where $\hat{f}(u) = \hat{T}\text{vec}^*(uu^\top)$ and the fourth inequality uses the Cauchy-Schwarz inequality. Next, we only need to upper bound $\max_{u: \|u\| \leq 2} \|f(u) - \hat{f}(u)\|$. Recall that

$$f(u) = 2\mathbb{E}\big[(u^\top y) \cdot (\mathbb{E}[(u^\top y)x^\top]\mathbb{E}[xx^\top]^{-1}x) \cdot (x \otimes x)\big] - \mathbb{E}\big[(u^\top y)^2(x \otimes x)\big]$$
$$+ \Big( \mathbb{E}\big[(u^\top y)^2\big] - 2\mathbb{E}\big[(u^\top y)x^\top\big]\mathbb{E}\big[xx^\top\big]^{-1}\mathbb{E}\big[(u^\top y)x\big] \Big)\mathbb{E}[x \otimes x],$$

and

$$\hat{f}(u) = 2\hat{\mathbb{E}}\big[(u^\top y) \cdot (\hat{\mathbb{E}}[(u^\top y)x^\top]\hat{\mathbb{E}}[xx^\top]^{-1}x) \cdot (x \otimes x)\big] - \hat{\mathbb{E}}\big[(u^\top y)^2(x \otimes x)\big]$$
$$+ \Big( \hat{\mathbb{E}}\big[(u^\top y)^2\big] - 2\hat{\mathbb{E}}\big[(u^\top y)x^\top\big]\hat{\mathbb{E}}\big[xx^\top\big]^{-1}\hat{\mathbb{E}}\big[(u^\top y)x\big] \Big)\hat{\mathbb{E}}[x \otimes x].$$

Notice that

$$\Big(\mathbb{E}\big[(u^\top y) \cdot (\mathbb{E}[(u^\top y)x^\top]\mathbb{E}[xx^\top]^{-1}x) \cdot (x \otimes x)\big]\Big)^\top = \mathbb{E}\big[(u^\top y)x^\top\big]\mathbb{E}\big[xx^\top\big]^{-1}\mathbb{E}\big[(u^\top y)x(x \otimes x)^\top\big].$$

We first show that given polynomial number of samples,

$$\Big\| 2\hat{\mathbb{E}}\big[(u^\top y)x^\top\big]\hat{\mathbb{E}}\big[xx^\top\big]^{-1}\hat{\mathbb{E}}\big[(u^\top y)x(x \otimes x)^\top\big] - 2\mathbb{E}\big[(u^\top y)x^\top\big]\mathbb{E}\big[xx^\top\big]^{-1}\mathbb{E}\big[(u^\top y)x(x \otimes x)^\top\big] \Big\|$$

is upper bounded with high probability.

Since each row of $W$ has unit norm, we have $\|W\| \leq \sqrt{k}$. Due to the assumption that $\|x\| \leq \Gamma, \|A\| \leq P_1, \|\xi\| \leq P_2$, we have

$$\|(u^\top y)x^\top\| \leq \|u\|\|A\sigma(Wx) + \xi\|\|x\|$$
$$\leq \|u\|(\|A\|\|W\|\|x\| + \|\xi\|)\|x\|$$
$$\leq 2\Gamma(\Gamma P_1 \sqrt{k} + P_2).$$

According to Lemma 24, we know given $O(\frac{\Gamma^2(\Gamma P_1\sqrt{k}+P_2)^2 \log(\frac{d}{\delta})}{\epsilon^2})$ number of samples,

$$\left\|\hat{\mathbb{E}}\big[(u^\top y)x^\top\big] - \mathbb{E}\big[(u^\top y)x^\top\big]\right\| \le \epsilon,$$

with probability at least $1 - \delta$.

Similarly, we can show that given $O(\frac{\Gamma^6(\Gamma P_1\sqrt{k}+P_2)^2 \log(\frac{d}{\delta})}{\epsilon^2})$ number of samples,

$$\left\|\hat{\mathbb{E}}\big[(u^\top y)x(x \otimes x)^\top\big] - \mathbb{E}\big[(u^\top y)x(x \otimes x)^\top\big]\right\| \le \epsilon,$$

with probability at least $1 - \delta$.

Since $\|xx^\top\| \le \Gamma^2$, we know that given $O(\frac{\Gamma^4 \log(\frac{d}{\delta})}{\epsilon^2})$ number of samples,

$$\left\|\hat{\mathbb{E}}\big[xx^\top\big] - \mathbb{E}\big[xx^\top\big]\right\| \le \epsilon,$$

with probability at least $1 - \delta$. Suppose that $\epsilon \le \gamma/2 \le \sigma_{\min}(\mathbb{E}[xx^\top])/2$, we know $\hat{\mathbb{E}}[xx^\top]$ has full rank. According to Lemma 29, we have

$$\left\|\hat{\mathbb{E}}\big[xx^\top\big]^{-1} - \mathbb{E}\big[xx^\top\big]^{-1}\right\| \le 2\sqrt{2}\frac{\|\hat{\mathbb{E}}\big[xx^\top\big] - \mathbb{E}\big[xx^\top\big]\|}{\sigma_{\min}^2(\mathbb{E}[xx^\top])} \le 2\sqrt{2}\epsilon/\gamma^2,$$

with probability at least $1 - \delta$.

By union bound, we know for any $\epsilon < \gamma/2$, given $O(\frac{\Gamma^6(\Gamma P_1\sqrt{k}+P_2)^2 \log(\frac{d}{\delta})}{\epsilon^2})$ number of samples, with probability at least $1 - \delta$, we have

$$\left\|\hat{\mathbb{E}}\big[(u^\top y)x^\top\big] - \mathbb{E}\big[(u^\top y)x^\top\big]\right\| \le \epsilon,$$

$$\left\|\hat{\mathbb{E}}\big[xx^\top\big]^{-1} - \mathbb{E}\big[xx^\top\big]^{-1}\right\| \le 2\sqrt{2}\epsilon/\gamma^2,$$

$$\left\|\hat{\mathbb{E}}\big[(u^\top y)x(x \otimes x)^\top\big] - \mathbb{E}\big[(u^\top y)x(x \otimes x)^\top\big]\right\| \le \epsilon.$$

Define

$$E_1 := \hat{\mathbb{E}}\big[(u^\top y)x^\top\big] - \mathbb{E}\big[(u^\top y)x^\top\big],$$
$$E_2 := \hat{\mathbb{E}}\big[xx^\top\big]^{-1} - \mathbb{E}\big[xx^\top\big]^{-1},$$
$$E_3 := \hat{\mathbb{E}}\big[(u^\top y)x(x \otimes x)^\top\big] - \mathbb{E}\big[(u^\top y)x(x \otimes x)^\top\big].$$

Then, we have

$$\left\|2\hat{\mathbb{E}}\big[(u^\top y)x^\top\big]\hat{\mathbb{E}}\big[xx^\top\big]^{-1}\hat{\mathbb{E}}\big[(u^\top y)x(x \otimes x)^\top\big] - 2\mathbb{E}\big[(u^\top y)x^\top\big]\mathbb{E}\big[xx^\top\big]^{-1}\mathbb{E}\big[(u^\top y)x(x \otimes x)^\top\big]\right\|$$

$$\le 2\|E_1\|\|E_2\|\|E_3\| + 2\|E_1\|\|\mathbb{E}\big[xx^\top\big]^{-1}\|\|\mathbb{E}\big[(u^\top y)x(x \otimes x)^\top\big]\|$$

$$\quad + 2\|\mathbb{E}\big[(u^\top y)x^\top\big]\|\|E_2\|\|\mathbb{E}\big[(u^\top y)x(x \otimes x)^\top\big]\| + 2\|\mathbb{E}\big[(u^\top y)x^\top\big]\|\|\mathbb{E}\big[xx^\top\big]^{-1}\|\|E_3\|$$

$$\quad + 2\|E_1\|\|E_2\|\|\mathbb{E}\big[(u^\top y)x(x \otimes x)^\top\big]\| + 2\|\mathbb{E}\big[(u^\top y)x^\top\big]\|\|E_2\|\|E_3\| + 2\|E_1\|\|\mathbb{E}\big[xx^\top\big]^{-1}\|\|E_3\|$$

$$\le 2\Big(\frac{2\sqrt{2}\epsilon^3}{\gamma^2} + \frac{2\Gamma^3(\Gamma P_1\sqrt{k} + P_2)\epsilon}{\gamma} + \frac{8\sqrt{2}\Gamma^4(\Gamma P_1\sqrt{k} + P_2)^2\epsilon}{\gamma^2} + \frac{2\Gamma(\Gamma P_1\sqrt{k} + P_2)\epsilon}{\gamma}$$

$$\quad + \frac{4\sqrt{2}\Gamma^3(\Gamma P_1\sqrt{k} + P_2)\epsilon^2}{\gamma^2} + \frac{4\sqrt{2}\Gamma(\Gamma P_1\sqrt{k} + P_2)\epsilon^2}{\gamma^2} + \frac{\epsilon^2}{\gamma}\Big)$$

$$= O\Big(\frac{\Gamma^4(\Gamma P_1\sqrt{k} + P_2)^2\epsilon}{\gamma^2}\Big).$$

Thus, given $O(\frac{\Gamma^{14}(\Gamma P_1\sqrt{k}+P_2)^6 \log(\frac{d}{\delta})}{\gamma^4\epsilon^2})$ number of samples, we know

$$\left\|2\hat{\mathbb{E}}\big[(u^\top y) \cdot (\hat{\mathbb{E}}\big[(u^\top y)x^\top\big]\hat{\mathbb{E}}\big[xx^\top\big]^{-1}x) \cdot (x \otimes x)\big] - 2\mathbb{E}\big[(u^\top y) \cdot (\mathbb{E}\big[(u^\top y)x^\top\big]\mathbb{E}\big[xx^\top\big]^{-1}x) \cdot (x \otimes x)\big]\right\|$$

$$= \left\|2\hat{\mathbb{E}}\big[(u^\top y)x^\top\big]\hat{\mathbb{E}}\big[xx^\top\big]^{-1}\hat{\mathbb{E}}\big[(u^\top y)x(x \otimes x)^\top\big] - 2\mathbb{E}\big[(u^\top y)x^\top\big]\mathbb{E}\big[xx^\top\big]^{-1}\mathbb{E}\big[(u^\top y)x(x \otimes x)^\top\big]\right\|$$

$$\le \epsilon,$$

with probability at least $1 - \delta$.

Now, let's consider the second term

$$\left\| \hat{\mathbb{E}}\big[(u^\top y)^2 (x \otimes x)\big] - \mathbb{E}\big[(u^\top y)^2 (x \otimes x)\big] \right\|.$$

Since $\left\| (u^\top y)^2 (x \otimes x) \right\| \le 4\Gamma^2 (\Gamma P_1 \sqrt{k} + P_2)^2$, according to Lemma 24, we know given $O(\frac{\Gamma^4 (\Gamma P_1 \sqrt{k} + P_2)^4 \log(\frac{d}{\delta})}{\epsilon^2})$,

$$\left\| \hat{\mathbb{E}}\big[(u^\top y)^2 (x \otimes x)\big] - \mathbb{E}\big[(u^\top y)^2 (x \otimes x)\big] \right\| \le \epsilon,$$

with probability at least $1 - \delta$.

Next, let's look at the third term

$$\left\| \hat{\mathbb{E}}\big[(u^\top y)^2\big] \hat{\mathbb{E}}[x \otimes x] - \mathbb{E}\big[(u^\top y)^2\big] \mathbb{E}[x \otimes x] \right\|.$$

Again, using Lemma 24 and union bound, we know given $O(\frac{\Gamma^2 (\Gamma P_1 \sqrt{k} + P_2)^2 \log(\frac{d}{\delta})}{\epsilon^2})$ number of samples, we have

$$\left\| \hat{\mathbb{E}}\big[(u^\top y)^2\big] - \mathbb{E}\big[(u^\top y)^2\big] \right\| \le \epsilon,$$

$$\left\| \hat{\mathbb{E}}[x \otimes x] - \mathbb{E}[x \otimes x] \right\| \le \epsilon.$$

Thus, Define

$$E_4 := \hat{\mathbb{E}}\big[(u^\top y)^2\big] - \mathbb{E}\big[(u^\top y)^2\big]$$

$$E_5 := \hat{\mathbb{E}}[x \otimes x] - \mathbb{E}[x \otimes x].$$

Then, we have

$$\begin{aligned}
\left\| \hat{\mathbb{E}}\big[(u^\top y)^2\big] \hat{\mathbb{E}}[x \otimes x] - \mathbb{E}\big[(u^\top y)^2\big] \mathbb{E}[x \otimes x] \right\| \le & \left\| E_4 \right\| \left\| E_5 \right\| + \left\| E_4 \right\| \left\| \mathbb{E}[x \otimes x] \right\| + \left\| \mathbb{E}\big[(u^\top y)^2\big] \right\| \left\| E_5 \right\| \\
\le & \epsilon^2 + \Gamma^2 \epsilon + 4(\Gamma P_1 \sqrt{k} + P_2)^2 \epsilon \\
= & O((\Gamma P_1 \sqrt{k} + P_2)^2 \epsilon).
\end{aligned}$$

Thus, we know that given $O(\frac{\Gamma^2 (\Gamma P_1 \sqrt{k} + P_2)^6 \log(\frac{d}{\delta})}{\epsilon^2})$ number of samples, we know

$$\left\| \hat{\mathbb{E}}\big[(u^\top y)^2\big] \hat{\mathbb{E}}[x \otimes x] - \mathbb{E}\big[(u^\top y)^2\big] \mathbb{E}[x \otimes x] \right\| \le \epsilon,$$

with probability at least $1 - \delta$.

Now, let's bound the last term,

$$\left\| 2\Big( \hat{\mathbb{E}}\big[(u^\top y)x^\top\big] \hat{\mathbb{E}}\big[xx^\top\big]^{-1} \hat{\mathbb{E}}\big[(u^\top y)x\big] \Big) \hat{\mathbb{E}}[x \otimes x] - 2\Big( \mathbb{E}\big[(u^\top y)x^\top\big] \mathbb{E}\big[xx^\top\big]^{-1} \mathbb{E}\big[(u^\top y)x\big] \Big) \mathbb{E}[x \otimes x] \right\|.$$

Similar as the first term, we can show that given $O(\frac{\Gamma^{10} (\Gamma P_1 \sqrt{k} + P_2)^6 \log(\frac{d}{\delta})}{\gamma^4 \epsilon^2})$ number of samples, we have

$$\left\| 2\Big( \hat{\mathbb{E}}\big[(u^\top y)x^\top\big] \hat{\mathbb{E}}\big[xx^\top\big]^{-1} \hat{\mathbb{E}}\big[(u^\top y)x\big] \Big) \hat{\mathbb{E}}[x \otimes x] - 2\Big( \mathbb{E}\big[(u^\top y)x^\top\big] \mathbb{E}\big[xx^\top\big]^{-1} \mathbb{E}\big[(u^\top y)x\big] \Big) \mathbb{E}[x \otimes x] \right\| \le \epsilon$$

with probability at least $1 - \delta$.

Now, we are ready to combine our bound for each of four terms. By union bound, we know given $O(\frac{\Gamma^{14} (\Gamma P_1 \sqrt{k} + P_2)^6 \log(\frac{d}{\delta})}{\gamma^4 \epsilon^2})$ number of samples,

$$\left\| 2\hat{\mathbb{E}}\big[(u^\top y) \cdot (\hat{\mathbb{E}}[(u^\top y)x^\top]\hat{\mathbb{E}}[xx^\top]^{-1}x) \cdot (x \otimes x)\big] - 2\mathbb{E}\big[(u^\top y) \cdot (\mathbb{E}[(u^\top y)x^\top]\mathbb{E}[xx^\top]^{-1}x) \cdot (x \otimes x)\big] \right\| \le \epsilon$$

$$\left\| \hat{\mathbb{E}}\big[(u^\top y)^2 (x \otimes x)\big] - \mathbb{E}\big[(u^\top y)^2 (x \otimes x)\big] \right\| \le \epsilon,$$

$$\left\| \hat{\mathbb{E}}\big[(u^\top y)^2\big] \hat{\mathbb{E}}[x \otimes x] - \mathbb{E}\big[(u^\top y)^2\big] \mathbb{E}[x \otimes x] \right\| \le \epsilon,$$

$$\left\| 2\Big( \hat{\mathbb{E}}\big[(u^\top y)x^\top\big] \hat{\mathbb{E}}\big[xx^\top\big]^{-1} \hat{\mathbb{E}}\big[(u^\top y)x\big] \Big) \hat{\mathbb{E}}[x \otimes x] - 2\Big( \mathbb{E}\big[(u^\top y)x^\top\big] \mathbb{E}\big[xx^\top\big]^{-1} \mathbb{E}\big[(u^\top y)x\big] \Big) \mathbb{E}[x \otimes x] \right\| \le \epsilon,$$

hold with probability at least $1 - \delta$. Thus, we know

$$\max_{u:\|u\|\leq 2} \|f(u) - \hat{f}(u)\| \leq \epsilon + \epsilon + \epsilon + \epsilon = 4\epsilon$$

with probability at least $1 - \delta$.

Recall that

$$\|\hat{T} - T\|_F \leq \sqrt{k_2 + k}\|\hat{T} - T\|$$
$$\leq k\sqrt{2k} \max_{u:\|u\|\leq 2} \|f(u) - \hat{f}(u)\|,$$

where the second inequality holds since $\|\hat{T} - T\| \leq \sqrt{2k} \max_{u:\|u\|\leq 2} \|f(u) - \hat{f}(u)\|$.

Thus, we know given $O(\frac{\Gamma^{14}(\Gamma P_1 \sqrt{k} + P_2)^6 \log(\frac{d}{\delta})}{\gamma^4 \epsilon^2})$ number of samples,

$$\|\hat{T} - T\|_F \leq k\sqrt{2k}\epsilon \leq d\sqrt{2d}\epsilon$$

with probability at least $1 - \delta$. Thus, given $O(\frac{d^3 \Gamma^{14}(\Gamma P_1 \sqrt{k} + P_2)^6 \log(\frac{d}{\delta})}{\gamma^4 \epsilon^2})$ number of samples,

$$\|\hat{T} - T\|_F \leq \epsilon$$

with probability at least $1 - \delta$. $\qquad\square$

## B.2 ROBUST ANALYSIS FOR SIMULTANEOUS DIAGONALIZATION

In this section, we will show that the simultaneous diagonalization step in our algorithm is robust. Let $S$ and $\hat{S}$ be two $(k_2 + k)$ by $k$ matrices, whose columns consist of the least $k$ right singular vectors of $T$ and $\hat{T}$ respectively.

According to Lemma 11, we know with polynomial number of samples, the Frobenius norm of $SS^\top - \hat{S}\hat{S}^\perp$ is well bounded. However, due to the rotation issue of subspace basis, we cannot conclude that $\|S - \hat{S}\|_F$ is small. Only after appropriate alignment, the difference between $S$ and $\hat{S}$ becomes small.

**Lemma 16.** *Let $S$ and $\hat{S}$ be two $(k_2 + k)$ by $k$ matrices, whose columns consist of the least $k$ right singular vectors of $T$ and $\hat{T}$ respectively. If $\|SS^\top - \hat{S}\hat{S}^\top\|_F \leq \epsilon$, there exists an rotation matrix $R \in \mathbb{R}^{k \times k}$ satisfying $RR^\top = R^\top R = I_k$, such that*

$$\|\hat{S} - SR\|_F \leq 2\epsilon.$$

*Proof.* Since $S$ has orthonormal columns, we have $\sigma_k(SS^\top) = 1$. Then, according to Lemma 35, we know there exists rotation matrix $R$ such that

$$\|\hat{S} - SR\|_F \leq \frac{\|SS^\top - \hat{S}\hat{S}^\top\|_F}{\sqrt{2(\sqrt{2} - 1)}\sqrt{\sigma_k(SS^\top)}} \leq 2\epsilon.$$

$\qquad\square$

Let the $k$ columns of $S$ be $\text{vec}^*(U_1), \text{vec}^*(U_2), \cdots, \text{vec}^*(U_k)$. Note each $U_i$ can be expressed as $A^{-\top} D_i A^{-1}$, where $D_i$ is a diagonal matrix. Let $Q$ be a $k \times k$ matrix, whose $i$-th column consists of the diagonal elements of $D_i$, such that $Q_{ij}$ equals the $j$-th diagonal element of $D_i$. Let $\text{vec}^*(X) = SR\zeta_1, \text{vec}^*(Y) = SR\zeta_2$, where $R$ is the rotation matrix in Lemma 16 and $\zeta_1, \zeta_2$ are two independent standard Gaussian vectors. Let $D_X = \text{diag}(QR\zeta_1)$ and $D_Y = \text{diag}(QR\zeta_2)$. It's not hard to check that $X = A^{-\top} D_X A^{-1}$ and $Y = A^{-\top} D_Y A^{-1}$. Furthermore, we have $XY^{-1} = A^{-\top} D_X D_Y^{-1} A^\top$. Next, we show that the diagonal elements of $D_X D_Y^{-1}$ are well separated.

**Lemma 17.** *Assume that $\|A\| \leq P_1, \sigma_{\min}(A) \geq \beta$. Then for any $\delta > 0$ we know with probability at least $1 - \delta$, we have*

$$sep(D_X D_Y^{-1}) \geq poly(1/d, 1/P_1, \beta, \delta),$$

*where $sep(D_X D_Y^{-1}) := \min_{i \neq j} |(D_X D_Y^{-1})_{ii} - (D_X D_Y^{-1})_{jj}|$.*

*Proof.* We first show that matrix $Q$ is well-conditioned. Since $U_i = A^{-\top} D_i A^{-1}$, we have $\text{vec}(U_i) = A^{-\top} \otimes A^{-\top} \text{vec}(D_i)$. Let $U$ be a $k^2 \times k$ matrix whose columns consist of $\text{vec}(U_i)$'s. Also define $\bar{Q}$ as a $k^2 \times k$ matrix whose columns are $\text{vec}(D_i)$'s. Note that matrix $\bar{Q}$ only has $k$ non-zero rows, which are exactly matrix $Q$. With the above definition, we have $U = A^{-\top} \otimes A^{-\top} \bar{Q}$. Since $\sigma_{\min}(U) \leq \|A^{-\top} \otimes A^{-\top}\| \sigma_{\min}(\bar{Q})$, we have

$$\sigma_{\min}(\bar{Q}) \geq \frac{\sigma_{\min}(U)}{\|A^{-\top} \otimes A^{-\top}\|}.$$

Notice that a subset of rows of $U$ constitute matrix $S$, which is an orthonormal matrix. Thus, we have $\sigma_{\min}(U) \geq \sigma_{\min}(S) = 1$. Since we assume $\sigma_{\min}(A) \geq \beta$, we have

$$\|A^{-\top} \otimes A^{-\top}\| = \|A^{-\top}\|^2 = \frac{1}{\sigma_{\min}(A)^2} \leq \frac{1}{\beta^2}.$$

Thus, we have $\sigma_{\min}(\bar{Q}) \geq \beta^2$, which implies $\sigma_{\min}(Q) \geq \beta^2$.

We also know $\|U\| \geq \sigma_{\min}(A^{-\top} \otimes A^{-\top})\|\bar{Q}\|$, thus

$$\|\bar{Q}\| \leq \frac{\|U\|}{\sigma_{\min}(A^{-\top} \otimes A^{-\top})}.$$

Since $\|S\| = 1$, we know $\|U\| \leq \sqrt{2}$. For the smallest singular value of $A^{-\top} \otimes A^{-\top}$, we have

$$\sigma_{\min}(A^{-\top} \otimes A^{-\top}) = \sigma_{\min}^2(A^{-1}) = \frac{1}{\|A\|^2} \geq \frac{1}{P_1^2}.$$

Thus, we have $\|\bar{Q}\| \leq \sqrt{2} P_1^2$, which implies $\|Q\| \leq \sqrt{2} P_1^2$.

Now, let's prove that the diagonal elements of $D_X D_Y^{-1}$ are well-separated. Let $q_i^\top$ be the $i$-th row vector of $Q$. Then we know the $i$-th diagonal element of $D_X D_Y^{-1}$ is $\frac{\langle q_i, R\zeta_1 \rangle}{\langle q_i, R\zeta_2 \rangle}$. Since $\|Q\| \leq \sqrt{2} P_1^2$, we have $\|q_i\| \leq \sqrt{2} P_1^2$ for every row vector.

It's not hard to show that with probability at least $1 - \exp(-d^{\Omega(1)})$, we have $|\langle q_i, R\zeta_2 \rangle| \leq \text{poly}(d, P_1)$ for each $i$. Now given $\zeta_2$ for which this happens, we have $\frac{\langle q_i, R\zeta_1 \rangle}{\langle q_i, R\zeta_2 \rangle} - \frac{\langle q_j, R\zeta_1 \rangle}{\langle q_j, R\zeta_2 \rangle} = c_i \langle q_i, R\zeta_1 \rangle - c_j \langle q_j, R\zeta_1 \rangle$, where $c_i, c_j$ have magnitude as least $\text{poly}(1/d, 1/P_1)$. Since $\sigma_{\min}(Q) \geq \beta^2$, we know $\|\text{Proj}_{q_j^\perp} q_i\| \geq \beta^2$ (because otherwise there exists $\lambda$ such that $\|(e_i + \lambda e_j)\| \sigma_{\min}(Q) \leq \|(e_i + \lambda e_j)^\top Q\| = \|\text{Proj}_{q_j^\perp} q_i\| < \beta^2$, which is a contradiction). Let $q_{i,j}^\perp = \text{Proj}_{q_j^\perp} q_i = q_i - \lambda_{i,j} q_j$, we can rewrite this as

$$\frac{\langle q_i, R\zeta_1 \rangle}{\langle q_i, R\zeta_2 \rangle} - \frac{\langle q_j, R\zeta_1 \rangle}{\langle q_j, R\zeta_2 \rangle} = c_i \langle q_i, R\zeta_1 \rangle - c_j \langle q_j, R\zeta_1 \rangle = c_i \langle q_{i,j}^\perp, R\zeta_1 \rangle - (c_j + \lambda_{i,j} c_i) \langle q_j, R\zeta_1 \rangle.$$

By properties of Gaussians, we know $\langle q_{i,j}^\perp, R\zeta_1 \rangle$ is independent of $\langle q_j, R\zeta_1 \rangle$, so we can first fix $\langle q_j, R\zeta_1 \rangle$ and apply anti-concentration of Gaussians (see Lemma 38) to $\langle q_{i,j}^\perp, R\zeta_1 \rangle$. As a result we know with probability at least $1 - \delta/k^2$:

$$\left| \frac{\langle q_i, R\zeta_1 \rangle}{\langle q_i, R\zeta_2 \rangle} - \frac{\langle q_j, R\zeta_1 \rangle}{\langle q_j, R\zeta_2 \rangle} \right| \geq \text{poly}(1/d, 1/P_1, \beta, \delta).$$

By union bound, we know with probability at least $1 - \delta$,

$$\text{sep}(D_X D_Y^{-1}) \geq \text{poly}(1/d, 1/P_1, \beta, \delta).$$

$\square$

Let $\hat{X} = \hat{S}\zeta_1$ and $\hat{Y} = \hat{S}\zeta_2$. Next, we prove that the eigenvectors of $\hat{X}\hat{Y}^{-1}$ are close to the eigenvectors of $XY^{-1}$.

**Lemma 12.** *Suppose that $\|SS^\top - \hat{S}\hat{S}^\top\|_F \leq \epsilon, \|A\| \leq P_1, \|\xi\| \leq P_2, \sigma_{\min}(\mathbb{E}[xx^\top]) \geq \gamma, \sigma_{\min}(A) \geq \beta$. Let $\hat{X} = mat^*(\hat{S}\zeta_1), \hat{Y} = mat^*(\hat{S}\zeta_2)$, where $\zeta_1$ and $\zeta_2$ are two independent standard Gaussian vectors. Let $z_1, \cdots z_k$ be the normalized row vectors of $A^{-1}$. Let $\hat{z}_1, ..., \hat{z}_k$ be the eigenvectors of $\hat{X}\hat{Y}^{-1}$ (after sign flip). For any $\delta > 0$ and small enough $\epsilon$, with $O(\frac{(\Gamma P_1 \sqrt{k} + P_2)^2 \log(d/\delta)}{\epsilon^2})$ number of i.i.d. samples in $\hat{\mathbb{E}}[\hat{z}_i^\top y]$, with probability at least $1 - \delta$ over the randomness of $\zeta_1, \zeta_2$ and i.i.d. samples, there exists a permutation $\pi(i) \in [k]$ such that*

$$\|\hat{z}_i - z_{\pi[i]}\| \leq poly(d, P_1, 1/\beta, \epsilon, 1/\delta),$$

*for any $1 \leq i \leq k$.*

*Proof.* Let $z_1', \cdots, z_k'$ be the eigenvectors of $XY^{-1}$ (before sign flip step). Similarly define $\hat{z}_1', \cdots, \hat{z}_k'$ for $\hat{X}\hat{Y}^{-1}$. We first prove that the eigenvectors of $\hat{X}\hat{Y}^{-1}$ are close to the eigenvectors of $XY^{-1}$.

Let $\hat{X} = X + E_X$ and $\hat{Y} = Y + E_Y$. Then we have

$$\hat{X}\hat{Y}^{-1} = XY^{-1}(I + F) + G,$$

where $F = -E_Y(I + Y^{-1}E_Y)^{-1}Y^{-1}$ and $G = E_X\hat{Y}^{-1}$. According to Lemma 34, we have $\|F\| \leq \frac{\|E_Y\|}{\sigma_{\min}(Y) - \|E_Y\|}$ and $\|G\| \leq \frac{\|E_X\|}{\sigma_{\min}(\hat{Y})}$. In order to bound the perturbation matrices $\|F\|$ and $\|G\|$, we need to first bound $\|E_X\|, \|E_Y\|$ and $\sigma_{\min}(Y), \sigma_{\min}(\hat{Y})$.

As we know, $E_X = \hat{X} - X = (\hat{S} - SR)\zeta_1$. According to Lemma 16, we have $\|\hat{S} - SR\| \leq \|\hat{S} - SR\|_F \leq 2\epsilon$. We also know with probability at least $1 - \exp(-d^{\Omega(1)})$, $\|\zeta_1\| \leq poly(d)$. Thus, we have

$$\|E_X\| = \|(\hat{S} - SR)\zeta_1\| \leq \|\hat{S} - SR\|\|\zeta_1\| \leq poly(\epsilon, d).$$

Similarly, with probability at least $1 - \exp(-d^{\Omega(1)})$, we also know $\|E_Y\| \leq poly(\epsilon, d)$.

Now, we lower bound the smallest singular value of $X$ and $Y$. Since $X = A^{-\top}D_X A^{-1}$, we have

$$\sigma_{\min}(X) \geq \sigma_{\min}^2(A^{-1})\sigma_{\min}(D_X)$$
$$= \frac{1}{\|A\|^2}\sigma_{\min}(D_X)$$
$$\geq \frac{1}{P_1^2}\sigma_{\min}(D_X).$$

Since $D_X$ is a diagonal matrix, its smallest singular value equals the smallest absolute value of its diagonal element. Recall each diagonal element of $D_X$ is $\langle q_i, R\zeta_1 \rangle$, which follows a Gaussian distribution whose standard deviation is at least $\|q_i\| \geq \beta^2$. By anti-concentration property of Gaussian (see Lemma 38), we know $|\langle q_i, R\zeta_2 \rangle| \geq \Omega(\delta\beta^2/k)$ for all $i$ with probability $1 - \delta/4$. Thus, we have $\sigma_{\min}(X) \geq poly(1/d, 1/P_1, \beta, \delta)$. Similarly we have the same conclusion for $Y$. For small enough $E_Y$, we have $\sigma_{\min}(\hat{Y}) \geq \sigma_{\min}(Y) - \|E_Y\|$.

Thus, for small enough $\epsilon$, we have $\|F\| \leq poly(d, P_1, 1/\beta, \epsilon, 1/\delta)$ and $\|G\| \leq poly(d, P_1, 1/\beta, \epsilon, 1/\delta)$. In order to apply Lemma 33, we also need to bound $\kappa(A^{-\top})$ and $\|XY^{-1}\|$. Since $\sigma_{\min}(A^{-\top}) \geq \frac{1}{P_1}$ and $\|A^{-\top}\| \leq 1/\beta$, we have $\kappa(A^{-\top}) \leq P_1/\beta$. For the norm of $XY^{-1}$, we have

$$\|XY^{-1}\| \leq \frac{\|X\|}{\sigma_{\min}(Y)} \leq \|X\|poly(d, P_1, 1/\beta, 1/\delta),$$

where the second inequality holds because $\sigma_{\min}(Y) \geq poly(1/d, 1/P_1, \beta, \delta)$. Recall that $X = mat^*(SR\zeta_1)$. It's not hard to verify that with probability at least $1 - \exp(-d^{\Omega(1)})$, we have $\|X\| \leq poly(d)$. Thus, we know $\|XY^{-1}\| \leq poly(d, P_1, 1/\beta, 1/\delta)$. Similarly, we can also prove that $\|D_X D_Y^{-1}\| \leq poly(d, P_1, 1/\beta, 1/\delta)$

Overall, we have

$$\kappa(A^{-\top})(\|XY^{-1}F\| + \|G\|) \leq \kappa(A^{-\top})(\|XY^{-1}\|\|F\| + \|G\|)$$
$$\leq \frac{P_1}{\beta}\Big(poly(d, P_1, 1/\beta, 1/\delta)poly(d, P_1, 1/\beta, \epsilon, 1/\delta) + poly(d, P_1, 1/\beta, \epsilon, 1/\delta)\Big)$$
$$\leq poly(d, P_1, 1/\beta, \epsilon, 1/\delta).$$

According to Lemma 17, we know with probability at least $1 - \delta/4$, $\text{sep}(D_X D_Y^{-1}) \geq \text{poly}(1/d, 1/P_1, \beta, \delta)$. Thus, by union bound, we know for small enough $\epsilon$, with probability at least $1 - \delta$,

$$\kappa(A^{-\top})(\|XY^{-1}F\| + \|G\|) < \text{sep}(D_X D_Y^{-1})/(2k).$$

According to Lemma 33, we know there exists a permutation $\pi[i] \in [k]$, such that

$$
\begin{aligned}
\|\hat{z}_i' - z_i'\| \leq & 3\frac{\|F\|\|D_X D_Y^{-1}\| + \|G\|}{\sigma_{\min}(A^{-\top})\text{sep}(D_X D_Y^{-1})} \\
\leq & 3\frac{\text{poly}(d, P_1, 1/\beta, \epsilon, 1/\delta)\text{poly}(d, P_1, 1/\beta, 1/\delta) + \text{poly}(d, P_1, 1/\beta, \epsilon, 1/\delta)}{1/P_1\text{poly}(1/d, 1/P_1, \beta, 1/\delta)} \\
\leq & \text{poly}(d, P_1, 1/\beta, \epsilon, 1/\delta).
\end{aligned}
$$

with probability at least $1 - \delta$.

According to Lemma 5, the eigenvectors of $XY^{-1}$ (after sign flip) are exactly the normalized rows of $A^{-1}$ (up to permutation). Now the only issue is the sign of $\hat{z}_i'$. By the robustness of sign flip step (see Lemma 18), we know for small enough $\epsilon$, with $O(\frac{(\Gamma P_1 \sqrt{k} + P_2)^2 \log(d/\delta)}{\epsilon^2})$ number of i.i.d. samples in $\hat{\mathbb{E}}[\hat{z}_i^\top y]$, with probability at least $1 - \delta$, the sign flip of $\hat{z}_i'$ is consistent with the sign flip of $z_i'$. $\qquad\square$

In the following lemma, we show that the sign flip step of $\hat{z}_i$ is robust.

**Lemma 18.** *Suppose that $\|x\| \leq \Gamma, \|A\| \leq P_1, \|\xi\| \leq P_2$. Let $z_1', \cdots, z_k'$ be the eigenvectors of $XY^{-1}$ (before sign flip step). Similarly define $\hat{z}_1', \cdots, \hat{z}_k'$ for $\hat{X}\hat{Y}^{-1}$. Suppose for each $i$, $\|z_i' - \hat{z}_i'\| \leq \epsilon$, where $\epsilon \leq \frac{\beta\gamma}{4\Gamma(1 + \Gamma P_1\sqrt{k} + P_2)}$. We know, for any $\delta < 1$, with $O(\frac{(\Gamma P_1\sqrt{k} + P_2)^2 \log(d/\delta)}{\epsilon^2})$ number of i.i.d. samples,*

$$\Pr\left[sign(\mathbb{E}[\langle z_i', y\rangle]) \neq sign(\hat{\mathbb{E}}[\langle \hat{z}_i', y\rangle])\right] \leq \delta.$$

*Proof.* We first show that $\mathbb{E}[\langle z_i', y\rangle]$ is bounded away from zero. Let $Z'$ be a $k \times k$ matrix, whose rows are $\{z_i'\}$. Without loss of generality, assume that $Z' = \text{diag}(\pm\lambda)A^{-1}$. Since $\|A^{-1}\| \leq 1/\beta$, we know $\lambda_i \geq \beta$, for each $i$. Thus, we have

$$|\mathbb{E}[\langle z_i', y\rangle]| = \lambda_i \mathbb{E}[\sigma(w_i^\top x)] \geq \beta\mathbb{E}[\sigma(w_i^\top x)].$$

In order to lowerbound $\mathbb{E}[\sigma(w_i^\top x)]$, let's first look at $\mathbb{E}[\sigma(w_i^\top x)x^\top]$.

$$
\begin{aligned}
\|\mathbb{E}[\sigma(w_i^\top x)x^\top]\| = & \frac{1}{2}\|\mathbb{E}[w_i^\top xx^\top]\| \\
\geq & \frac{1}{2}\|w_i\|\sigma_{\min}(\mathbb{E}[xx^\top]) \\
\geq & \frac{\gamma}{2}.
\end{aligned}
$$

Now, let's connect $\|\mathbb{E}[\sigma(w_i^\top x)x^\top]\|$ with $\mathbb{E}[\sigma(w_i^\top x)]$.

$$
\begin{aligned}
\|\mathbb{E}[\sigma(w_i^\top x)x^\top]\| \leq & \mathbb{E}[\|\sigma(w_i^\top x)x^\top\|] \\
= & \mathbb{E}[\sigma(w_i^\top x)\|x\|] \\
\leq & \Gamma\mathbb{E}[\sigma(w_i^\top x)].
\end{aligned}
$$

Thus, we have $\mathbb{E}[\sigma(w_i^\top x)] \geq \frac{\gamma}{2\Gamma}$, and further $|\mathbb{E}[\langle z_i', y\rangle]| \geq \frac{\beta\gamma}{2\Gamma}$.

Now, let's bound $\|\mathbb{E}[\langle z_i', y\rangle] - \hat{\mathbb{E}}[\langle \hat{z}_i', y\rangle]\|$.

$$
\begin{aligned}
\|\mathbb{E}[\langle z_i', y\rangle] - \hat{\mathbb{E}}[\langle \hat{z}_i', y\rangle]\| = & \|\mathbb{E}[\langle z_i', y\rangle] - \mathbb{E}[\langle \hat{z}_i', y\rangle] + \mathbb{E}[\langle \hat{z}_i', y\rangle] - \hat{\mathbb{E}}[\langle \hat{z}_i', y\rangle]\| \\
\leq & \|\mathbb{E}[\langle z_i' - \hat{z}_i', y\rangle]\| + \|\mathbb{E}[\langle \hat{z}_i', y\rangle] - \hat{\mathbb{E}}[\langle \hat{z}_i', y\rangle]\| \\
\leq & (\Gamma P_1\sqrt{k} + P_2)\epsilon + \|\mathbb{E}[\langle \hat{z}_i', y\rangle] - \hat{\mathbb{E}}[\langle \hat{z}_i', y\rangle]\|.
\end{aligned}
$$

Note that we reserve fresh samples for this step, thus $\hat{z}_i'$ is independent with samples in $\hat{\mathbb{E}}[\langle \hat{z}_i', y \rangle]]$. Since $\|\langle \hat{z}_i', y \rangle\| \leq \Gamma P_1 \sqrt{k} + P_2$, we know with $O(\frac{(\Gamma P_1 \sqrt{k} + P_2)^2 \log(d/\delta)}{\epsilon^2})$ number of samples, we have

$$\|\mathbb{E}[\langle \hat{z}_i', y \rangle] - \hat{\mathbb{E}}[\langle \hat{z}_i', y \rangle]]\| \leq \epsilon.$$

Overall, we have $\|\mathbb{E}[\langle z_i', y \rangle] - \hat{\mathbb{E}}[\langle \hat{z}_i', y \rangle]]\| \leq (1 + \Gamma P_1 \sqrt{k} + P_2)\epsilon$. Combined with the fact that $|\mathbb{E}[\langle z_i', y \rangle]| \geq \frac{\beta\gamma}{2\Gamma}$, we know as long as $\epsilon \leq \frac{\beta\gamma}{4\Gamma(1+\Gamma P_1 \sqrt{k}+P_2)}$, with $O(\frac{(\Gamma P_1 \sqrt{k}+P_2)^2 \log(d/\delta)}{\epsilon^2})$ number of samples, we have

$$\Pr[\text{sign}(\mathbb{E}[\langle z_i', y \rangle]) \neq \text{sign}(\hat{\mathbb{E}}[\langle \hat{z}_i', y \rangle])] \leq \delta.$$

$\square$

### B.3 ROBUST ANALYSIS FOR RECOVERING FIRST LAYER WEIGHTS

We will first show that Algorithm 1 is robust.

**Theorem 8.** *Assume that* $\|x\| \leq \Gamma, \|w\| \leq 1, |\xi| \leq P_2$ *and* $\sigma_{\min}(\mathbb{E}[xx^\top]) \geq \gamma$. *Then for any* $\epsilon \leq \gamma/2$, *for any* $1 > \delta > 0$, *given* $O(\frac{(\Gamma^2+P_2\Gamma)^4 \log(\frac{d}{\delta})}{\gamma^4 \epsilon^2})$ *number of i.i.d. samples, we know with probability at least* $1 - \delta$,

$$\|\hat{w} - w\| \leq \epsilon,$$

*where* $\hat{w}$ *is the learned weight vector.*

*Proof.* We first show that given polynomial number of i.i.d. samples, $\|\hat{\mathbb{E}}[yx] - \mathbb{E}[yx]\|$ is upper bounded with high probability, where $\hat{\mathbb{E}}[yx]$ is the empirical estimate of $\mathbb{E}[yx]$. Due to the assumption that $\|x\| \leq \Gamma, \|w\| \leq 1, \|\xi\| \leq P_2$, we have

$$
\begin{aligned}
\|yx\| &= \|(\sigma(w^\top x) + \xi)x\| \\
&\leq \|w^\top x x\| + \|\xi x\| \\
&\leq \|w\|\|x\|^2 + |\xi|\|x\| \\
&\leq \Gamma^2 + P_2\Gamma
\end{aligned}
$$

According to Lemma 24, we know given $O(\frac{(\Gamma^2+P_2\Gamma)^2 \log(\frac{d}{\delta})}{\epsilon^2})$ number of samples, with probability at least $1 - \delta$, we have $\|\hat{\mathbb{E}}[yx] - \mathbb{E}[yx]\| \leq \epsilon$.

Since $\|xx^\top\| \leq \Gamma^2$, we know that given $O(\frac{\Gamma^4 \log(\frac{d}{\delta})}{\epsilon^2})$ number of samples, $\|\hat{\mathbb{E}}[xx^\top] - \mathbb{E}[xx^\top]\| \leq \epsilon$ with probability at least $1 - \delta$. Suppose that $\epsilon \leq \gamma/2 \leq \sigma_{\min}(\mathbb{E}[xx^\top])/2$, we know $\hat{\mathbb{E}}[xx^\top]$ has full rank. According to Lemma 29, we have

$$\left\|\hat{\mathbb{E}}[xx^\top]^{-1} - \mathbb{E}[xx^\top]^{-1}\right\| \leq 2\sqrt{2}\frac{\|\hat{\mathbb{E}}[xx^\top] - \mathbb{E}[xx^\top]\|}{\sigma_{\min}^2(\mathbb{E}[xx^\top])} \leq 2\sqrt{2}\epsilon/\gamma^2,$$

with probability at least $1 - \delta$.

By union bound, we know for any $\epsilon \leq \gamma/2$, given $O(\frac{(\Gamma^2+P_2\Gamma)^2 \log(\frac{d}{\delta})}{\epsilon^2})$ number of samples, with probability at least $1 - \delta$, we have

$$\|\hat{\mathbb{E}}[xx^\top]^{-1} - \mathbb{E}[xx^\top]^{-1}\| \leq 2\sqrt{2}\epsilon/\gamma^2$$
$$\|\hat{\mathbb{E}}[yx] - \mathbb{E}[yx]\| \leq \epsilon$$

Denote

$$
\begin{aligned}
E_1 &:= \hat{\mathbb{E}}[xx^\top]^{-1} - \mathbb{E}[xx^\top]^{-1} \\
E_2 &:= \hat{\mathbb{E}}[yx] - \mathbb{E}[yx].
\end{aligned}
$$

Then, we have

$$
\begin{aligned}
\|\hat{w} - w\| =& 2\|\hat{\mathbb{E}}[xx^\top]^{-1}\hat{\mathbb{E}}[yx] - \mathbb{E}[xx^\top]^{-1}\mathbb{E}[yx]\| \\
\leq& 2\|E_1\|\|E_2\| + 2\|E_1\|\|\mathbb{E}[yx]\| + 2\|\mathbb{E}[xx^\top]^{-1}\|\|E_2\| \\
\leq& \frac{4\sqrt{2}\epsilon^2}{\gamma^2} + \frac{4\sqrt{2}(\Gamma^2 + P_2\Gamma)\epsilon}{\gamma^2} + \frac{2\epsilon}{\gamma} \\
\leq& O(\frac{(\Gamma^2 + P_2\Gamma)\epsilon}{\gamma^2}).
\end{aligned}
$$

Thus, given $O(\frac{(\Gamma^2 + P_2\Gamma)^4 \log(\frac{d}{\delta})}{\gamma^4 \epsilon^2})$ number of samples, with probability at least $1 - \delta$, we have

$$
\|\hat{w} - w\| \leq \epsilon.
$$

$\square$

Now let's go back to the call to Algorithm 1 in Algorithm 3. Let $z_i$'s be the normalized rows of $A^{-1}$, and let $\hat{z}_i$'s be the eigenvectors of $\hat{X}\hat{Y}^{-1}$ (with correct sign). From Lemma 12, we know $\{\hat{z}_i\}$ are close to $\{z_i\}$ with permutation. Without loss of generality, we assume the permutation here is just an identity mapping, which means $\|z_i - \hat{z}_i\|$ is small for each $i$.

For each $z_i$, let $v_i$ be the output of Algorithm 1 given infinite number of inputs $(x, z_i^\top y)$. For each $\hat{z}_i$, let $\hat{v}_i$ be the output of Algorithm 1 given only finite number of samples $(x, \hat{z}_i^\top y)$. In this section, we show that suppose $\|z_i - \hat{z}_i\|$ is bounded, with polynomial number of samples, $\|v_i - \hat{v}_i\|$ is also bounded.

The input for Algorithm 1 is $(x, \hat{z}_i^\top y)$. We view $\hat{z}_i^\top y$ as the summation of $z_i^\top y$ and a noise term $(\hat{z}_i - z_i)^\top y$. Here, the issue is that the noise term $(\hat{z}_i - z_i)^\top y$ is not independent with the sample $(x, \hat{z}_i^\top y)$, which makes the robust analysis in Theorem 8 not applicable. On the other hand, since we reserve a separate set of samples for Algorithm 1, the estimate $\hat{z}_i$ is independent with the samples $(x, y)$'s used by Algorithm 1. Thus, the samples $(x, \hat{z}_i^\top y)$'s here are still i.i.d., which enables us to use matrix concentration bounds to show the robustness here.

**Lemma 13.** *Assume that $\|x\| \leq \Gamma, \|A\| \leq P_1, \|\xi\| \leq P_2$ and $\sigma_{\min}(\mathbb{E}[xx^\top]) \geq \gamma$. Suppose that for each $1 \leq i \leq k$, $\|\hat{z}_i - z_i\| \leq \tau$. Then for any $\epsilon \leq \gamma/2$ and $\delta < 1$, given $O(\frac{(\Gamma^2 + P_2\Gamma)^4 \log(\frac{d}{\delta})}{\gamma^4 \epsilon^2})$ number of samples for Algorithm 1, we know with probability at least $1 - \delta$,*

$$
\|v_i - \hat{v}_i\| \leq \frac{2\tau(\Gamma P_1 \sqrt{k} + P_2)\Gamma}{\gamma} + 2\epsilon,
$$

*for each $1 \leq i \leq k$.*

*Proof.* For any $i$, let's bound $\|v_i - \hat{v}_i\|$. As we know, $v_i = 2\mathbb{E}[xx^\top]^{-1}\mathbb{E}[z_i^\top yx]$ and $\hat{v}_i = 2\hat{\mathbb{E}}[xx^\top]^{-1}\hat{\mathbb{E}}[\hat{z}_i^\top yx]$. Thus, in order to bound $\|v_i - \hat{v}_i\|$, we only need to show

$$
\left\|\mathbb{E}[xx^\top]^{-1}\mathbb{E}[\hat{z}_i^\top yx] - \mathbb{E}[xx^\top]^{-1}\mathbb{E}[z_i^\top yx]\right\| \text{ and } \left\|\mathbb{E}[xx^\top]^{-1}\mathbb{E}[\hat{z}_i^\top yx] - \hat{\mathbb{E}}[xx^\top]^{-1}\hat{\mathbb{E}}[\hat{z}_i^\top yx]\right\|
$$

are both bounded.

The first term can be bounded as follows.

$$
\begin{aligned}
\left\|\mathbb{E}[xx^\top]^{-1}\mathbb{E}[\hat{z}_i^\top yx] - \mathbb{E}[xx^\top]^{-1}\mathbb{E}[z_i^\top yx]\right\| =& \left\|\mathbb{E}[xx^\top]^{-1}\mathbb{E}[(\hat{z}_i - z_i)^\top yx]\right\| \\
\leq& \left\|\mathbb{E}[xx^\top]^{-1}\right\|\|\hat{z}_i - z_i\|\|A\sigma(Wx) + \xi\|\|x\| \\
\leq& \frac{\tau(\Gamma P_1 \sqrt{k} + P_2)\Gamma}{\gamma}
\end{aligned}
$$

We can use standard matrix concentration bounds to upper bound the second term. By similar analysis of Theorem 1, we know given $O(\frac{(\Gamma^2 + P_2\Gamma)^4 \log(\frac{d}{\delta})}{\gamma^4 \epsilon^2})$ number of i.i.d. samples, with probability at least $1 - \delta$,

$$
\left\|\mathbb{E}[xx^\top]^{-1}\mathbb{E}[\hat{z}_i^\top yx] - \hat{\mathbb{E}}[xx^\top]^{-1}\hat{\mathbb{E}}[\hat{z}_i^\top yx]\right\| \leq \epsilon.
$$

Overall, we have

$$
\begin{aligned}
\|v_i - \hat{v}_i\| =& \big\|2\mathbb{E}[xx^\top]^{-1}\mathbb{E}[z_i^\top yx] - 2\hat{\mathbb{E}}[xx^\top]^{-1}\hat{\mathbb{E}}[\hat{z}_i^\top yx]\big\| \\
\leq& 2\big\|\mathbb{E}[xx^\top]^{-1}\mathbb{E}[\hat{z}_i^\top yx] - \mathbb{E}[xx^\top]^{-1}\mathbb{E}[z_i^\top yx]\big\| + 2\big\|\mathbb{E}[xx^\top]^{-1}\mathbb{E}[\hat{z}_i^\top yx] - \hat{\mathbb{E}}[xx^\top]^{-1}\hat{\mathbb{E}}[\hat{z}_i^\top yx]\big\| \\
\leq& \frac{2\tau(\Gamma P_1\sqrt{k} + P_2)\Gamma}{\gamma} + 2\epsilon.
\end{aligned}
$$

By union bound, we know given $O(\frac{(\Gamma^2 + P_2\Gamma)^4 \log(\frac{d}{\delta})}{\gamma^4\epsilon^2})$ number of i.i.d. samples, with probability at least $1 - \delta$,

$$
\|v_i - \hat{v}_i\| \leq \frac{2\tau(\Gamma P_1\sqrt{k} + P_2)\Gamma}{\gamma} + 2\epsilon.
$$

for any $1 \leq i \leq k$. $\qquad\square$

### B.4 PROOF OF THEOREM 7

**Proof of Theorem 7.** Combining Lemma 11, Lemma 12 and Lemma 13, we know given $\text{poly}\big(\Gamma, P_1, P_2, d, 1/\epsilon, 1/\gamma, 1/\alpha, 1/\beta, 1/\delta\big)$ number of i.i.d. samples, with probability at least $1 - \delta$,

$$
\|z_i - \hat{z}_i\| \leq \epsilon, \|v_i - \hat{v}_i\| \leq \epsilon
$$

for each $1 \leq i \leq k$.

Let $V$ be a $k \times d$ matrix whose rows are $v_i$'s. Similarly define matrix $\hat{V}$ for $\hat{v}_i$'s. Since $\|v_i - \hat{v}_i\| \leq \epsilon$ for any $i$, we know every row vector of $V - \hat{V}$ has norm at most $\epsilon$, which implies $\|V - \hat{V}\| \leq \sqrt{k}\epsilon$.

Let $Z$ be a $k \times k$ matrix whose rows are $z_i$'s. Similarly define matrix $\hat{Z}$ for $\hat{z}_i$'s. Again, we have $\|Z - \hat{Z}\| \leq \sqrt{k}\epsilon$. In order to show $\|Z^{-1} - \hat{Z}^{-1}\|$ is small using standard matrix perturbation bounds (Lemma 29), we need to lower bound $\sigma_{\min}(Z)$. Notice that $Z$ is just matrix $A^{-1}$ with normalized row vectors. As we know, $\sigma_{\min}(A^{-1}) \geq 1/P_1$, and $\|A^{-1}\| \leq 1/\beta$, which implies that every row vector of $A^{-1}$ has norm at most $1/\beta$. Let $D_z$ be the diagonal matrix whose $i, i$-th entry is the norm of $i$-th row of $A^{-1}$, then $Z = D_z^{-1}A^{-1}$, and we know $\sigma_{min}(Z) \geq \sigma_{min}(D_z^{-1})\sigma_{min}(A^{-1}) \geq \beta/P_1$.

Then, according to Lemma 29, as long as $\epsilon \leq \frac{\beta}{2P_1}$, we have

$$
\|Z^{-1} - \hat{Z}^{-1}\| \leq 2\sqrt{2}\frac{\|Z - \hat{Z}\|}{\sigma_{\min}^2(Z)} \leq 2\sqrt{2}P_1^2\sqrt{k}\epsilon/\beta^2.
$$

Define

$$
\begin{aligned}
E_1 :=& Z^{-1} - \hat{Z}^{-1} \\
E_2 :=& V - \hat{V}.
\end{aligned}
$$

We know $\|E_1\| \leq 2\sqrt{2}P_1^2\sqrt{k}\epsilon/\beta^2$ and $\|E_2\| \leq \sqrt{k}\epsilon$. In order to bound $\|V\|$, we can bound the norm of its row vectors. We have,

$$
\begin{aligned}
\|v_i\| =& \|2\mathbb{E}[xx^\top]^{-1}\mathbb{E}[z_i^\top yx]\| \\
\leq& \frac{2\Gamma(\Gamma P_1\sqrt{k} + P_2)}{\gamma},
\end{aligned}
$$

which implies $\|V\| \leq \frac{2\sqrt{k}\Gamma(\Gamma P_1\sqrt{k} + P_2)}{\gamma}$. Now we can bound $\|Z^{-1}\sigma(Vx) - \hat{Z}^{-1}\sigma(\hat{V}x)\|$ as follows.

$$
\begin{aligned}
\|Z^{-1}\sigma(Vx) - \hat{Z}^{-1}\sigma(\hat{V}x)\| \leq& \|E_1\|\|E_2 x\| + \|E_1\|\|Vx\| + \|Z^{-1}\|\|E_2 x\| \\
\leq& \frac{2\sqrt{2}P_1^2\Gamma k\epsilon^2}{\beta^2} + \frac{4\sqrt{2}P_1^2\Gamma^2(\Gamma P_1\sqrt{k} + P_2)k\epsilon}{\beta^2\gamma} + \frac{P_1\Gamma\sqrt{k}\epsilon}{\beta},
\end{aligned}
$$

where the first inequality holds since $\|\sigma(Vx)\| \leq \|Vx\|$ and $\|\sigma(\hat{V}x) - \sigma(Vx)\| \leq \|\hat{V}x - Vx\|$.

Thus, we know given $\text{poly}\big(\Gamma, P_1, P_2, d, 1/\epsilon, 1/\gamma, 1/\alpha, 1/\beta, 1/\delta\big)$ number of i.i.d. samples, with probability at least $1 - \delta$,

$$
\|A\sigma(Wx) - \hat{Z}^{-1}\sigma(\hat{V}x)\| = \|Z^{-1}\sigma(Vx) - \hat{Z}^{-1}\sigma(\hat{V}x)\| \leq \epsilon,
$$

where the first equality holds because $A\sigma(Wx) = Z^{-1}\sigma(Vx)$, as shown in Theorem 5. $\qquad\square$

## C  SMOOTHED ANALYSIS FOR DISTINGUISHING MATRICES

In smoothed analysis, it's clear that after adding small Gaussian perturbations, matrix $A$ and $W$ will become robustly full rank with reasonable probability (Lemma 36). In this section, we will focus on the tricky part, using smoothed analysis framework to show that it is natural to assume the distinguishing matrix is robustly full rank. We will consider two settings. In the first case, the input distribution is the Gaussian distribution $\mathcal{N}(0, I_d)$, and the weights for the first layer matrix $W$ is perturbed by a small Gaussian noise. In this case we show that the augmented distinguishing matrix $M$ has smallest singular value $\sigma_{\min}(M)$ that depends polynomially on the dimension and the amount of perturbation. This shows that for the Gaussian input distribution, $\sigma_{\min}(M)$ is lower bounded as long as $W$ is in general position. In the second case, we will fix a full rank weight matrix $W$, and consider an arbitrary symmetric input distribution $\mathcal{D}$. There is no standard way of perturbing a symmetric distribution, we give a simple perturbation $\mathcal{D}'$ that can be arbitrarily close to $\mathcal{D}$, and prove that $\sigma_{\min}(M^{\mathcal{D}'})$ is lowerbounded.

**Perturbing $W$ for Gaussian Input**  We first consider the case when the input follows standard Gaussian distribution $\mathcal{N}(0, I_d)$. The weight matrix $W$ is perturbed to $\widetilde{W}$ where

$$\widetilde{W} = W + \rho E. \tag{18}$$

Here $E \in \mathbb{R}^{k \times d}$ is a random matrix whose entries are i.i.d. standard Gaussians. We will use $\widetilde{M}$ to denote the perturbed version of the augmented distinguishing matrix $M$. Recall that the columns of $\widetilde{M}$ has the form:

$$\widetilde{M}_{ij} = \mathbb{E}\big[(\widetilde{w}_i^\top x)(\widetilde{w}_j^\top x)(x \otimes x)\mathbb{1}\{\widetilde{w}_i^\top x \widetilde{w}_j^\top x \leq 0\}\big],$$

where $\widetilde{w}_i^\top$ is the $i$-th row of $\widetilde{W}$. Also, since $\widetilde{M}$ is the augmented distinguishing matrix it has a final column $\widetilde{M_0} = \text{vec}(I_d)$. We show that the smallest singular value of $\widetilde{M}$ is lower bounded with high probability.

**Theorem 9.** *Suppose that $k \leq d/5$, and the input follows standard Gaussian distribution $\mathcal{N}(0, I_d)$. Given any weight matrix $W$ with $\|w_i\| \leq \tau$ for each row vector, let $\widetilde{W}$ be a perturbed version of $W$ according to Equation* (18) *and $\widetilde{M}$ be the perturbed augmented distinguishing matrix. With probability at least $1 - \exp(-d^{\Omega(1)})$, we have*

$$\sigma_{\min}(\widetilde{M}) \geq poly(1/\tau, 1/d, \rho).$$

We will prove this Theorem in Section C.1.

**Perturbing the Input Distribution**  Our algorithm works for a general symmetric input distribution $\mathcal{D}$. However, we cannot hope to get a result like Theorem 9 for every symmetric input distribution $\mathcal{D}$. As a simple example, if $\mathcal{D}$ is just concentrated on 0, then we do not get any information about weights and the problem is highly degenerate. Therefore, we must specify a way to perturb the input distribution.

We define a perturbation that is parametrized by a random Gaussian matrix $Q$ and a parameter $\lambda \in (0, 1)$. The random matrix $Q$ is used to generate a Gaussian distribution $\mathcal{D}_Q$ with a random covariance matrix. To sample a point in $\mathcal{D}_Q$, first sample $n \sim \mathcal{N}(0, I_d)$, and then output $Qn$. The $(Q, \lambda)$ perturbation of a distribution $\mathcal{D}$, which we denote by $\mathcal{D}_{Q,\lambda}$ is a mixture between the distribution $\mathcal{D}$ and the distribution $\mathcal{D}_Q$. More precisely, to sample $x$ from $\mathcal{D}_{Q,\lambda}$, pick $z$ as a Bernoulli random variable where $\Pr[z = 1] = \lambda$ and $\Pr[z = 0] = 1 - \lambda$; pick $x'$ according to $\mathcal{D}$ and pick $x'' = Qn$ according to distribution $\mathcal{D}_Q$, then let

$$x = \begin{cases} x' & z = 0 \\ x'' & z = 1 \end{cases}$$

Intuitively, the $(Q, \lambda)$ perturbation of a distribution $\mathcal{D}$ mixes the distribution $\mathcal{D}$ with a Gaussian distribution $\mathcal{D}_Q$ with covariance matrix $QQ^\top$. Since both $\mathcal{D}$ and $\mathcal{D}_Q$ are symmetric, their mixture is also symmetric. Also, the TV-distance between $\mathcal{D}$ and $\mathcal{D}_{Q,\lambda}$ is bounded by $\lambda$. Throughout this section we will use $\mathcal{D}'$ to denote the perturbed distribution $\mathcal{D}_{Q,\lambda}$

We show that given any input distribution, after applying $(Q, \lambda)$-perturbation with a random Gaussian matrix $Q$, the smallest singular value of the augmented distinguishing matrix $M^{\mathcal{D}'}$ is lower bounded. Recall that $M^{\mathcal{D}'}$ is defined as

$$M_{ij}^{\mathcal{D}'} = \mathbb{E}_{x \sim \mathcal{D}'}\big[(w_i^\top x)(w_j^\top x)(x \otimes x)\mathbb{1}\{w_i^\top x w_j^\top x \leq 0\}\big],$$

as the first $\binom{k}{2}$ columns and has $\mathbb{E}_{x \sim \mathcal{D}'}[x \otimes x]$ as the last column.

**Theorem 10.** *Given weight matrix $W$ with $\|w_i\| \leq \tau$ for each row vector and symmetric input distribution $\mathcal{D}$. Suppose that $k \leq d/7$ and $\sigma_{\min}(W) \geq \rho$, after applying $(Q, \lambda)$-perturbations to yield perturbed input distribution $\mathcal{D}'$, where $Q$ is a $d \times d$ matrix whose entries are i.i.d. Gaussians, we have with probability at least $1 - \exp(-d^{\Omega(1)})$ over the randomness of $Q$,*

$$\sigma_{\min}(M^{\mathcal{D}'}) \geq poly(1/\tau, 1/d, \rho, \lambda).$$

We will prove this later in Section C.3.

## C.1 SMOOTHED ANALYSIS FOR GAUSSIAN INPUTS

In this section, we will prove Theorem 9, as restated below:

**Theorem 9.** *Suppose that $k \leq d/5$, and the input follows standard Gaussian distribution $\mathcal{N}(0, I_d)$. Given any weight matrix $W$ with $\|w_i\| \leq \tau$ for each row vector, let $\widetilde{W}$ be a perturbed version of $W$ according to Equation (18) and $\widetilde{M}$ be the perturbed augmented distinguishing matrix. With probability at least $1 - \exp(-d^{\Omega(1)})$, we have*

$$\sigma_{\min}(\widetilde{M}) \geq poly(1/\tau, 1/d, \rho).$$

To prove this theorem, recall the definition of $M_{ij}$:

$$\mathrm{mat}(M_{ij}) = \mathbb{E}\big[(w_i^\top x)(w_j^\top x)(xx^\top)\mathbb{1}\{w_i^\top x w_j^\top x \leq 0\}\big].$$

Since Gaussian distribution is highly symmetric, for every direction $u$ that is orthogonal to both $w_i$ and $w_j$, we have $u^\top \mathrm{mat}(M_{ij})u$ be a constant. We can compute this constant as

$$m_{ij} := \mathbb{E}\big[(w_i^\top x)(w_j^\top x)\mathbb{1}\{w_i^\top x w_j^\top x \leq 0\}\big].$$

This implies that if we consider $\mathrm{mat}(M_{ij}) - m_{ij}I_d$, it is going to be a matrix whose rows and columns are in span of $w_i$ and $w_j$. In fact we can compute the matrix explicitly as the following lemma:

**Lemma 19.** *Suppose input $x$ follows standard Gaussian distribution $\mathcal{N}(0, I_d)$, and suppose weight matrix $W$ has full-row rank, then for any $1 \leq i < j \leq k$, we have*

$$\mathrm{mat}(M_{ij}) = \frac{1}{\pi}\big(\phi_{ij}\cos(\phi_{ij}) - \sin(\phi_{ij})\big)\|w_i\|\|w_j\|I_d + \frac{\phi_{ij}}{\pi}(w_i w_j^\top + w_j w_i^\top)$$
$$- \frac{\sin(\phi_{ij})}{\pi}\big(\frac{\|w_i\|}{\|w_j\|}w_j w_j^\top + \frac{\|w_j\|}{\|w_i\|}w_i w_i^\top\big),$$

*where $0 < \phi_{ij} < \pi$ is the angle between weight vectors $w_i$ and $w_j$.*

Of course, the same lemma would be applicable to $\widetilde{W}$, so we have an explicit formula for $\widetilde{M}_{ij}$. We will bound the smallest singular value using the idea of leave-one-out distance (as previously used in Rudelson & Vershynin (2009)).

**Leave-one-out Distance**    Leave-one-out distance is a metric that is closely related to the smallest singular value but often much easier to estimate.

**Definition 2.** *For a matrix $A \in \mathbb{R}^{d \times n}(d \geq n)$, the leave-one-out distance $d(A)$ is defined to be the smallest distance between a column of $A$ to the span of other columns. More precisely, let $A_i$ be the $i$-th column of $A$ and $S_{-i}$ be the span of all the columns except for $A_i$, then*

$$d(A) := \min_{i \in [n]} \|(I_d - Proj_{S_{-i}})A_i\|.$$

Rudelson & Vershynin (2009) showed that one can lowerbound the smallest singular value of a matrix by its leave-one-out distance.

**Lemma 20** (Rudelson & Vershynin (2009))**.** *For matrix $A \in \mathbb{R}^{d \times n}(d \geq n)$, we always have $d(A) \geq \sigma_{\min}(A) \geq \frac{1}{\sqrt{n}}d(A)$.*

Therefore, to bound $\sigma_{\min}(\widetilde{M})$ we just need to lowerbound $d(\widetilde{M})$. We use the ideas similar to Bhaskara et al. (2014) and Ma et al. (2016). Since every column of $\widetilde{M}$ (except for $\widetilde{M}_0$) is random, we will try to show that even if we condition on all the other columns, because of the randomness in $\widetilde{M}_{ij}$, the distance between $\widetilde{M}_{ij}$ to the span of other columns is large. However, there are several obstacles in this approach:

1. The augmented distinguishing matrix $\widetilde{M}$ has a special column $\widetilde{M}_0 = \text{vec}(I_d)$ that does not have any randomness.

2. The closed form expression for $\widetilde{M}$ (as in Lemma 19) has complicated coefficients that are not linear in the vectors $\widetilde{w}_i$ and $\widetilde{w}_j$.

3. The columns of $\widetilde{M}_{ij}$ are not independent with each other, so if we condition on all the other columns, $\widetilde{M}_{ij}$ is no longer random.

To address the first obstacle, we will prove a stronger version of Lemma 20 that allows a special column.

**Lemma 21.** *Let $A \in \mathbb{R}^{d \times (n+1)}$ $(d \geq n + 1)$ be an arbitrary matrix whose columns are $A_0, A_1, ..., A_n$. For any $i = 1, 2, ..., n$, let $S_{-i}$ be the subspace spanned by all the other columns (including $A_0$) except for $A_i$, and let $d'(A) := \min_{i=1,...,n} \|(I_d - Proj_{S_{-i}})A_i\|$. Suppose the column $A_0$ has norm $\sqrt{d}$ and $A_1, ..., A_n$ has norm at most $C$, then*

$$\sigma_{\min}(A) \geq \min \Big( \sqrt{\frac{nC^2 d}{4nC^2 + d}}, \sqrt{\frac{d}{4n^2 C^2 + nd}} d'(A) \Big).$$

This lemma shows that if we can bound the leave-one-out distance for all but one column, then the smallest singular value of the matrix is still lowerbounded as long as the columns do not have very different norms. We defer the proof to Section C.2.

For the second obstacle, we show that these coefficients are lowerbounded with high probability. Therefore we can condition on the event that all the coefficients are large enough.

**Lemma 22.** *Given weight vectors $w_i$ and $w_j$ with norm $\|w_i\|, \|w_j\| \leq \tau$, let $\widetilde{w}_i = w_i + \rho\varepsilon_i, \widetilde{w}_j = w_j + \rho\varepsilon_j$ where $\varepsilon_i, \varepsilon_j$ are i.i.d. Gaussian random vectors. With probability at least $1 - \exp(-d^{\Omega(1)})$, we know $\|\widetilde{w}_i\| \leq \tau + \sqrt{3\rho^2 d/2}$, $\|\widetilde{w}_j\| \leq \tau + \sqrt{3\rho^2 d/2}$ and*

$$\widetilde{\phi}_{ij} \geq \frac{\sqrt{\rho^2(d-2)}}{\sqrt{2}\tau + \sqrt{3\rho^2 d}},$$

*where $\widetilde{\phi}_{ij}$ is the angle between $\widetilde{w}_i$ and $\widetilde{w}_j$. In particular, if $\widetilde{W} = W + \rho E$ where $E$ is an i.i.d. Gaussian random matrix, with probability at least $1 - \exp(-d^{\Omega(1)})$, for all $i$, $\|\widetilde{w}_i\| \leq \tau + \sqrt{3\rho^2 d/2}$, and for all $i < j$, the coefficient $\widetilde{\phi}_{ij}/\pi$ in front of the term $\widetilde{w}_i \widetilde{w}_j^\top + \widetilde{w}_j \widetilde{w}_i^\top$ is at least $\frac{\sqrt{\rho^2(d-2)}}{(\sqrt{2}\tau + \sqrt{3\rho^2 d})\pi}$.*

This lemma intuitively says that after the perturbation $\widetilde{w}_i$ and $\widetilde{w}_j$ cannot be close to co-linear. We defer the detailed proof to Section C.2.

For the final obstacle, we use ideas very similar to Ma et al. (2016) which decouples the randomness of the columns.

**Proof of Theorem 9.** Let $E_1$ be the event that Lemma 22 does not hold. Event $E_1$ will be one of the bad events (but note that we do not condition on $E_1$ not happening, we use a union bound at the end).

We partition $[d]$ into two disjoint subsets $L_1, L_2$ of size $d/2$. Let $\widetilde{M}'$ be the set of rows of $\widetilde{M}$ indexed by $L_1 \times L_2$. That is, the columns of $\widetilde{M}'$ are

$$\widetilde{M}'_{ij} = \frac{\widetilde{\phi}_{ij}}{\pi}(\widetilde{w}_{i,L_1} \otimes \widetilde{w}_{j,L_2} + \widetilde{w}_{j,L_1} \otimes \widetilde{w}_{i,L_2}) - \frac{\sin(\widetilde{\phi}_{ij})}{\pi}\left(\frac{\|\widetilde{w}_i\|}{\|\widetilde{w}_j\|}\widetilde{w}_{j,L_1} \otimes \widetilde{w}_{j,L_2} + \frac{\|\widetilde{w}_j\|}{\|\widetilde{w}_i\|}\widetilde{w}_{i,L_1} \otimes \widetilde{w}_{i,L_2}\right),$$

for $i < j$, where $\widetilde{w}_{i,L}$ denotes the restriction of vector $\widetilde{w}_i$ to the subset $L$. Note that the restriction of $\text{vec}(I_d)$ to the rows indexed by $L_1 \times L_2$ is just an all zero vector.

We will focus on a column $\widetilde{M}'_{ij}$ with $i < j$ and try to prove it has a large distance to the span of all the other columns. Let $V_{ij}$ be the span of all other columns, which is equal to $V_{ij} = \text{span}\{\widetilde{M}'_{kl} : k < l \wedge (k,l) \neq (i,j)\}$ (note that we do not need to consider $\widetilde{M}_0$ because that column is 0 when restricted to $L_1 \times L_2$.

It's clear that $V_{ij}$ is correlated with $\widetilde{M}'_{ij}$, which is bad for the proof. To get around this problem, we follow the idea of Ma et al. (2016) and define the following subspace that contains $V_{ij}$,

$$\hat{V}_{ij} = \text{span}\left\{\widetilde{w}_{k,L_1} \otimes x, x \otimes \widetilde{w}_{k,L_2}, \widetilde{w}_{j,L_1} \otimes x, x \otimes \widetilde{w}_{i,L_2} \Big| k \notin \{i,j\}, x \in \mathbb{R}^{d/2}\right\}.$$

By definition $V_{ij} \subset \hat{V}_{ij}$, and thus $\hat{V}_{ij}^\perp \subset V_{ij}^\perp$, where $V_{ij}^\perp$ denotes the orthogonal subspace of $V_{ij}$. Observe that $\widetilde{w}_{j,L_1} \otimes \widetilde{w}_{i,L_2}, \widetilde{w}_{j,L_1} \otimes \widetilde{w}_{j,L_2}, \widetilde{w}_{i,L_1} \otimes \widetilde{w}_{i,L_2} \in \hat{V}_{ij}$, thus

$$\text{Proj}_{\hat{V}_{ij}^\perp}\widetilde{M}'_{ij} = \frac{\widetilde{\phi}_{ij}}{\pi}\text{Proj}_{\hat{V}_{ij}^\perp}\left(\widetilde{w}_{i,L_1} \otimes \widetilde{w}_{j,L_2}\right).$$

Note that $\widetilde{w}_{i,L_1} \otimes \widetilde{w}_{j,L_2}$ is independent with $\hat{V}_{ij}$. Moreover, subspace $\hat{V}_{ij}$ has dimension at most $(k-2)d/2+(k-2)d/2+d/2+d/2 = (k-1)d < \frac{4}{5} \cdot \frac{d^2}{4}$. Then by Lemma 31, we know that with probability at least $1 - \exp(-d^{\Omega(1)})$,

$$\text{Proj}_{\hat{V}_{ij}^\perp}\left(\widetilde{w}_{i,L_1} \otimes \widetilde{w}_{j,L_2}\right) \geq \text{poly}(1/d, \rho).$$

Let $E_2$ be the event that this inequality does not hold for some $i, j$.

Let $S_{ij} = \text{span}\{\widetilde{M}_0, \widetilde{M}_{kl} : k < l \wedge (k,l) \neq (i,j)\}$. Now we know when neither bad events $E_1$ or $E_2$ happens, for every pair $i < j$,

$$\begin{aligned}
\text{Proj}_{S_{ij}^\perp}\widetilde{M}_{ij} &\geq \text{Proj}_{V_{ij}^\perp}\widetilde{M}'_{ij} \\
&\geq \text{Proj}_{\hat{V}_{ij}^\perp}\widetilde{M}'_{ij} \\
&= \frac{\widetilde{\phi}_{ij}}{\pi}\text{Proj}_{\hat{V}_{ij}^\perp}\left(\widetilde{w}_{i,L_1} \otimes \widetilde{w}_{j,L_2}\right) \\
&\geq \text{poly}(1/\tau, 1/d, \rho).
\end{aligned}$$

Currently, we have proved that for any $i < j$, the distance between column $\widetilde{M}_{ij}$ and the span of other columns is at least inverse polynomial. To use Lemma 21 we just need to give a bound on the norms of these columns. By Lemma 22, we know when $E_1$ does not happen

$$\forall i, \|\widetilde{w}_i\| \leq \tau + \sqrt{\frac{3\rho^2 d}{2}},$$

where $\tau$ is the uniform upper bound of the norm of every row vector of $W$. Let $\widetilde{\tau} = \tau + \sqrt{\frac{3\rho^2 d}{2}}$, we know $\widetilde{\tau} = \text{poly}(\tau, d, \rho)$.

Thus, we have

$$
\begin{aligned}
\|\widetilde{M}_{ij}\| \leq & \frac{1}{\pi}\big(\widetilde{\phi}_{ij}|\cos(\widetilde{\phi}_{ij})| + \sin(\widetilde{\phi}_{ij})\big)\|\widetilde{w}_i\|\|\widetilde{w}_j\|\sqrt{d} \\
& + \frac{\widetilde{\phi}_{ij}}{\pi}(\|\widetilde{w}_i\|\|\widetilde{w}_j\| + \|\widetilde{w}_j\|\|\widetilde{w}_i\|) \\
& + \frac{\sin(\widetilde{\phi}_{ij})}{\pi}(\|\widetilde{w}_i\|\|\widetilde{w}_j\| + \|\widetilde{w}_j\|\|\widetilde{w}_i\|) \\
\leq & \frac{\widetilde{\tau}^2}{\pi}(\pi+1)\sqrt{d} + 2\widetilde{\tau}^2 + \frac{2\widetilde{\tau}^2}{\pi}.
\end{aligned}
$$

Thus, there exists $C = \text{poly}(\tau, d, \rho)$, such that $\|\widetilde{M}_{ij}\| \leq C$ for every $i < j$. Now applying Lemma 21 immediately gives the result. $\qquad\square$

### C.2   PROOF OF AUXILIARY LEMMAS FOR SECTION C.1

We will first prove the characterization for columns in the augmented distinguishing matrix.

**Proof of Lemma 19.** For simplicity, we start by assuming that every weight vector $w_i$ has unit norm. At the end of the proof we will discuss how to incorporate the norms of $w_i$, $w_j$. Also throughout the proof we will abuse notation to use $M_{ij}$ as its matrix form $\text{mat}(M_{ij})$.

Let $\mathcal{S}_{ij}$ be the subspace spanned by $w_i$ and $w_j$. Let $\mathcal{S}_{ij}^{\perp}$ be the orthogonal subspace of $\mathcal{S}_{ij}$. Let $\{e_1^{(i,j)}, e_2^{(i,j)}\}$ be a set of orthonormal basis for $\mathcal{S}_{ij}$ such that $e_1^{(i,j)} = w_i$ and $\langle e_2^{(i,j)}, w_j \rangle > 0$. We use matrix $S_{ij} \in \mathbb{R}^{d \times 2}$ to represent subspace $\mathcal{S}_{ij}$, which matrix has $e_1^{(i,j)}$ and $e_2^{(i,j)}$ as two columns. Also, let $S_{ij}^{\perp}$ be a $d \times (d-2)$ matrix, whose columns constitute an orthonormal basis of $\mathcal{S}_{ij}^{\perp}$.

Let $\text{Proj}_{S_{ij}} = S_{ij}S_{ij}^{\top}$, and $\text{Proj}_{S_{ij}^{\perp}} = I_d - S_{ij}S_{ij}^{\top}$. Then, we have

$$
\begin{aligned}
M_{ij} &= \text{Proj}_{S_{ij}^{\perp}}M_{ij} + \text{Proj}_{S_{ij}}M_{ij} \\
&= (\text{Proj}_{S_{ij}^{\perp}}M_{ij} + m_{ij}\text{Proj}_{S_{ij}}I_d) + (\text{Proj}_{S_{ij}}M_{ij} - m_{ij}\text{Proj}_{S_{ij}}I_d) \\
&= (\text{Proj}_{S_{ij}^{\perp}}M_{ij} + m_{ij}\text{Proj}_{S_{ij}}I_d) + (\text{Proj}_{S_{ij}}M_{ij} - m_{ij}S_{ij}S_{ij}^{\top})
\end{aligned}
$$

First, we show that

$$
\text{Proj}_{S_{ij}^{\perp}}M_{ij} + m_{ij}\text{Proj}_{S_{ij}}I_d = m_{ij}I_d,
$$

which is equivalent to proving that $\text{Proj}_{S_{ij}^{\perp}}M_{ij} = m_{ij}\text{Proj}_{S_{ij}^{\perp}}I_d$. It's obvious that the column span of $\text{Proj}_{S_{ij}^{\perp}}M_{ij}$ belongs to the subspace $\mathcal{S}_{ij}^{\perp}$. Actually, the row span of $\text{Proj}_{S_{ij}^{\perp}}M_{ij}$ also belongs to the subspace $\mathcal{S}_{ij}^{\perp}$. To show this, let's consider $u^{\top}(\text{Proj}_{S_{ij}^{\perp}}M_{ij})v$, where $u \in \mathcal{S}_{ij}^{\perp}$ and $v \in \mathcal{S}_{ij}$.

$$
\begin{aligned}
u^{\top}(\text{Proj}_{S_{ij}^{\perp}}M_{ij})v &= u^{\top}(I_d - S_{ij}S_{ij}^{\top})M_{ij}v \\
&= (u^{\top} - u^{\top}S_{ij}S_{ij}^{\top})M_{ij}v \\
&= u^{\top}M_{ij}v,
\end{aligned}
$$

where the last equality holds since $u \in \mathcal{S}_{ij}^{\perp}$ is orthogonal to $e_1^{(i,j)}$ and $e_2^{(i,j)}$. We also know that

$$
\begin{aligned}
u^{\top}M_{ij}v &= u^{\top}\mathbb{E}\big[(w_i^{\top}x)(w_j^{\top}x)(xx^{\top})\mathbb{1}\{w_i^{\top}xw_j^{\top}x \leq 0\}\big]v \\
&= \mathbb{E}\big[(w_i^{\top}x)(w_j^{\top}x)(u^{\top}x)(v^{\top}x)\mathbb{1}\{w_i^{\top}xw_j^{\top}x \leq 0\}\big] \\
&= \mathbb{E}\big[(w_i^{\top}x)(w_j^{\top}x)(v^{\top}x)\mathbb{1}\{w_i^{\top}xw_j^{\top}x \leq 0\}\big]\mathbb{E}[(u^{\top}x)] \\
&= 0,
\end{aligned}
$$

where the third equality holds because $u^{\top}x$ is independent with $w_i^{\top}x, w_j^{\top}x$ and $v^{\top}$. Note since $u$ is orthogonal with $w_i, w_j, v$, we know for standard Gaussian vector $x$, random variable $u^{\top}x$ is independent with $w_i^{\top}x, w_j^{\top}x, v^{\top}x$.

Since the column span and row span of $\text{Proj}_{S_{ij}^{\perp}} M_{ij}$ both belong to the subspace $\mathcal{S}_{ij}^{\perp}$, there must exist a $(d-2) \times (d-2)$ matrix $C$, such that $\text{Proj}_{S_{ij}^{\perp}} M_{ij} = S_{ij}^{\perp} C (S_{ij}^{\perp})^{\top}$. We only need to show this matrix $C$ must be $m_{ij} I_{d-2}$. In order to show this, we prove for any $u, v \in \mathcal{S}_{ij}^{\perp}$, $u^{\top}(\text{Proj}_{S_{ij}^{\perp}} M_{ij}) v = m_{ij} u^{\top} v$.

$$
\begin{aligned}
u^{\top}(\text{Proj}_{S_{ij}^{\perp}} M_{ij}) v &= u^{\top} M_{ij} v \\
&= u^{\top} \mathbb{E}\big[(w_i^{\top} x)(w_j^{\top} x)(xx^{\top})\mathbb{1}\{w_i^{\top} x w_j^{\top} x \le 0\}\big] v \\
&= \mathbb{E}\big[(w_i^{\top} x)(w_j^{\top} x)(u^{\top} x)(v^{\top} x)\mathbb{1}\{w_i^{\top} x w_j^{\top} x \le 0\}\big] \\
&= \mathbb{E}\big[(w_i^{\top} x)(w_j^{\top} x)\mathbb{1}\{w_i^{\top} x w_j^{\top} x \le 0\}\big]\mathbb{E}[u^{\top} x v^{\top} x] \\
&= m_{ij} u^{\top} \mathbb{E}[xx^{\top}] v \\
&= m_{ij} u^{\top} v,
\end{aligned}
$$

where the fourth equality holds because $u^{\top} x, v^{\top} x$ are independent with $w_i^{\top} x, w_j^{\top} x$.

Thus, we know

$$
\begin{aligned}
M_{ij} &= (\text{Proj}_{S_{ij}^{\perp}} M_{ij} + m_{ij} \text{Proj}_{S_{ij}} I_d) + (\text{Proj}_{S_{ij}} M_{ij} - m_{ij} S_{ij} S_{ij}^{\top}) \\
&= m_{ij} I_d + (\text{Proj}_{S_{ij}} M_{ij} - m_{ij} S_{ij} S_{ij}^{\top}).
\end{aligned}
$$

Let's now compute the closed form for $m_{ij}$. Recall that

$$
m_{ij} := \mathbb{E}\big[(w_i^{\top} x)(w_j^{\top} x)\mathbb{1}\{w_i^{\top} x w_j^{\top} x \le 0\}\big].
$$

Note, we only need to consider input $x$ within subspace $\mathcal{S}_{ij}$, which subspace has dimension two. Using the polar representation of two-dimensional Gaussian random variables ($r$ is the radius and $\theta$ is the angle), we have

$$
m_{ij} = \frac{1}{2\pi} \int_0^{\infty} r^3 \exp(-\frac{r^2}{2})dr \int_{\frac{\pi}{2}-\phi_{ij}}^{\frac{\pi}{2}} 2\cos(\theta)\cos(\theta + \phi_{ij})d\theta = \frac{1}{\pi}(\phi_{ij}\cos(\phi_{ij}) - \sin(\phi_{ij})).
$$

Next, we compute the closed form of $\text{Proj}_{S_{ij}} M_{ij}$. Note $\text{Proj}_{S_{ij}} M_{ij}$ is symmetric, because $\text{Proj}_{S_{ij}} M_{ij} = M_{ij} - m_{ij} I_d + m_{ij} S_{ij} S_{ij}^{\top}$, and $M_{ij}, I_d$ and $S_{ij} S_{ij}^{\top}$ are all symmetric. It's obvious that the column span of $\text{Proj}_{S_{ij}} M_{ij}$ belongs to subspace $\mathcal{S}_{ij}$. Combined with the fact that $\text{Proj}_{S_{ij}} M_{ij}$ is symmetric, we know the row span of $\text{Proj}_{S_{ij}} M_{ij}$ also belongs to the subspace $\mathcal{S}_{ij}$. Thus, matrix $\text{Proj}_{S_{ij}} M_{ij}$ can be represented as a linear combination of $(e_1^{(i,j)})(e_1^{(i,j)})^{\top}$, $(e_1^{(i,j)})(e_2^{(i,j)})^{\top}, (e_2^{(i,j)})(e_1^{(i,j)})^{\top}$ and $(e_2^{(i,j)})(e_2^{(i,j)})^{\top}$, which means

$$\text{Proj}_{S_{ij}} M_{ij} = c_{11}^{(i,j)}(e_1^{(i,j)})(e_1^{(i,j)})^{\top} + c_{12}^{(i,j)}(e_1^{(i,j)})(e_2^{(i,j)})^{\top} + c_{21}^{(i,j)}(e_2^{(i,j)})(e_1^{(i,j)})^{\top} + c_{22}^{(i,j)}(e_2^{(i,j)})(e_2^{(i,j)})^{\top},$$

where $c_{11}^{(i,j)}, c_{12}^{(i,j)}, c_{21}^{(i,j)}$ and $c_{22}^{(i,j)}$ are four coefficients. Now, we only need to figure out the four coefficients of this linear combination. Similar as the computation for $m_{ij}$, we use polar integration to show that,

$$
\begin{aligned}
c_{11}^{(i,j)} &= \langle Proj_{S_{ij}} M_{ij}, (e_1^{(i,j)})(e_1^{(i,j)})^T \rangle \\
&= \frac{1}{2\pi} \int_0^{\infty} r^5 \exp(-\frac{r^2}{2})dr \int_{\frac{\pi}{2}-\phi_{ij}}^{\frac{\pi}{2}} \big(\cos^3(\theta)\cos(\theta + \phi_{ij}) + \cos(\theta)\cos^3(\theta + \phi_{ij})\big)d\theta \\
&= \frac{1}{4\pi}(12\phi_{ij}\cos(\phi_{ij}) - 9\sin(\phi_{ij}) - \sin(3\phi_{ij})),
\end{aligned}
$$

where the first equality holds because $e_1^{(i,j)}$ is orthogonal with $e_2^{(i,j)}$. Similarly, we can show that

$$
\begin{aligned}
c_{22}^{(i,j)} &= \langle Proj_{S_{ij}} M_{ij}, (e_2^{(i,j)})(e_2^{(i,j)})^T \rangle \\
&= \frac{1}{2\pi} \int_0^{\infty} r^5 \exp(-\frac{r^2}{2})dr \int_{\frac{\pi}{2}-\phi_{ij}}^{\frac{\pi}{2}} \big(\cos(\theta)\cos(\theta + \phi_{ij})\sin^2(\theta) + \cos(\theta)\cos(\theta + \phi_{ij})\sin^2(\theta + \phi_{ij})\big)d\theta \\
&= \frac{1}{4\pi}(4\phi_{ij}\cos(\phi_{ij}) - 7\sin(\phi_{ij}) + \sin(3\phi_{ij})),
\end{aligned}
$$

and

$$
\begin{aligned}
c_{21}^{(i,j)} =& \langle Proj_{S_{ij}} M_{ij}, (e_2^{(i,j)})(e_1^{(i,j)})^T \rangle \\
=& \frac{1}{2\pi} \int_0^\infty r^5 \exp(-\frac{r^2}{2}) dr \int_{\frac{\pi}{2}-\phi_{ij}}^{\frac{\pi}{2}} \big( -\cos^2(\theta)\cos(\theta+\phi_{ij})\sin(\theta) + \cos(\theta)\cos^2(\theta+\phi_{ij})\sin(\theta+\phi_{ij}) \big) d\theta \\
=& \frac{1}{\pi} (\phi_{ij}\sin(\phi_{ij}) - \cos(\phi_{ij})\sin^2(\phi_{ij})).
\end{aligned}
$$

It's easy to check that $c_{12}^{(i,j)} = c_{21}^{(i,j)}$. Let $M'_{ij}$ be $Proj_{S_{ij}} M_{ij} - m_{ij} S_{ij} S_{ij}$. Then, according to above computation, we know

$$
\begin{aligned}
M'_{ij} =& Proj_{S_{ij}} M_{ij} - m_{ij} S_{ij} S_{ij} \\
=& (c_{11}^{(i,j)} - m_{ij})(e_1^{(i,j)})(e_1^{(i,j)})^\top + c_{12}^{(i,j)}(e_1^{(i,j)})(e_2^{(i,j)})^\top + c_{21}^{(i,j)}(e_2^{(i,j)})(e_1^{(i,j)})^\top + (c_{22}^{(i,j)} - m_{ij})(e_2^{(i,j)})(e_2^{(i,j)})^\top \\
=& \frac{1}{4\pi} \big( 8\phi_{ij}\cos(\phi_{ij}) - 5\sin(\phi_{ij}) - \sin(3\phi_{ij}) \big)(e_1^{(i,j)})(e_1^{(i,j)})^\top \\
& + \frac{1}{4\pi} \big( -3\sin(\phi_{ij}) + \sin(3\phi_{ij}) \big)(e_2^{(i,j)})(e_2^{(i,j)})^\top \\
& + \frac{1}{\pi} \big( \phi_{ij}\sin(\phi_{ij}) - \cos(\phi_{ij})\sin^2(\phi_{ij}) \big) \big( (e_1^{(i,j)})(e_2^{(i,j)})^\top + (e_2^{(i,j)})(e_1^{(i,j)})^\top \big).
\end{aligned}
$$

Since $e_1^{(i,j)} = w_i$ and $e_2^{(i,j)} = \frac{1}{\sin(\phi_{ij})} w_j - \cot(\phi_{ij}) w_i$, we can also express $M'_{ij}$ as a linear combination of $w_i w_i^\top, w_j w_j^\top, w_i w_j^\top$ and $w_j w_i^\top$.

$$
\begin{aligned}
M'_{ij} =& \frac{1}{4\pi} \big( 8\phi_{ij}\cos(\phi_{ij}) - 5\sin(\phi_{ij}) - \sin(3\phi_{ij}) \big) w_i w_i^\top \\
& + \frac{1}{4\pi} \big( -3\sin(\phi_{ij}) + \sin(3\phi_{ij}) \big)\big( \frac{w_j}{\sin(\phi_{ij})} - \cot(\phi_{ij})w_i \big)\big( \frac{w_j}{\sin(\phi_{ij})} - \cot(\phi_{ij})w_i \big)^\top \\
& + \frac{1}{\pi} \big( \phi_{ij}\sin(\phi_{ij}) - \cos(\phi_{ij})\sin^2(\phi_{ij}) \big)\big( w_i(\frac{w_j}{\sin(\phi_{ij})} - \cot(\phi_{ij})w_i)^\top + (\frac{w_j}{\sin(\phi_{ij})} - \cot(\phi_{ij})w_i)w_i^\top \big) \\
=& \frac{\phi_{ij}}{\pi}(w_i w_j^\top + w_j w_i^\top) - \frac{\sin(\phi_{ij})}{\pi}(w_j w_j^\top + w_i w_i^\top).
\end{aligned}
$$

Thus,

$$
\begin{aligned}
M_{ij} =& m_{ij} I_d + M'_{ij} \\
=& \frac{1}{\pi}(\phi_{ij}\cos(\phi_{ij}) - \sin(\phi_{ij})) I_d + \frac{\phi_{ij}}{\pi}(w_i w_j^\top + w_j w_i^\top) - \frac{\sin(\phi_{ij})}{\pi}(w_j w_j^\top + w_i w_i^\top).
\end{aligned}
$$

Finally, if the rows $w_i, w_j$ do not have unit norm, let $\bar{w}_i = w_i/\|w_i\|, \bar{w}_j = w_j/\|w_j\|$, we know

$$
\begin{aligned}
m_{ij} &= \mathbb{E}\big[ (w_i^\top x)(w_j^\top x)\mathbb{1}\{w_i^\top x w_j^\top x \le 0\} \big] \\
&= \|w_i\|\|w_j\| \mathbb{E}\big[ (\bar{w}_i^\top x)(\bar{w}_j^\top x)\mathbb{1}\{w_i^\top x w_j^\top x \le 0\} \big] \\
&= \frac{1}{\pi}(\phi_{ij}\cos(\phi_{ij}) - \sin(\phi_{ij}))\|w_i\|\|w_j\|.
\end{aligned}
$$

Here we used the fact that the indicator variable does not change whether we use $w_i, w_j$ or $\bar{w}_i, \bar{w}_j$. Similarly,

$$
\begin{aligned}
M_{ij} &= \mathbb{E}\big[ (w_i^\top x)(w_j^\top x)x x^\top \mathbb{1}\{w_i^\top x w_j^\top x \le 0\} \big] \\
&= \|w_i\|\|w_j\| \mathbb{E}\big[ (\bar{w}_i^\top x)(\bar{w}_j^\top x)x x^\top \mathbb{1}\{w_i^\top x w_j^\top x \le 0\} \big] \\
&= \frac{1}{\pi}(\phi_{ij}\cos(\phi_{ij}) - \sin(\phi_{ij}))\|w_i\|\|w_j\| I_d + \frac{\phi_{ij}}{\pi}(w_i w_j^\top + w_j w_i^\top) - \frac{\sin(\phi_{ij})}{\pi}\big( \frac{\|w_i\|}{\|w_j\|}w_j w_j^\top + \frac{\|w_j\|}{\|w_i\|}w_i w_i^\top \big).
\end{aligned}
$$

$\square$

Now we can prove the lemmas used to handle the two obstacles. First we give the stronger leave-one-out distance bound.

**Proof of Lemma 21.** The smallest singular value of $A$ can be defined as follows:

$$\sigma_{\min}(A) := \min_{u:\|u\|=1} \|Au\|.$$

Suppose $u^* \in \operatorname{argmin}_{u:\|u\|=1} \|Au\|$. Let $u_i^*$ be the coordinate corresponding to the column $A_i$, for $0 \leq i \leq n$. We consider two cases here. If $|u_0^*| \geq \sqrt{\frac{4nC^2}{4nC^2+d}}$, then we have

$$
\begin{aligned}
\sigma_{\min}(A) =& \|Au^*\| \\
=& \Big\| u_0^* A_0 + \sum_{1 \leq i \leq n} u_i^* A_i \Big\| \\
\geq& \big\| u_0^* A_0 \big\| - \Big\| \sum_{1 \leq i \leq n} u_i^* A_i \Big\| \\
\geq& \sqrt{\frac{4nC^2}{4nC^2+d}}\sqrt{d} - \Big( \sum_{1 \leq i \leq n} |u_i^*| \Big)C \\
\geq& \sqrt{\frac{4nC^2 d}{4nC^2+d}} - \sqrt{n}\sqrt{\frac{d}{4nC^2+d}}C \\
=& \sqrt{\frac{nC^2 d}{4nC^2+d}},
\end{aligned}
$$

where the third inequality uses Cauchy-Schwarz inequality.

If $|u_0^*| < \sqrt{\frac{4nC^2}{4nC^2+d}}$, we know $\sum_{1 \leq i \leq n} |u_i^*|^2 \geq \frac{d}{4nC^2+d}$. Let $k \in \operatorname{argmax}_{1 \leq i \leq n} |u_i^*|$. We know that $|u_k^*| \geq \sqrt{\frac{d}{4n^2C^2+nd}}$. Thus,

$$
\begin{aligned}
\sigma_{\min}(A) \geq& \Big\| u_k^* A_k + \sum_{i:i\neq k} u_i^* A_i \Big\| \\
=& |u_k^*| \Big\| A_k + \sum_{i:i\neq k} \frac{u_i^*}{u_k^*} A_i \Big\| \\
\geq& |u_k^*| \big\| (I_d - \operatorname{Proj}_{S_{-k}}) A_k \big\| \\
\geq& |u_k^*| d'(A) \\
\geq& \sqrt{\frac{d}{4n^2C^2+nd}} d'(A).
\end{aligned}
$$

Above all, we know that the smallest singular value of $A$ is lower bounded as follows,

$$\sigma_{\min}(A) \geq \min \Big( \sqrt{\frac{nC^2 d}{4nC^2+d}}, \sqrt{\frac{d}{4n^2C^2+nd}} d'(A) \Big).$$

$\square$

Next we give the bound on the angle between two perturbed vectors $\widetilde{w}_i$ and $\widetilde{w}_j$.

**Proof of Lemma 22.** According to the definition of $\rho$-perturbation, we know $\widetilde{w}_i = w_i + \rho\varepsilon_i$, $\widetilde{w}_j = w_j + \rho\varepsilon_j$, where $\varepsilon_i, \varepsilon_j$ are i.i.d. standard Gaussian vectors. First, we show that with high probability, the projection of $\widetilde{w}_i$ on the orthogonal subspace of $\widetilde{w}_j$ is lower bounded. Denote the subspace spanned by $\widetilde{w}_j$ as $\mathcal{S}_{\widetilde{w}_j}$, and denote the subspace spanned by $\{\widetilde{w}_j, w_i\}$ as $\mathcal{S}_{\widetilde{w}_j \cup w_i}$. Thus, we have

$$
\begin{aligned}
\big\| \operatorname{Proj}_{\mathcal{S}_{\widetilde{w}_j}^{\perp}} \widetilde{w}_i \big\| \geq& \big\| \operatorname{Proj}_{\mathcal{S}_{\widetilde{w}_j \cup w_i}^{\perp}} (w_i + \rho\varepsilon_i) \big\| \\
=& \rho \big\| \operatorname{Proj}_{\mathcal{S}_{\widetilde{w}_j \cup w_i}^{\perp}} \varepsilon_i \big\|
\end{aligned}
$$

where $\mathcal{S}_{\widetilde{w}_j}^{\perp}$ is the orthogonal subspace of $\mathcal{S}_{\widetilde{w}_j}$.

Fix $\varepsilon_j$, then $\mathcal{S}^{\perp}_{\widetilde{w}_j \cup w_i}$ is a fixed subspace of $\mathbb{R}^d$ with dimension $d - 2$. Let $U$ be a $d \times (d - 2)$ matrix, whose columns constitute a set of orthonormal basis for the subspace $\mathcal{S}^{\perp}_{\widetilde{w}_j \cup w_i}$. Thus, it's not hard to check that $\text{Proj}_{\mathcal{S}^{\perp}_{\widetilde{w}_j \cup w_i}} \varepsilon_i \overset{d}{=} U\varepsilon$, where $\varepsilon \in \mathbb{R}^{d-2}$ is a standard Gaussian vector. Denote $Y := \|\text{Proj}_{\mathcal{S}^{\perp}_{\widetilde{w}_j \cup w_i}} \varepsilon_i\|^2 \overset{d}{=} \|U\varepsilon\|^2 = \|\varepsilon\|^2$, which is a chi-squared random variable with $(d - 2)$ degrees of freedom. According to the tail bound for chi-squared random variable, we have

$$\Pr[|\frac{1}{d-2} Y - 1| \geq t] \leq 2e^{\frac{-(d-2)t^2}{8}}, \ \forall t \in (0, 1).$$

Let $t = \frac{1}{2}$, we know that with probability at least $1 - 2\exp(\frac{-(d-2)}{32})$,

$$Y \geq \frac{d - 2}{2}.$$

Thus, we have $\|\text{Proj}_{\mathcal{S}^{\perp}_{\widetilde{w}_j}} \widetilde{w}_i\| \geq \rho\|\text{Proj}_{\mathcal{S}^{\perp}_{\widetilde{w}_j \cup w_i}} \varepsilon_i\| \geq \sqrt{\frac{\rho^2(d-2)}{2}}$. Recall that

$$\|\text{Proj}_{\mathcal{S}^{\perp}_{\widetilde{w}_j}} \widetilde{w}_i\| = \sin(\widetilde{\phi}_{ij})\|\widetilde{w}_i\|.$$

We also know

$$\begin{aligned}
\|\widetilde{w}_i\| =& \|w_i + \rho\varepsilon_i\| \\
\leq& \|w_i\| + \rho\|\varepsilon_i\| \\
=& \tau + \rho\|\varepsilon_i\|,
\end{aligned}$$

where the last equality holds since $\|w_i\| \leq \tau$. Note $\|\varepsilon_i\|^2$ is another chi-squared random variable with $d$ degrees of freedom. Similar as above, we can show that with probability at least $1 - 2\exp(\frac{-d}{32})$,

$$\|\varepsilon_i\|^2 \leq \frac{3d}{2}.$$

By union bound, we know with probability at least $1 - 2\exp(\frac{-d}{32}) - 2\exp(\frac{-(d-2)}{32})$,

$$\sin(\widetilde{\phi}_{ij})\|\widetilde{w}_i\| \geq \sqrt{\frac{\rho^2(d-2)}{2}},$$

$$\|\widetilde{w}_i\| \leq \tau + \sqrt{\frac{3\rho^2 d}{2}}.$$

Combined with the fact that $\widetilde{\phi}_{ij} \geq \sin(\widetilde{\phi}_{ij})$ when $\widetilde{\phi}_{ij} \in [0, \pi]$, we know with probability at least $1 - 2\exp(\frac{-d}{32}) - 2\exp(\frac{-(d-2)}{32})$,

$$\begin{aligned}
\widetilde{\phi}_{ij} \geq& \sin(\widetilde{\phi}_{ij}) \\
=& \frac{\sin(\widetilde{\phi}_{ij})\|\widetilde{w}_i\|}{\|\widetilde{w}_i\|} \\
\geq& \frac{\sqrt{\frac{\rho^2(d-2)}{2}}}{\tau + \sqrt{\frac{3\rho^2 d}{2}}} \\
=& \frac{\sqrt{\rho^2(d - 2)}}{\sqrt{2}\tau + \sqrt{3\rho^2 d}}
\end{aligned}$$

Given $\widetilde{W} = W + \rho E$, where $E$ is an i.i.d. Gaussian matrix, by union bound, we know with probability at least $1 - \exp(-d^{\Omega(1)})$,

$$\forall i, \|\widetilde{w}_i\| \leq \tau + \sqrt{3\rho^2 d/2}$$

$$\forall i < j, \frac{\widetilde{\phi}_{ij}}{\pi} \geq \frac{\sqrt{\rho^2(d - 2)}}{(\sqrt{2}\tau + \sqrt{3\rho^2 d})\pi}$$

$\square$

## C.3  SMOOTHED ANALYSIS FOR GENERAL INPUTS

In this section, we show that starting from any well-conditioned weight matrix $W$, and any symmetric input distribution $\mathcal{D}$, how to perturb the distribution locally to $\mathcal{D}'$ so that the smallest singular value of $M^{\mathcal{D}'}$ is at least inverse polynomial.

Recall the definition of $(Q, \lambda)$-perturbation: we mix the original distribution $\mathcal{D}$ with a distribution $\mathcal{D}_Q$ which is just a Gaussian $\mathcal{N}(0, QQ^\top)$. To create a sample $x$ in $\mathcal{D}'$, with probability $1 - \lambda$ we draw a sample from $\mathcal{D}$; otherwise we draw a standard Gaussian $n \sim \mathcal{N}(0, I_d)$ and let $x = Qn$. We will prove Theorem 10 which we restate below:

**Theorem 10.** *Given weight matrix $W$ with $\|w_i\| \leq \tau$ for each row vector and symmetric input distribution $\mathcal{D}$. Suppose that $k \leq d/7$ and $\sigma_{\min}(W) \geq \rho$, after applying $(Q, \lambda)$-perturbations to yield perturbed input distribution $\mathcal{D}'$, where $Q$ is a $d \times d$ matrix whose entries are i.i.d. Gaussians, we have with probability at least $1 - \exp(-d^{\Omega(1)})$ over the randomness of $Q$,*

$$\sigma_{\min}(M^{\mathcal{D}'}) \geq poly(1/\tau, 1/d, \rho, \lambda).$$

To prove this, let us first take a look at the structure of augmented distinguishing matrix for these distributions. Let $M^{\mathcal{D}}$, $M^{\mathcal{D}_Q}$, $M^{\mathcal{D}'}$ be the augmented distinguishing matrices for distributions $\mathcal{D}$, $\mathcal{D}_Q$ and $\mathcal{D}'$ respectively. Since $\mathcal{D}'$ is a mixture of $\mathcal{D}$ and $\mathcal{D}_Q$, and the augmented distinguishing matrix is defined as expectations over samples, we immediately have

$$M^{\mathcal{D}'} = (1 - \lambda)M^{\mathcal{D}} + \lambda M^{\mathcal{D}_Q}.$$

Our proof will go in two steps. First we will show that $\sigma_{\min}(M^{\mathcal{D}_Q})$ is large. Then we will show that even mixing with $M^{\mathcal{D}}$ will not significantly reduce the smallest singular value, so $\sigma_{\min}(M^{\mathcal{D}'})$ is also large. In addition to the techniques that we developed in Section C.1, we need two ideas that we call *noise domination* and *subspace decoupling* to solve the new challenges here.

**Noise Domination**   First let us focus on $\sigma_{\min}(M^{\mathcal{D}_Q})$. This instance has weight $W$ and input distribution $\mathcal{N}(0, QQ^\top)$. Let $M^{WQ}$ be the augmented distinguishing matrix for an instance with weight $WQ$ and input distribution $\mathcal{N}(0, I_d)$. Our first observation shows that $M^{\mathcal{D}_Q}$ and $M^{WQ}$ are closely related, and we only need to analyze the smallest singular value of $M^{WQ}$. The problem now is very similar to what we did in Theorem 9, except that the weight $WQ$ is not an i.i.d. Gaussian matrix. However, we will still be able to use Theorem 9 as a black-box because the amount of noise in $WQ$ is in some sense dominating the noise in a standard Gaussian. More precisely, we use the following simple claim:

**Claim 1.** *Suppose property $\mathcal{P}$ holds for $\mathcal{N}(\mu, I_d)$ for any $\mu$, and the property $\mathcal{P}$ is convex (in the sense that if $\mathcal{P}$ holds for two distributions it also holds for their mixture), then for any covariance matrix $\Sigma \succeq I_d$, we know $\mathcal{P}$ also holds for $\mathcal{N}(\mu, \Sigma)$.*

Intuitively the claim says that if the property holds for a Gaussian distribution with smaller variance regardless of the mean, then it will also hold for a Gaussian distribution with larger variance. The proof is quite simple:

*Proof.* Let $\Sigma' = \Sigma - I_d$, by assumption we know $\Sigma'$ is still a positive semidefinite matrix. Let $x \sim \mathcal{N}(\mu, \Sigma)$, $x' \sim \mathcal{N}(\mu, \Sigma')$ and $\delta \sim \mathcal{N}(0, I_d)$, by property of Gaussians it is easy to see that $x \stackrel{d}{=} x' + \delta$. Let $d_x$, $d_{x'}$ and $d_\delta$ be the density function for $x, x', \delta$ respectively, then we know for any point $u$

$$d_x(u) = \mathbb{E}_{x' \sim \mathcal{N}(\mu, \Sigma')}[d_\delta(u - x')].$$

That is, $\mathcal{N}(\mu, \Sigma)$ is a mixture of $\mathcal{N}(x', I)$. Since property $\mathcal{P}$ is true for all $\mathcal{N}(x', I)$, it is also true for $\mathcal{N}(\mu, \Sigma)$. $\qquad\qquad\square$

With this claim we can immediately use the result of Theorem 9 to show $\sigma_{\min}(M^{\mathcal{D}_Q})$ is large.

**Subspace Decoupling**    Next we need to consider the mixture $M^{\mathcal{D}'}$. The worry here is that although $\sigma_{\min}(M^{\mathcal{D}_Q})$ is large, mixing with $\mathcal{D}$ might introduce some cancellations and make $\sigma_{\min}(M^{\mathcal{D}'})$ much smaller. To prove that this cannot happen with high probability, the key observation is that in the first step, to prove $\sigma_{\min}(M^{WQ})$ is large we have only used the property of $WQ$. If we let $\bar{Q}$ be the projection of $Q$ to the orthogonal space of row span of $W$, then $\bar{Q}$ is still a Gaussian random matrix even if we condition on the value of $WQ$! Therefore in the second step we will use the additional randomness in $\bar{Q}$ to show that the cancellation cannot happen. The idea of partitioning the randomness of Gaussian matrices has been widely used in analysis of approximate message passing algorithms. The actual proof is more involved and we will need to partition the Gaussian matrix $Q$ into more parts in order to handle the special column in the augmented distinguishing matrix .

Now we are ready to give the full proof of Theorem 10

**Proof of Theorem 10.** Let us first recall the definition of augmented distinguishing matrix: $M^{\mathcal{D}'}$ is a $d^2$ by $(k_2 + 1)$ matrix, where the first $k_2$ columns consist of

$$M_{ij}^{\mathcal{D}'} := \mathbb{E}_{x \sim \mathcal{D}'}\big[(w_i^\top x)(w_j^\top x)(x \otimes x)\mathbb{1}\{w_i^\top x w_j^\top x \le 0\}\big],$$

and the last column is $\mathbb{E}_{x \sim \mathcal{D}'}[x \otimes x]$. According to the definition of $(Q, \lambda)$-perturbation, if we let $\mathcal{D}_Q$ be $\mathcal{N}(0, QQ^\top)$, then we have

$$M^{\mathcal{D}'} = (1 - \lambda)M^{\mathcal{D}} + \lambda M^{\mathcal{D}_Q}.$$

In the first step, we will try to analyze $M^{\mathcal{D}_Q}$. The first $k_2$ columns of this matrix $M^{\mathcal{D}_Q}$ can be written as:

$$\begin{aligned}
M_{ij}^{\mathcal{D}_Q} &= \mathbb{E}_{x \sim \mathcal{D}_Q}\big[(w_i^\top x)(w_j^\top x)(x \otimes x)\mathbb{1}\{w_i^\top x w_j^\top x \le 0\}\big] \\
&= \mathbb{E}_{n \sim \mathcal{N}(0, I_d)}\big[(w_i^\top Qn)(w_j^\top Qn)(Qn \otimes Qn)\mathbb{1}\{w_i^\top Qn w_j^\top Qn \le 0\}\big] \\
&= Q \otimes Q\, \mathbb{E}_{n \sim \mathcal{N}(0, I_d)}\big[(w_i^\top Qn)(w_j^\top Qn)(n \otimes n)\mathbb{1}\{w_i^\top Qn w_j^\top Qn \le 0\}\big]
\end{aligned}$$

for any $i < j$, and the last column is

$$\begin{aligned}
\mathbb{E}_{x \sim \mathcal{D}_Q}[x \otimes x] &= \mathbb{E}_{n \sim \mathcal{N}(0, I_d)}[Qn \otimes Qn] \\
&= Q \otimes Q\, \mathbb{E}_{n \sim \mathcal{N}(0, I_d)}[n \otimes n].
\end{aligned}$$

Except for the factor $Q \otimes Q$, the remainder of these columns are exactly the same as the augmented distinguishing matrix of a network whose first layer weight matrix is $WQ$ and input distribution is $\mathcal{N}(0, I_d)$. We use $M^{WQ}$ to denote the augmented distinguishing matrix of such a network, then we have

$$M^{\mathcal{D}_Q} = Q \otimes Q M^{WQ}.$$

Therefore we can first analyze the smallest singular value of $M^{WQ}$. Let $\widetilde{W} = WQ$. Note that $Q$ is a Gaussian matrix, and $W$ is fixed, so $WQ$ is also a Gaussian random matrix except its entries are not i.i.d. More precisely, there are only correlations within columns of $WQ$, and for any column of $WQ$, the covariance matrix is $WW^\top$. Since the smallest singular value of $W$ is at least $\rho$, we know $\sigma_{\min}(WW^\top) \ge \rho^2$. Let the covariance matrix of $WQ$ be $\Sigma_{WQ} \in \mathbb{R}^{kd \times kd}$, which has smallest singular value at least $\rho^2$. Therefore we know $\Sigma_{WQ} \succeq \rho^2 I_{kd}$. It's not hard to verify that with probability at least $1 - \exp(-d^{\Omega(1)})$, the norm of every row of $WQ$ is upper bounded by $\mathrm{poly}(\tau, d)$. By Claim 1, any convex property that holds for any $\mathcal{N}(0, \rho^2 I_{kd})$ perturbation must also hold for $\Sigma_{WQ}$ [4]. Thus, we know with probability at least $1 - \exp(-d^{\Omega(1)})$,

$$\sigma_{\min}(M^{WQ}) \ge \mathrm{poly}(1/\tau, 1/d, \rho).$$

To prepare for the next step, we will rewrite $M^{WQ}$ as the product of two matrices. According to the closed form of $M_{ij}^{WQ}$ in Lemma 19, we know each column of $M^{WQ}$ can be expressed as a linear combination of $\widetilde{w}_i \otimes \widetilde{w}_j$'s and $\mathrm{vec}(I_d)$. Therefore:

$$M^{WQ} = \Big[\widetilde{W}^\top \otimes \widetilde{W}^\top, \mathrm{vec}(I_d)\Big]R,$$

---

[4]The conclusion of Theorem 9 is clearly convex because it is a probability, and probabilities are linear in terms of mixture of distributions.

where matrix $R$ has dimension $(k^2 + 1) \times (k_2 + 1)$. It's not hard to verify that

$$\sigma_{\min}(M^{WQ}) \leq \left\| \left[ \widetilde{W}^\top \otimes \widetilde{W}^\top, \mathrm{vec}(I_d) \right] \right\| \sigma_{\min}(R).$$

Thus,

$$\sigma_{\min}(R) \geq \frac{\sigma_{\min}(M^{WQ})}{\left\| \left[ \widetilde{W}^\top \otimes \widetilde{W}^\top, \mathrm{vec}(I_d) \right] \right\|}.$$

Note that $W$ is a $k \times d$ matrix with $\|w_i\| \leq \tau$ for every row vector, and $\widetilde{W} = WQ$, where $Q$ is an standard Gaussian matrix. Thus, similar as the proof in Lemma 22, we can show that with probability at least $1 - \exp(-d^{\Omega(1)})$,

$$\left\| \left[ \widetilde{W}^\top \otimes \widetilde{W}^\top, \mathrm{vec}(I_d) \right] \right\| \leq \left\| \left[ \widetilde{W}^\top \otimes \widetilde{W}^\top, \mathrm{vec}(I_d) \right] \right\|_F \leq \mathrm{poly}(\tau, d).$$

Thus, we know

$$\sigma_{\min}(R) \geq \mathrm{poly}(1/\tau, 1/d, \rho).$$

Now we will try to perform the second step using the idea of subspace decoupling. Let $\mathrm{Proj}_W$ be the projection matrix to the row span of $W$, and let $\mathrm{Proj}_{W^\perp} = I_d - \mathrm{Proj}_W$. Let $\bar{Q} = \mathrm{Proj}_{W^\perp} Q$. Let the columns of $U \in \mathbb{R}^{d \times (d-k)}$ be a set of orthonormal basis for the orthogonal subspace $W^\perp$. By symmetry of Gaussian, $\mathrm{Proj}_W Q$ is independent with $\mathrm{Proj}_{W^\perp} Q$. Thus, from now on we will condition on $\mathrm{Proj}_W Q$, and still treat $\mathrm{Proj}_{W^\perp} Q$ as a Gaussian random matrix. More precisely, $\mathrm{Proj}_{W^\perp} Q$ has the same distribution as $UP$, where $P \in \mathbb{R}^{(d-k) \times d}$ is a standard Gaussian matrix.

We further decouple the $P$ part into two subspaces (this is done mostly to handle the special column in augmented distinguishing matrix). Let the rows of $V \in \mathbb{R}^{k \times d}$ be a set of orthonormal basis for the row span of $\widetilde{W} = WQ$. And let the rows of $V^\perp \in \mathbb{R}^{(d-k) \times d}$ be a a set of orthonormal basis for the orthogonal subspace $\widetilde{W}^\perp$. We can then decompose $UP$ into the row span of $V$ and $V^\perp$ as follows,

$$UP \overset{d}{=} UP_1 V^\perp + UP_2 V,$$

where $P_1 \in \mathbb{R}^{(d-k) \times (d-k)}$ and $P_2 \in \mathbb{R}^{(d-k) \times k}$ are two independent standard Gaussian matrices. After this decomposition, we have

$$\bar{Q} \otimes \bar{Q} \left[ \widetilde{W}^\top \otimes \widetilde{W}^\top, \mathrm{vec}(I_d) \right] R$$
$$\overset{d}{=} UP \otimes UP \left[ \widetilde{W}^\top \otimes \widetilde{W}^\top, \mathrm{vec}(I_d) \right] R$$
$$\overset{d}{=} (UP_1 V^\perp + UP_2 V) \otimes (UP_1 V^\perp + UP_2 V) \left[ \widetilde{W}^\top \otimes \widetilde{W}^\top, \mathrm{vec}(I_d) \right] R$$
$$= U \otimes U (P_1 V^\perp + P_2 V) \otimes (P_1 V^\perp + P_2 V) \left[ \widetilde{W}^\top \otimes \widetilde{W}^\top, \mathrm{vec}(I_d) \right] R$$
$$= U \otimes U \left[ (P_1 V^\perp + P_2 V) \widetilde{W}^\top \otimes (P_1 V^\perp + P_2 V) \widetilde{W}^\top, \mathrm{vec}((P_1 V^\perp + P_2 V)(P_1 V^\perp + P_2 V)^\top) \right] R$$
$$= U \otimes U \left[ P_2 V \widetilde{W}^\top \otimes P_2 V \widetilde{W}^\top, \mathrm{vec}(P_1 P_1^\top + P_2 P_2^\top) \right] R,$$

where the last equality holds because the row span of $V^\perp$ is orthogonal to the column span of $\widetilde{W}^\top$.

Now, we go back to matrix $M^{\mathcal{D}'}$. Let $\mathrm{Proj}_{W^\perp \otimes W^\perp}$ be $\mathrm{Proj}_{W^\perp} \otimes \mathrm{Proj}_{W^\perp}$. We have,

$$\sigma_{\min}(M^{\mathcal{D}'}) = \sigma_{\min}\left( (1-\lambda) M^{\mathcal{D}} + \lambda M^{\mathcal{D}_Q} \right)$$
$$= \sigma_{\min}\left( (1-\lambda) M^{\mathcal{D}} + \lambda Q \otimes Q \left[ \widetilde{W}^\top \otimes \widetilde{W}^\top, \mathrm{vec}(I_d) \right] R \right)$$
$$\geq \sigma_{\min}\left( (1-\lambda) \mathrm{Proj}_{W^\perp \otimes W^\perp} M^{\mathcal{D}} + \lambda \mathrm{Proj}_{W^\perp \otimes W^\perp} Q \otimes Q \left[ \widetilde{W}^\top \otimes \widetilde{W}^\top, \mathrm{vec}(I_d) \right] R \right)$$
$$= \sigma_{\min}\left( (1-\lambda) \mathrm{Proj}_{W^\perp \otimes W^\perp} M^{\mathcal{D}} + \lambda \bar{Q} \otimes \bar{Q} \left[ \widetilde{W}^\top \otimes \widetilde{W}^\top, \mathrm{vec}(I_d) \right] R \right)$$
$$= \sigma_{\min}\left( (1-\lambda) \mathrm{Proj}_{W^\perp \otimes W^\perp} M^{\mathcal{D}} + \lambda U \otimes U \left[ P_2 V \widetilde{W}^\top \otimes P_2 V \widetilde{W}^\top, \mathrm{vec}(P_1 P_1^\top + P_2 P_2^\top) \right] R \right)$$

Since $R$ has full row rank, we know that the row span of $\text{Proj}_{W^\perp \otimes W^\perp} M^{\mathcal{D}}$ belongs to the row span of $R$. According to the definition of $U$, it's also clear that the column span of $\text{Proj}_{W^\perp \otimes W^\perp} M^{\mathcal{D}}$ belongs to the column span of $U \otimes U$. Thus, there exists matrix $C \in R^{(d-k)^2 \times (k^2+1)}$ such that

$$\text{Proj}_{W^\perp \otimes W^\perp} M^{\mathcal{D}} = U \otimes UCR.$$

Thus,

$$
\begin{aligned}
\sigma_{\min}(M^{\mathcal{D}'}) &= \sigma_{\min}\big((1-\lambda)M^{\mathcal{D}} + \lambda M^{\mathcal{D}_Q}\big) \\
&\geq \sigma_{\min}\Big((1-\lambda)\text{Proj}_{W^\perp \otimes W^\perp} M^{\mathcal{D}} + \lambda U \otimes U\Big[P_2 V \widetilde{W}^\top \otimes P_2 V \widetilde{W}^\top, \text{vec}(P_1 P_1^\top + P_2 P_2^\top)\Big]R\Big) \\
&= \sigma_{\min}\Big(\lambda U \otimes U\Big(\frac{1-\lambda}{\lambda}C + \big[P_2 V \widetilde{W}^\top \otimes P_2 V \widetilde{W}^\top, \text{vec}(P_1 P_1^\top + P_2 P_2^\top)\big]\Big)R\Big).
\end{aligned}
$$

Note that $C$ only depends on $U$ and $R$, $U$ only depends on $W$, and $R$ only depends on $WQ$. With $WQ$ fixed, $C$ is also fixed. Clearly, $C$ is independent with $P_1$ and $P_2$. For convenience, denote

$$H := \frac{1-\lambda}{\lambda}C + \big[P_2 V \widetilde{W}^\top \otimes P_2 V \widetilde{W}^\top, \text{vec}(P_1 P_1^\top + P_2 P_2^\top)\big].$$

Now, let's prove that the smallest singular value of matrix $H \in \mathbb{R}^{(d-k)^2 \times (k^2+1)}$ is lower bounded using leave-one-out distance. Let's first consider its submatrix $\hat{H}$ which consists of the first $k^2$ columns of $H$. Note that within random matrix $P_2 V \widetilde{W}^\top$, every row are independent with each other. Within each row, the covariance matrix is $\widetilde{W}\widetilde{W}^\top$. Recall that $\widetilde{W}$ is a random matrix whose covariance $\Sigma_{WQ} \succeq \rho^2 I_{kd}$, we can again apply Claim 1 with the property proved in Lemma 37. As a result, with probability at least $1 - \exp(-d^{\Omega(1)})$,

$$\sigma_{\min}(\widetilde{W}) \geq \text{poly}(1/d, \rho).$$

Thus the covariance matrix of each row of $P_2 V \widetilde{W}^\top$ has smallest singular value at least $\gamma := \text{poly}(1/d, \rho)$.

We can view $P_2 V \widetilde{W}^\top$ as the summation of two independent Gaussian matrix, one of which has covariance matrix $\gamma I_{(d-k)k}$. For this matrix, we will do something very similar to Theorem 9 in order to lowerbound its smallest singular value.

**Claim 2.** *For a random matrix $K \in \mathbb{R}^{(d-k) \times k}$ that is equal to $K^o + E$ where $E$ is a Gaussian random matrix whose entries have variance $\gamma$. If $d \geq 7k$, for any subspace $S_C$ that is independent of $K$ and has dimension at most $k^2 + 1$, the leave-one-out distance $d(\text{Proj}_{S_C^\perp} K \otimes K)$ is at least $\text{poly}(\gamma, 1/d)$.*

The proof idea is similar as Theorem 9, and we try to apply Lemma 31 to $K \otimes K$. In the proof we should think of $K := P_2 V \widetilde{W}^\top$, and denote $i$-th column of $K$ as $K_i$. We also think of the space $S_C$ as the column span of $C$.

As we did in Theorem 9, we partition $[d-k]$ into 2 disjoint subsets $L_1$ and $L_2$ of size $(d-k)/2$. Let $\hat{H}'$ be the set of rows of $\hat{H}$ indexed by $L_1 \times L_2$.

We fix a column $\hat{H}'_{ij}$, $i \neq j \in [k]$. Let $S = \text{span}\{\hat{H}'_{kl} : (k,l) \neq (i,j)\}$. It's clear that $S$ is correlated with $\hat{H}'_{ij}$. Let $C'$ be the set of rows of $C$ indexed by $L_1 \times L_2$. Let $S_{C'}$ be the column span of $C'$, which has dimension at most $k^2 + 1$. We define the following subspace that contains $S$,

$$\hat{S} = S_{C'} \cup \text{span}\Big\{K_{j,L_1} \otimes x, x \otimes K_{i,L_2}, K_{l,L_1} \otimes x, x \otimes K_{l,L_2}\Big| x \in \mathbb{R}^{(d-k)/2}, l \notin \{i,j\}\Big\}$$

Therefor by definition $S \subset \hat{S}$, and thus $\hat{S}^\perp \subset S^\perp$, where $S^\perp$ denotes the orthogonal subspace of $S$. Notice that $\hat{H}'_{ij} = K_{i,L_1} \otimes K_{j,L_2} + \frac{\lambda}{1-\lambda}C'_{ij}$ is independent with $\hat{S}$, assuming $C$ is fixed. Moreover, $\hat{S}$ has dimension at most

$$(d-k)/2 + (d-k)/2 + (k-2)(d-k)/2 + (k-2)(d-k)/2 + k^2 + 1 \leq \frac{4}{5}\frac{(d-k)^2}{4},$$

if $k \leq d/7$. Then, according to Lemma 31, we know with probability at least $1 - \exp(-d^{\Omega(1)})$,

$$\left\|\text{Proj}_{\hat{S}^\perp} \hat{H}'_{ij}\right\| \geq \text{poly}(1/d, \rho).$$

For the column $\hat{H}'_{ii}, i \in [k]$, we define subspace $\hat{S}$ slightly different,

$$\hat{S} = S_{C'} \cup \text{span}\{K_{l,L_1} \otimes x, x \otimes K_{l,L_2} \Big| x \in \mathbb{R}^{(d-k)/2}, l \neq i\}.$$

Here the dimension of $\hat{S}$ is also smaller than $(d-k)^2/5$, assuming that $k \leq d/7$. We can similarly show that with probability at least $1 - \exp(-d^{\Omega(1)})$,

$$\left\|\text{Proj}_{\hat{S}^\perp} \hat{H}'_{ii}\right\| \geq \text{poly}(1/d, \rho).$$

Thus, by union bound, we know that the leave-one-out distance of matrix $\hat{H}'$ is lower bounded by $\text{poly}(1/d, \rho)$.

Now, let's add the additional column $\text{vec}(P_1 P_1^\top + P_2 P_2^\top)$ into consideration. For convenience we denote this column by $b$. We will first prove that the vector $b$ has large norm when projected to the orthogonal subspace of columns in $\hat{H}$, then we will combine this with the fact that $\sigma_{\min}(\hat{H})$ is large to show that $\sigma_{\min}(H)$ is also large (this last step is very similar to Lemma 21).

We know matrix $\hat{H}$ only depends on the randomness of $P_2$. Thus, with $P_2$ fixed, all columns in $H$ are fixed except for $b$. Now, we rely on the randomness in $P_1$ to show that the distance between $b$ and the span of other columns in $H$ is lower bounded. In order to get ride of the correlation within column $b$, we also need to consider a subset of its rows indexed by $L_1 \times L_2$, denoted by $b'$. Let the first column of $P_1$ be $p \in \mathbb{R}^{d-k}$, and the submatrix consisting of other columns be $\hat{P}_1$. Let $S_{\hat{H}'}$ be the column span of $\hat{H}'$. Let

$$\hat{S}_{\hat{H}'} = S_{\hat{H}'} \cup S_{C'} \cup \text{span}(\text{vec}(\hat{P}_1 \hat{P}_1^\top)', \text{vec}(P_2 P_2^\top)'),$$

where $\text{vec}(\hat{P}_1 \hat{P}_1^\top)'$ is the restriction of $\text{vec}(\hat{P}_1 \hat{P}_1^\top)$ to the rows indexed by $L_1 \times L_2$. The dimension of $\hat{S}_{\hat{H}'}$ is at most $k^2 + k^2 + 1 + 2 \leq (d-k)^2/12$, assuming that $k \leq d/7$. Clearly,

$$\text{Proj}_{S_{\hat{H}'}^\perp} b' \geq \text{Proj}_{\hat{S}_{\hat{H}'}^\perp} b'$$
$$= \text{Proj}_{\hat{S}_{\hat{H}'}^\perp} p_{L_1} \otimes p_{L_2},$$

where $p_L$ is the restriction of $p$ to rows indexed by $L$. Note that $p_{L_1}$ and $p_{L_2}$ are two independent standard Gaussian vectors. Thus, according to Lemma 31, we know with probability at least $1 - \exp(-d^{\Omega(1)})$, the distance between $b'$ and the column span of $\hat{H}'$ is at least $\text{poly}(1/d)$.

**Claim 3.** *For any matrix $A \in \mathbb{R}^{\frac{(d-k)^2}{4} \times k^2}$ and a vector $v \in \mathbb{R}^{\frac{(d-k)^2}{4}}$, if the leave-one-out distance $d(A) \geq \delta$, $\|\text{Proj}_{A^\perp} b\| \geq \zeta$, and $\|v\| \leq C_1$, let $B \in \mathbb{R}^{\frac{(d-k)^2}{4} \times (k^2+1)}$ be the matrix that is the concatenation of $A$ and $v$, then the leave-one-out distance $d(B) \geq \text{poly}(\zeta, \delta, 1/C_1)$.*

The proof idea is similar as the proof in Lemma 21. In the proof, we should think of $A := \hat{H}', v := b'$ and $B := H'$, where $H'$ is the subset of rows of $H$ indexed by $L_1 \times L_2$. We know that $d(\hat{H}') \geq \delta = \text{poly}(1/d, \rho), \|\text{Proj}_{A^\perp} b\| \geq \zeta = \text{poly}(1/d)$. It's not hard to show that with probability at least $1 - \exp(-d^{\Omega(1)})$,

$$\|b'\| \leq \text{poly}(d).$$

Thus, there exists $C_1 = \text{poly}(d)$, such that $\|b'\| \leq C_1$. We already proved that the leave-one-out distance of $b'$ in $H'$ is lower bounded. We only need to show the leave-one-out distance for the first $k^2$ columns, which are $\hat{H}'_{ij}, i, j \in [k]$.

For any $i, j \in [k]$, the leave-one-out distance for $\hat{H}'_{ij}$ within matrix $H'$ can be expressed as follows

$$\min_{c_{kl}, c_b} \|\hat{H}'_{ij} + \sum_{(k,l) \neq (i,j)} c_{kl} \hat{H}'_{kl} + c_b b'\|$$

Let $\{c^*_{kl}, c^*_b\}$ be one set of the optimal solutions to $\min_{c_{kl}, c_b} \|\hat{H}'_{ij} + \sum_{(k,l) \neq (i,j)} \hat{H}'_{kl} + c_b b'\|$. If $c^*_b = 0$, we immediately have

$$\|\hat{H}'_{ij} + \sum_{(k,l) \neq (i,j)} c^*_{kl} \hat{H}'_{kl} + c^*_b b'\|$$

$$= \|\hat{H}'_{ij} + \sum_{(k,l) \neq (i,j)} c^*_{kl} \hat{H}'_{kl}\|$$

$$\geq \min_{c_{kl}} \|\hat{H}'_{ij} + \sum_{(k,l) \neq (i,j)} c_{kl} \hat{H}'_{kl}\|$$

$$\geq \delta,$$

where the last inequality holds because the leave-one-out distance of matrix $\hat{H}'$ is lower bounded by $\delta$.

If $c^*_b \neq 0$, we need to be more careful. In this case, we have,

$$\|\hat{H}'_{ij} + \sum_{(k,l) \neq (i,j)} c^*_{kl} \hat{H}'_{kl} + c^*_b b'\|$$

$$= |c^*_b| \left\| \frac{1}{c^*_b} \hat{H}'_{ij} + \sum_{(k,l) \neq (i,j)} \frac{c^*_{kl}}{c^*_b} \hat{H}'_{kl} + b' \right\|$$

$$\geq |c^*_b| \zeta,$$

where the last inequality holds because the distance of $b'$ to the column span of $\hat{H}'$ is lower bounded. If $|c^*_b| \geq \frac{\delta}{2C_1}$, we have $\|\hat{H}'_{ij} + \sum_{(k,l) \neq (i,j)} c^*_{kl} \hat{H}'_{kl} + c^*_b b'\| \geq \frac{\delta \zeta}{2C_1}$.

If $|c^*_b| < \frac{\delta}{2C_1}$, we have

$$\|\hat{H}'_{ij} + \sum_{(k,l) \neq (i,j)} c^*_{kl} \hat{H}'_{kl} + c^*_b b'\|$$

$$\geq \|\hat{H}'_{ij} + \sum_{(k,l) \neq (i,j)} c^*_{kl} \hat{H}'_{kl}\| - \|c^*_b b'\|$$

$$\geq \min_{c_{kl}} \|\hat{H}'_{ij} + \sum_{(k,l) \neq (i,j)} c_{kl} \hat{H}'_{kl}\| - |c^*_b| \|b'\|$$

$$\geq \delta - \frac{\delta}{2C_1} C_1$$

$$= \delta/2.$$

Thus, the leave-one-out distance of $H'$ is lower bounded by $\text{poly}(\zeta, \delta, 1/C_1)$. Recall that with probability at least $1 - \exp(-d^{\Omega(1)})$, we have $\delta = \text{poly}(1/d, \rho), \zeta = \text{poly}(1/d), C_1 = \text{poly}(d)$. Thus, we have

$$d(H') \geq \text{poly}(1/d, \rho).$$

According to Lemma 20, we have

$$\sigma_{\min}(H') \geq \frac{1}{\sqrt{k^2 + 1}} d(H')$$

$$\geq \text{poly}(1/d, \rho).$$

Finally, we put everything together. Since $H'$ is a full column rank matrix, we know that $\sigma_{\min}(H) \geq \sigma_{\min}(H') \geq \text{poly}(1/d, \rho)$. By union bound, we know with probability at least $1 - \exp(-d^{\Omega(1)})$,

$$\sigma_{\min}(H) \geq \text{poly}(1/d, \rho)$$
$$\sigma_{\min}(R) \geq \text{poly}(1/\tau, 1/d, \rho).$$

Since $U$ is an orthonormal matrix, we know $\sigma_{\min}(U \otimes U) = 1$. According to Eq. 19, we know with probability at least $1 - \exp(-d^{\Omega(1)})$,

$$
\begin{aligned}
\sigma_{\min}(M^{\mathcal{D}'}) &\geq \sigma_{\min}(\lambda U \otimes UHR) \\
&\geq \lambda \sigma_{\min}(U \otimes U)\sigma_{\min}(H)\sigma_{\min}(R) \\
&\geq \text{poly}(1/\tau, 1/d, \rho, \lambda)
\end{aligned}
$$

where the second inequality holds since all of $U \otimes U$, $H$ and $R$ have full column rank. $\qquad\square$

## D  TOOLS

In this section, we collect some known results on matrix perturbations and concentration bounds. Basically, we used matrix concentration bounds to do the robust analysis and used matrix perturbation bounds to do the smoothed analysis. We also proved several corollaries that are useful in our setting.

### D.1  MATRIX CONCENTRATION BOUNDS

Matrix concentration bounds tell us that with enough number of independent samples, the empirical mean of a random matrix can converge to the mean of this matrix.

**Lemma 23** (Matrix Bernstein; Theorem 1.6 in Tropp (2012)). *Consider a finite sequence $\{Z_k\}$ of independent, random matrices with dimension $d_1 \times d_2$. Assume that each random matrix satisfies*

$$
\mathbb{E}[Z_k] = 0 \text{ and } \|Z_k\| \leq R \text{ almost surely.}
$$

*Define*

$$
\sigma^2 := \max\left\{ \Big\| \sum_k \mathbb{E}[Z_k Z_k^*] \Big\|, \Big\| \sum_k \mathbb{E}[Z_k^* Z_k] \Big\| \right\}.
$$

*Then, for all $t \geq 0$,*

$$
\Pr\left\{ \Big\| \sum_k Z_k \Big\| \geq t \right\} \leq (d_1 + d_2)\exp\left( \frac{-t^2/2}{\sigma^2 + Rt/3} \right).
$$

As a corollary, we have:

**Lemma 24.** *Consider a finite sequence $\{Z_1, Z_2 \cdots Z_m\}$ of independent, random matrices with dimension $d_1 \times d_2$. Assume that each random matrix satisfies*

$$
\|Z_k\| \leq R, \ 1 \leq k \leq m.
$$

*Then, for all $t \geq 0$,*

$$
\Pr\left\{ \Big\| \sum_{k=1}^m (Z_k - \mathbb{E}[Z_k]) \Big\| \geq t \right\} \leq (d_1 + d_2)\exp\left( \frac{-t^2/2}{4mR^2 + (2Rt)/3} \right).
$$

*Proof.* For each $k$, let $Z_k' = Z_k - \mathbb{E}[Z_k]$ be the new random matrices. It's clear that $\mathbb{E}[Z_k'] = 0$ and $\|Z_k'\| \leq 2R$. For the variance,

$$
\sigma^2 \leq \sum_{k=1}^m \|\mathbb{E}[Z_k' Z_k'^*]\| \tag{19}
$$

$$
\leq \sum_{k=1}^m 4R^2 \tag{20}
$$

$$
= 4mR^2. \tag{21}
$$

$$
\tag{22}
$$

Thus, according to Lemma 23, we have

$$
\Pr\left\{ \Big\| \sum_{k=1}^m (Z_k - \mathbb{E}[Z_k]) \Big\| \geq t \right\} = \Pr\left\{ \Big\| \sum_{k=1}^m (Z_k') \Big\| \geq t \right\} \leq (d_1 + d_2)\exp\left( \frac{-t^2/2}{4mR^2 + (2Rt)/3} \right).
$$

$\qquad\square$

### D.2 MATRIX PERTURBATION BOUNDS

**Perturbation Bound for Singular Vectors** For singular vectors, the perturbation is bounded by Wedin's Theorem.

**Lemma 25** (Wedin's theorem; Theorem 4.1, p.260 in Stewart & Sun (1990).). *Given matrices $A, E \in \mathbb{R}^{m \times n}$ with $m \geq n$. Let $A$ have the singular value decomposition*

$$A = [U_1, U_2, U_3] \begin{bmatrix} \Sigma_1 & 0 \\ 0 & \Sigma_2 \\ 0 & 0 \end{bmatrix} [V_1, V_2]^\top$$

*Let $\hat{A} = A + E$, with analogous singular value decomposition. Let $\Phi$ be the matrix of canonical angles between the column span of $U_1$ and that of $\hat{U}_1$, and $\Theta$ be the matrix of canonical angles between the column span of $V_1$ and that of $\hat{V}_1$. Suppose that there exists a $\delta$ such that*

$$\min_{i,j} |[\Sigma_1]_{i,i} - [\Sigma_2]_{j,j}| > \delta \text{ and } \min_i |[\Sigma_1]_{i,i}| > \delta,$$

*then*

$$\| \sin(\Phi) \|^2 + \| \sin(\Theta) \|^2 \leq 2 \frac{\|E\|^2}{\delta^2}.$$

In order to show the robustness of least $k$ right singular vectors of $T$, we combine Wedin's theorem with the following Lemma.

**Lemma 26** (Theorem 4.5, p.92 in Stewart & Sun (1990).). *Let $\Phi$ be the matrix of canonical angles between the column span of $U$ and that of $\hat{U}$, then*

$$\|Proj_{\hat{U}} - Proj_U\| = \| \sin(\Phi) \|.$$

The exact lemma used in our proof is the following corollary in Ge et al. (2015).

**Lemma 27** (Lemma G.5 in Ge et al. (2015)). *Given matrix $A, E \in \mathbb{R}^{m \times n}$ with $m \geq n$. Suppose that the $A$ has rank $k$. Let $\mathcal{S}$ and $\hat{\mathcal{S}}$ be the subspaces spanned by the first $k$ right singular vectors of $A$ and $\hat{A} = A + E$, respectively. Then, we have:*

$$\|Proj_{\hat{S}^\perp} - Proj_{S^\perp}\| \leq \frac{\sqrt{2}\|E\|_F}{\sigma_k(A)}.$$

**Perturbation Bound for pseudo-inverse** With a lowerbound on $\sigma_{\min}(A)$, we can get bounds for the perturbation of pseudo-inverse.

**Lemma 28** (Theorem 3.4 in Stewart (1977)). *Consider the perturbation of a matrix $A \in \mathbb{R}^{m \times n}$: $B = A + E$. Assume that $rank(A) = rank(B) = n$, then*

$$\|B^\dagger - A^\dagger\| \leq \sqrt{2}\|A^\dagger\|\|B^\dagger\|\|E\|.$$

The following corollary is particularly useful for us.

**Lemma 29** (Lemma G.8 in Ge et al. (2015)). *Consider the perturbation of a matrix $A \in \mathbb{R}^{m \times n}$: $B = A + E$ where $\|E\| \leq \sigma_{\min}(A)/2$. Assume that $rank(A) = rank(B) = n$, then*

$$\|B^\dagger - A^\dagger\| \leq 2\sqrt{2}\|E\|/\sigma_{\min}(A)^2.$$

**Perturbation Bound for Tensor** To lowerbound the leave-one-out distance in augmented distinguishing matrix , we use the following Lemma as the main tool.

**Lemma 30** (Theorem 3.6 in Bhaskara et al. (2014)). *For any constant $\delta \in (0, 1)$, given any subspace $\mathcal{V}$ of dimension $\delta d^l$ in $\mathbb{R}^{d^l}$, there exist vectors $v_1, v_2, \cdots, v_r$ in $\mathcal{V}$ with unit norm, such that for random $(\rho/\sqrt{d})$-perturbations $\widetilde{x}^{(1)}, \widetilde{x}^{(2)}, \cdots, \widetilde{x}^{(l)} \in \mathbb{R}^d$ of any vector $x^{(1)}, x^{(2)}, \cdots, x^{(l)} \in \mathbb{R}^d$, we know with probability at least $1 - \exp(-\delta d^{1/(2l)^l})$,*

$$\exists j \in [r], \ \langle v_j, \widetilde{x}^{(1)} \otimes \widetilde{x}^{(2)} \otimes \cdots \otimes \widetilde{x}^{(l)} \rangle \geq \rho^l (\frac{1}{d})^{3^l}.$$

For second-order tensor, we have the following corollary.

**Lemma 31.** *For any constant $\delta \in (0, 1)$, given any subspace $\mathcal{V}$ of dimension $\delta d^2$ in $\mathbb{R}^{d^2}$, there exist vectors $v_1, v_2, \cdots, v_r$ in $\mathcal{V}$ with unit norm, such that for random $(\rho/\sqrt{d})$-perturbations $\widetilde{x}^{(1)}, \widetilde{x}^{(2)} \in \mathbb{R}^d$ of any vector $x^{(1)}, x^{(2)} \in \mathbb{R}^d$, we know with probability at least $1 - \exp(-\delta d^{1/16})$,*

$$\exists j \in [r], \ \langle v_j, \widetilde{x}^{(1)} \otimes \widetilde{x}^{(2)} \rangle \geq \rho^2 (\frac{1}{d})^9.$$

**Perturbation Bound for Eigendecomposition**   Here, We restate some generic results from B-haskara et al. (2014) on the stability of a matrix's eigendecomposition under perturbation. Let $M$ and $\hat{M}$ be two $n \times n$ mtrices such that $M = UDU^{-1}$ and $\hat{M} = M(I + E) + F$.

**Definition 3** (Definition A.1 in Bhaskara et al. (2014)). *Let $sep(D) = \min_{i \neq j} |D_{ii} - D_{jj}|$.*

The following Lemma guarantees that the eigenvalues of $\hat{M}$ are distinct if the perturbation are not too large.

**Lemma 32** (Lemma A.2 in Bhaskara et al. (2014)). *If $\kappa(U)(\|ME\| + \|F\|) < sep(D)/2n$, then the eigenvalues of $\hat{M}$ are distinct and diagonalizable.*

The following Lemma further upperbound the difference between corresponding eigenvectors.

**Lemma 33** (Lemma A.3 in Bhaskara et al. (2014)). *Let $u_1, ..., u_n$ and $\hat{u}_1, ..., \hat{u}_n$ respectively be the eigenvectors of $M$ and $\hat{M}$, ordered by their corresponding eigenvalues. If $\kappa(U)(\|ME\| + \|F\|) < sep(D)/2n$, then for all $i$ we have $\|\hat{u}_i - u_i\| \leq 3 \frac{\sigma_{\max}(E)\sigma_{\max}(D) + \sigma_{\max}(F)}{\sigma_{\min}(U)sep(D)}$.*

In the setting of simultaneous diagonalization, let $N_a = T_a + E_a$ and $N_b = T_b + E_b$, we have
$$N_a N_b^{-1} = T_a T_b^{-1}(I + F) + G,$$
where $F = -E_b(I + T_b^{-1}E_b)^{-1}T_b^{-1}$ and $G = E_a N_b^{-1}$. The following lemma bound the maximum singular value of perturbation matrix $F$ and $G$.

**Lemma 34** (Claim A.5 in Bhaskara et al. (2014)). *$\sigma_{\max}(F) \leq \frac{\sigma_{\max}(E_b)}{\sigma_{\min}(T_b) - \sigma_{\max}(E_b)}$ and $\sigma_{\max}(G) \leq \frac{\sigma_{\max}(E_a)}{\sigma_{\min}(N_b)}$*

**Alignment of Subspace Basis.**   Due to the rotation issue, we cannot conclude that $\|S - \hat{S}\|$ is small even we know $\|SS^\top - \hat{S}\hat{S}^\top\|$ is bounded. The following Lemma shows that after appropriate alignment, $S$ is indeed close to $\hat{S}$.

**Lemma 35** (Lemma 6 in Ge et al. (2017a)). *Given matrices $S, \hat{S} \in \mathbb{R}^{d \times r}$, we have*

$$\min_{Z^\top Z = ZZ^\top = I_r} \|\hat{S} - SZ\|_F^2 \leq \frac{\|SS^\top - \hat{S}\hat{S}^\top\|_F^2}{2(\sqrt{2} - 1)\sigma_r(SS^\top)}$$

### D.3   SMALLEST SINGULAR VALUE OF RANDOM MATRICES

For a random rectangular matrix where each element is an inpdependent Gaussian variable, Rudelson & Vershynin (2009) gives the following result:

**Lemma 36** (Theorem 1.1 in Rudelson & Vershynin (2009)). *Let $A \in R^{m \times n}$ and suppose that $m \geq n$. Assume that the entries of $A$ are independent standard Gaussian variable, then for every $\epsilon > 0$, with probability at least $1 - (C\epsilon)^{m-n+1} + e^{-C'n}$, where $C, C'$ are two absolute constants, we have:*
$$\sigma_n(A) \geq \epsilon(\sqrt{m} - \sqrt{n-1}).$$

However, in our setting, we are more interested in fixed matrices perturbed by Gaussian variables. The smallest singular value of these "perturbed rectangular matrices" can be bounded as follows.

**Lemma 37** (Lemma G.16 in Ge et al. (2015)). *Let $A \in \mathbb{R}^{m \times n}$ and suppose that $m \geq 3n$. If all the entries of $A$ are independently $\rho$-perturbed to yield $\widetilde{A}$, then for any $\epsilon > 0$, with probability at least $1 - (C\epsilon)^{0.25m}$, for some absolute constant $C$, the smallest singular value of $\widetilde{A}$ is bounded below by:*
$$\sigma_n(\widetilde{A}) \geq \epsilon \rho \sqrt{m}.$$

## D.4 Anti-Concentration

We use the anti-concentration property for Gaussian random variables in our proof of Lemma 17.

**Lemma 38** (Anti-concentration in Carbery & Wright (2001)). *Let $x \in \mathbb{R}^n$ be a Gaussian variable $x \in N(0, I)$, for any polynomial $p(x)$ of degree $d$, there exists a constant $\kappa$ such that*

$$\Pr\left[|p(x)| \leq \epsilon\sqrt{Var[p(x)]}\right] \leq \kappa\epsilon^{1/d}.$$

