# OpenReview forum: "Learning Two-layer Neural Networks with Symmetric Inputs"
_ICLR.cc/2019/Conference_

### Official Review · AnonReviewer3 · 2018-10-29
**A Strong Theory Paper**

**Rating:** 7
**Confidence:** 5

**Review:**

This is a strong theory paper and I recommend to accept.

Paper Summary:
This paper studies the problem of learning a two-layer fully connected neural network where both the output layer and the first layer are unknown. In contrast to previous papers in this line which require the input distribution being standard Gaussian, this paper only requires the input distribution is symmetric. This paper proposes an algorithm which only uses polynomial samples and runs in polynomial time.
The algorithm proposed in this paper is based on the method-of-moments framework and several new techniques that are specially designed to exploit this two-layer architecture and the symmetric input assumption.
This paper also presents experiments to illustrate the effectiveness of the proposed approach (though in experiments, the algorithm is slightly modified).

Novelty:
1. This paper extends the key observation by Goel et al. 2018 to higher orders (Lemma 6). I believe this is an important generalization as it is very useful in studying multi-neuron neural networks.
2. This paper proposes the notation, distinguishing matrix, which is a natural concept to study multi-neuron neural networks in the population level.
3. The “Pure Neuron Detector” procedure is very interesting, as it reduces the problem of learning a group of weights to a much easier problem, learning a single weight vector.

Clarity:
This paper is well written.

Major comments:
My major concern is on the requirement of the output dimension. In the main text, this paper assumes the output dimension is the same as the number of neurons and in the appendix, the authors show this condition can be relaxed to the output dimension being larger than the number of neurons. This is a strong assumption, as in practice, the output dimension is usually 1 for many regression problems or the number of classes for classification problems.
Furthermore, this assumption is actually crucial for the algorithm proposed in this paper. If the output dimension is small, then the “Pure Neuron Detection” step does work. Please clarify if I understand incorrectly. If this is indeed the case, I suggest discussing this strong assumption in the main text and listing the problem of relaxing it as an open problem.


Minor comments:
1. I suggest adding the following papers to the related work section in the final version:
https://arxiv.org/abs/1805.06523
https://arxiv.org/abs/1810.02054
https://arxiv.org/abs/1810.04133
https://arxiv.org/abs/1712.00779
These paper are relatively new but very relevant.

2. There are many typos in the references. For example, “relu” should be ReLU.

---

> ### Author Response · Authors · 2018-11-25
> **Thank you for your feedback!**
>
> Thank you very much for reviewing our paper. We really appreciate your positive reviews and insightful comments.
>
> Thanks for pointing out several related papers, we have already added these papers in our updated version. As you mentioned in the review, our technique cannot immediately apply to the case where the dimension of the output is smaller than the number of hidden units. This is a very interesting question, and we leave it as an open problem in the paper.

---

### Official Review · AnonReviewer2 · 2018-11-05
**Requiring a symmetric distribution and ReLU activation functions seems to be too strong.**

**Rating:** 6
**Confidence:** 4

**Review:**

This paper studies the problem of learning the parameters of a two-layer (or one-hidden layer) ReLU network $y=A\sigma(Wx)$, under the assumption that the distribution of $x$ is symmetric. The main technique here is the "pure neuron detector", which is a high-order moment function of a vector. It can be proved that the pure neuron detector is zero if and only if the vector is equal to the row vector of A^{-1}. Hence, we can "purify" the two layer neural network into independent one layer neural networks, and solve the problem easily.

This paper proposes interesting ideas, supported by mathematical proofs. This paper contains analysis of the algorithm itself, analysis of finding z_i's from span(z_i z_i^T), and analysis of the noisy case.
This paper is reasonably well-written in the sense that the main technical ideas are easy to follow, but there are several grammatical errors, some of which I list below. I list my major comments below:

1) [strong assumptions] The result critically depends on the fact that $x$ is symmetric around the origin and the requirement that activation function is a ReLU. Lemma 1, 2, 3 and Lemma 6 in the appendix are based on these two assumptions. For example, the algorithm fails if $x$ is symmetric around a number other than zero or there is a bias term (i.e. $y=A \sigma(Wx+b) + b'$ ). This strong assumptions significantly weaken the general message of this paper. Add a discussion on how to generalize the idea to more general cases, at least when the bias term is present.

2) [sample efficiency] Tensor decomposition methods tend to suffer in sample efficiency, requiring a large number of samples. In the proposed algorithm (Algorithm 2), estimation of $E[y \otimes x^{\otimes 3}]$ and $E[y \otimes y \otimes (x \otimes x)]$ are needed. How is the sample complexity with respect to the dimension? The theory in this paper suggests a poly(d, 1/\epsilon) sample efficiency, but the exponent of the poly is not known. In Section 4.1, the authors talk about the sample efficiency and claim that the sample efficiency is 5x the number of parameters, but this does not match the result in Figure 2. In the left of Figure 2, when d=10, we need no more than 500 samples to get error of W and A very small, but in the right, when d=32, 10000 samples can not give very small error of W and A. I suspect that the required number of samples to achieve small error scales quadratically in the number of parameters in the neural network. Some theoretical or experimental investigation to identify the exponent of the polynomial on d is in order. Also, perhaps plotting in log-y is better for Figure 2.

3) The idea of "purifying" the neurons has a potential to provide new techniques to analyze deeper neural networks. Explain how one might use the "purification" idea for deeper neural networks and what the main challenges are.

Minor comments:

"Why can we efficiently learn a neural network even if we assume one exists?" -> "The question of whether we can efficiently learn a neural network still remains generally open, even when the data is drawn from a neural network."

"with simple input distribution" -> "with a simple input distribution"

---

> ### Author Response · Authors · 2018-11-25
> **Thank you for your review!**
>
> Thanks a lot for your efforts in the review process. We really appreciate your valuable suggestions and detailed comments.
>
> -generalize technique to shifted input or bias term.
>
> Our current technique does not generalize to the case where the input is shifted or there is a bias term in the output. We think this is a very interesting open question and we are now discussing that in the conclusion. Note that many previous works (Goel et al. 2017, Ge et al. 2017) also do not handle bias terms. It has been empirically observed that for many networks fixing the bias to be 0 only makes the performance slightly worse.
>
> -generalize purifying idea to general depth neural network.
>
> Our idea basically removes the last linear layer given good understanding of what happens in previous layers. If there are results that can learn a p-layer network, it is possible that similar ideas could allow it to learn a p+1-layer network whose last layer is linear. However, there are no general algorithms for learning a neural network under symmetric input for p > 1, so we leave this as an open problem.
>
> -sample complexity of the algorithm
>
> We've added a third plot in Figure 2 of the performance of our algorithm as a function of the dimension of A and W. One thing to keep in mind is that the number of parameters also grow quadratically with the dimension of A and W, so we expect the squared error to grow quadratically with the dimension of A and W. To account for this phenomenon, we've plotted the square root of the error normalized by the dimension of A and W so as to more clearly illustrate the extent to which our algorithm's actual performance deteriorates for high-dimensional A and W.
>
> As illustrated by the flatness of the error curves, the performance of our algorithm remains stable as the dimension of A and W grows from 10 to 32. Note that this is much better than what our theory predicts and obtaining tighter sample complexity is an open problem. We believe that truly determining the exponent of our algorithm's asymptotic performance will necessitate evaluating the algorithm with much larger A and W. Such an experiment will require considerable computational resources and is beyond the present scope of the work.

---

### Official Review · AnonReviewer1 · 2018-11-05
**interesting, technical results on learning one hidden layer NN**

**Rating:** 7
**Confidence:** 4

**Review:**

This paper pushes forward our understanding of learning neural networks. The authors show that they can learn a two-layer (one hidden layer) NN, under the assumption that the input distribution is symmetric. The authors convincingly argue that this is not an excessive limitation, particularly in view of the fact that this is intended to be a theoretical contribution. Specifically, the main result of the paper relies on the concept of smoothed analysis. It states that give data generated from a network, the input distribution can be perturbed so that their algorithm then returns an epsilon solution.

The main machinery of this paper is using a tensor approach (method of moments) that allows them to obtain a system of equations that give them their “neuron detector.” The resulting quadratic equations are linearized through the standard lifting approach (making a single variable in the place of products of variables).

This is an interesting paper. As with other papers in this area, it is somewhat difficult to imagine that the results would extend to tell us about guarantees on learning a general depth neural network. Nevertheless, the tools and ideas used are of interest, and while already quite difficult and sophisticated, perhaps do not yet seem stretched to their limits.

---

> ### Author Response · Authors · 2018-11-25
> **Thanks for your review!**
>
> Thanks a lot for your positive feedback! It’s definitely an important question to study general depth neural network, as we also discuss in the conclusions. We hope that our technique could be further improved to help to learn deeper neural networks.

---

### Meta-Review · Area_Chair1 · 2018-12-17
**A very interesting theoretical contribution on learning 1-hidden-layer neural networks**

**Confidence:** 5
**Recommendation:** Accept (Poster)

**Metareview:**

Although the paper considers a somewhat limited problem of learning a neural network with a single hidden layer, it achieves a surprisingly strong result that such a network can be learned exactly (or well approximated under sampling) under weaker assumptions than recent work.  The reviewers unanimously recommended the paper be accepted.  The paper would be more impactful if the authors could clarify the barriers to extending the technique of pure neuron detection to deeper networks, as well as the barriers to incorporating bias to eliminate the symmetry assumption.